# I Predict Therefore I Am: Is Next Token Prediction Enough to Learn Human-Interpretable Concepts from Data?

**Yuhang Liu[1,2], Dong Gong[3], Yichao Cai[2], Erdun Gao[1,2], Zhen Zhang[1,2], Biwei Huang[4], Mingming Gong[5], Anton van den Hengel[1,2], Javen Qinfeng Shi[1,2]**

[1]Responsible AI Research Centre, Australia
[2]Australian Institute for Machine Learning, Adelaide University
[3]School of Computer Science and Engineering, The University of New South Wales
[4]Halıcıoğlu Data Science Institute, University of California San Diego
[5]School of Mathematics and Statistics, The University of Melbourne
`yuhang.liu01@adelaide.edu.au`

**Project:** `https://sites.google.com/view/yuhangliu/projects/ntp`

## Abstract

Recent empirical evidence shows that LLM representations encode human-interpretable concepts. Nevertheless, the mechanisms by which these representations emerge remain largely unexplored. To shed further light on this, we introduce a novel generative model that generates tokens on the basis of such concepts formulated as latent discrete variables. Under mild conditions, even when the mapping from the latent space to the observed space is non-invertible, we establish rigorous identifiability result: the representations learned by LLMs through next-token prediction can be approximately modeled as the logarithm of the posterior probabilities of these latent discrete concepts given input context, up to an linear transformation. This theoretical finding: 1) provides evidence that LLMs capture essential underlying generative factors, 2) offers a unified and principled perspective for understanding the linear representation hypothesis, and 3) motivates a theoretically grounded approach for evaluating sparse autoencoders. Empirically, we validate our theoretical results through evaluations on both simulation data and the Pythia, Llama, and DeepSeek model families.

## 1 Introduction

Large language models (LLMs) are trained on extensive datasets, primarily sourced from the Internet, enabling them to excel in a wide range of downstream tasks, such as language translation, text summarization, and question answering (Zhao et al., 2024; Bommasani et al., 2021; Olah et al., 2020). Despite this success, their internal representations remain largely opaque, a fact that has sparked a series of efforts toward theoretical understanding and practical interpretability. A promising direction arises from recent empirical evidence showing that LLM representations (often called activations in the natural language processing community) encode latent concepts, such as sentiment (Turner et al., 2023) or writing style (Lyu et al., 2023), that align with human-interpretable abstractions (Acerbi & Stubbersfield, 2023; Manning et al., 2020; Sajjad et al., 2022; Sharkey et al., 2025). Uncovering the underlying reasons for this phenomenon, i.e., developing a principled framework that formally links learned representations to latent concepts, holds promise for advancing our understanding of LLMs.

Some recent works attempt to formulate the problem of linking LLM representations to latent concepts through latent variable models (Park et al., 2023; 2024; Rajendran et al., 2024) in which human-interpretable concepts are formulated as latent variables. For example, Park et al. (2023) propose a latent variable model that focuses solely on binary latent concepts. The work of Park et al. (2024) extends this formalization to categorical concepts, but its focus on hierarchical relationships between concepts may not capture other possible structures in textual data. Rajendran et al. (2024)

instead model both latent concepts and, in particular, observed text data, as continuous variables which may deviate from the discrete sense of language. See Appendix A for additional related work.

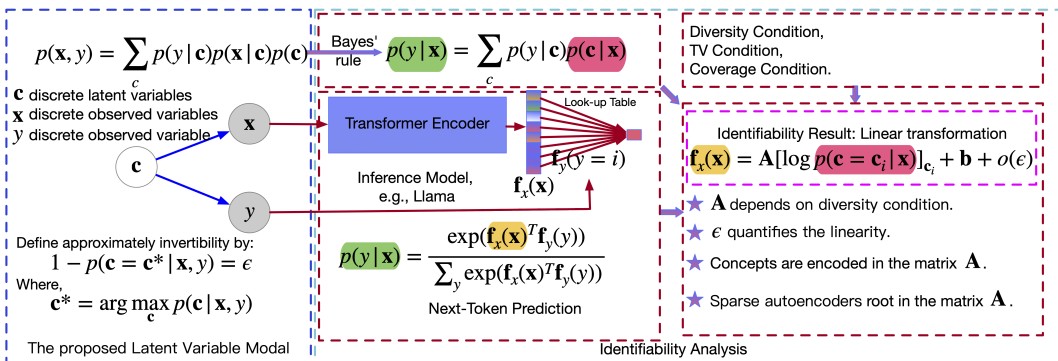

Figure 1: An overview of the main contributions of this work. On the left, we illustrate the proposed latent variable model that represents concepts as latent variables $\mathbf{c}$, which generates both the input $\mathbf{x}$ and output $y$ within a next-token prediction framework. Leveraging Bayes' rule and the certain conditions, we establish an identifiability result: LLM representations are related to a linear transformation of the logarithm of the posterior distribution of latent variables conditioned on input context, i.e., $\mathbf{f_x}(\mathbf{x}) = \mathbf{A}[\log p(\mathbf{c} = \mathbf{c}_i|\mathbf{x})]_i + \mathbf{b} + o(\epsilon)$, where $\mathbf{b}$ is a constant, and $o(\epsilon)$ represents a term that grows asymptotically smaller than $\epsilon$ as $\epsilon \to 0$.

*We investigate here how LLMs, despite relying solely on next-token prediction, can grasp human-interpretable concepts, through a new latent variable model without the limitations above.*

We begin by introducing a latent variable model, illustrated on the left of Figure 1. In this model, human-interpretable concepts are formulated as latent variables connected by arbitrary causal relationships, which in turn generate observed text data through an underlying generative process. Building on this model we develop a theoretical framework to assess whether next-token prediction enables LLMs trained on such text data to recover the underlying latent variables. Through identifiability analysis, we show that, under mild conditions, LLM representations approximately correspond to a *linear* transformation of the logarithm of the posterior distribution of latent variables conditioned on input context. We refer to this as the *linear property* of LLM representations.

Taking a step further, we demonstrate that this *linear property* provides both theoretical and practical insights for *linear* phenomena observed in empirical studies. On the theoretical side, it offers a unified perspective for understanding the various forms from the linear representation hypothesis, e.g., steering vector (Turner et al., 2023) and linear probing (Park et al., 2023; Marks & Tegmark, 2023). On the practical side, the linear property motivates a principled evaluation strategy for sparse autoencoders (SAEs), which aim to extract concepts in an *unsupervised* manner (Huben et al., 2024; Bricken et al., 2023). Specifically, *supervised* concept extraction methods, e.g., linear probing, can serve as an upper bound to evaluate how well SAE features recover the underlying concepts.

**Contributions.** To investigate the relationship between LLM representations and human-interpretable concepts, this paper makes the following contributions. We introduce a novel latent variable model in Sec. 2, in which human-interpretable concepts are formulated as latent variables. Building on this setup, we establish the existence of a *linear* transformation between LLM representations and human-interpretable concepts in Sec. 3. In Sec. 4, we show how the linear property above provides theoretical support for the empirical linear representation hypothesis in LLMs and establishes a unified perspective. In Sec. 5, we show how the *linear* property motivate a principled approach for evaluating SAEs. Additionally, we offer an early exploration of a promising method: structured SAEs, a variant that incorporates structured sparsity, to improve performance under this evaluation. In Sec. 6, we present experiments on simulation data, and real data across the Pythia (Biderman et al., 2023), Llama (Touvron et al., 2023a; Dubey et al., 2024), and DeepSeek-R1 (Guo et al., 2025) models families, with results consistent with our findings. We further present empirical results on the proposed evaluation approach, and the structured SAEs.

## 2    SETUP: LATENT VARIABLE MODEL FOR TEXT DATA GENERATION

We begin by a novel latent variable model designed to capture text data generation process, as depicted on the left in Figure 1. The proposed generative model can be expressed probabilistically as:

$$p(\mathbf{x}, y) = \sum_{\mathbf{c}} p(\mathbf{x}|\mathbf{c})p(y|\mathbf{c})p(\mathbf{c}), \tag{1}$$

where we formulate human-interpretable concepts as latent variables $\mathbf{c}$, and observed variables $\mathbf{x}$ and $y$ represent text generated by the mapping $\mathbf{g}$ on $\mathbf{c}$. We here distinguish observed variables into $\mathbf{x}$ and $y$, to align the input, i.e., context, and output token in next-token prediction framework, respectively.

Our goal is to consider realistic assumptions on the proposed latent variable model, ensuring the model's flexibility to closely approximate real text data. To this end, we do not impose any specific graph structure over latent variables. For example, we allow for any directed acyclic graph structure over latent variables from the perspective of causal representation learning. Most importantly, we highlight the following two factors that distinguish our model from existing latent variable models (Rajendran et al., 2024; Yan et al., 2024; Jin et al., 2020; Jiang et al., 2024), underscoring its novelty.

**Discrete Modeling for Text Data**    Some existing latent variable models (Rajendran et al., 2024; Yan et al., 2024) for text data predominantly assume that both latent and observed variables are continuous, primarily to simplify identifiability analysis. However, this assumption may overlook the largely discrete characteristics of text data. In contrast, we assume throughout this work that all variables, both latent variables $\mathbf{c}$ and observed $\mathbf{x}$ and $y$, are discrete. This assumption can be justified for observed variables $\mathbf{x}$ and $y$, considering the inherently discrete structure of text data. Likewise, latent variables, which represent underlying concepts, can be well-suited to discrete modeling, as they often correspond to categorical distinctions or finite sets of semantic properties commonly observed in text data. For instance, in a topic modeling scenario, latent variables can represent distinct topics, such as "sports", "politics", or "technology", each of which forms a discrete category. This discrete structure aligns with the way humans typically interpret and classify information in text, making the assumption not only intuitive but also practical for real-world applications, as also assumed in prior work (Blei et al., 2003; Jin et al., 2020; Jiang et al., 2024).

**No Invertibility Requirement**    Unlike existing approaches (Jiang et al., 2024; Rajendran et al., 2024), we do not require the mapping $\mathbf{g}$ from latent $\mathbf{c}$ to observed variables $(\mathbf{x}, y)$ to be invertible. Allowing non-invertibility is a deliberate design choice driven by fact that the mapping from latent to observed space is often complex and unknown. Imposing constraints on it may unnecessarily limit its flexibility. Moreover, in the context of text data, there are two key considerations. First, in natural language processing, the mapping from latent to observed space often involves a many-to-one relationship. For example, different combinations of emotional concepts can lead to the same sentiment label. In sentiment analysis, the combination of "positive sentiment" and "excitement" might result in an observed outcome such as "This is amazing!" (Miller, 1995). Second, some latent concepts may not be explicitly manifested in the observed sentence. For instance, a speaker's intent, tone, or implicit connotations may influence how a sentence is constructed, yet remain unobservable in surface-level text. This phenomenon is particularly prominent in pragmatics and discourse analysis, where unspoken contextual factors significantly shape meaning (Levinson, 1983). To formulate approximate invertibility, we define the mapping $\mathbf{g}$ from latent to observed space as follows:

**Definition 2.1.** We define the degree of approximate invertibility of the mapping $\mathbf{g}$ from $\mathbf{c}$ to $(\mathbf{x}, y)$ by introducing an error term $\epsilon$, as follows: $1 - p(\mathbf{c} = \mathbf{c}^*|\mathbf{x}, y) = \epsilon$, where $0 \leq \epsilon < 1$, and $\mathbf{c}^* = \arg\max_{\mathbf{c}} p(\mathbf{c}|\mathbf{x}, y)$ represents the dominant mode of the posterior.

*Remark* 2.2. Such a relaxed condition naturally implies that we may not achieve exact identifiability results, as established in previous works, due to inevitable information loss in the context of non-invertible mappings. Nevertheless, this non-invertibility still allows for considering identifiability in an approximate sense, which will be discussed in Sec. 3.

## 3    THEORY: IDENTIFIABILITY OF THE PROPOSED LATENT VARIABLE MODEL

We now turn to the identifiability analysis of the proposed latent variable model, i.e., whether the latent variables $\mathbf{c}$ can be uniquely recovered (up to an equivalence class) from observed data $\mathbf{x}$ and $y$

under certain assumptions. We conduct this analysis within the next-token prediction framework, a widely adopted and empirically significant paradigm for training LLMs (Brown et al., 2020; Touvron et al., 2023b; Bi et al., 2024; Zhao et al., 2023).

We begin by introducing the general form of next-token prediction, which serves as the foundation for our analysis. The goal of next-token prediction is to learn a model that predicts the conditional distribution over the next token $p(y|\mathbf{x})$, which can be achieved by applying the softmax function over the logits produced by the model's final layer. This process mirrors multinomial logistic regression, where the model approximates the true conditional distribution $p(y|\mathbf{x})$ by minimizing the cross-entropy loss. In theory, assuming sufficient training data, a sufficiently expressive architecture and proper optimization, the model's predictions will converge to the true conditional distribution. We emphasize that these assumptions are standard in the literature on identifiability analyses.

$$p(y|\mathbf{x}) = \frac{\exp\left(\mathbf{f_x}(\mathbf{x})^T \mathbf{f}_y(y)\right)}{\sum_{y_j} \exp\left(\mathbf{f_x}(\mathbf{x})^T \mathbf{f}_y(y_j)\right)}, \tag{2}$$

where $y_j$ denotes a specific value of the observed variable $y$. The function $\mathbf{f_x}(\mathbf{x})$ maps input $\mathbf{x}$ to a representation space, and $\mathbf{f}_y(y)$ corresponds to the model's final-layer weights, i.e., look-up table.

On the other hand, the true $p(y|\mathbf{x})$ can be derived from Eq. 1 using Bayes' rule:

$$p(y|\mathbf{x}) = \sum_{\mathbf{c}} p(y|\mathbf{c})p(\mathbf{c}|\mathbf{x}). \tag{3}$$

By the expression for $p(y|\mathbf{x})$ in both Eq. 2 and Eq. 3, we can align the right-hand sides of these two equations. Taking the logarithm of both sides yields:

$$\mathbf{f_x}(\mathbf{x})^T \mathbf{f}_y(y) - \log\left(\sum_{y_j} \exp(\mathbf{f_x}(\mathbf{x})^T \mathbf{f}_y(y_j))\right) = \log\sum_{\mathbf{c}} p(y|\mathbf{c})p(\mathbf{c}|\mathbf{x}). \tag{4}$$

We now, as shown in Eq. 4, establish an initial connection between the LLM representations $\mathbf{f_x}$ in the inference model (left-hand side) and the latent variables $\mathbf{c}$ in the generative model (right-hand side). To further study this connection, we introduce the following diversity condition.

**Notation Regarding c**   Let $\mathbf{c} = [c^1, \ldots, c^m]$ denote the latent variables, where $c^k \in \mathcal{V}_k$ with $|\mathcal{V}_k| = n_k, k = 1, \ldots, m$, so that $\mathbf{c}$ can take $\ell = \prod_{i=1}^{m} n_k$ possible values. Let $\mathbf{c}_i, i = 1, \ldots, \ell$, denote all possible values of $\mathbf{c}$.

**Diversity Condition**   We assume there exist $\ell' + 1$ values of $y$, where $\ell'$ denotes the representation dimension of LLMs, such that the following matrix is invertible:

$$\hat{\mathbf{L}} = \left[\, \mathbf{f}_y(y_1) - \mathbf{f}_y(y_0),\ \mathbf{f}_y(y_2) - \mathbf{f}_y(y_0),\ \ldots,\ \mathbf{f}_y(y'_\ell) - \mathbf{f}_y(y_0)\,\right] \in \mathbb{R}^{\ell' \times \ell'}.$$

This assumption was originally developed in nonlinear independent component analysis (Hyvarinen & Morioka, 2016; Hyvarinen et al., 2019; Khemakhem et al., 2020), and has been adopted in identifiability analysis for LLMs (Marconato et al., 2024; Roeder et al., 2021; Rajendran et al., 2024). The invertibility of $\hat{\mathbf{L}}$ indicates that there exists a sufficiently diverse set of $y$ values such that the corresponding difference vectors $\mathbf{f}_y(y_j) - \mathbf{f}_y(y_0)$ form a linearly independent set spanning the image of $\mathbf{f}_y$. This assumption is generally mild: as noted by Roeder et al. (2021), the probability that randomly initialized and stochastically updated parameters of $\mathbf{f}_y$ produce linearly dependent difference vectors is effectively zero.

**TV Condition**   For any two tokens in $y_0, y_1, \ldots y_\ell$ in the Diversity Condition above, e.g., $y_i, y_0$ in, we assume:

$$\mathrm{TV}\big(p(\mathbf{c} \mid y_i), p(\mathbf{c} \mid y_0)\big) = \frac{1}{2}\sum_{\mathbf{c}} \big|p(\mathbf{c} \mid y_i) - p(\mathbf{c} \mid y_0)\big| = o\Big(\frac{1}{|\log \epsilon|}\Big),$$

where $\epsilon$ is defined in Definition 2.1, TV denotes the total variation distance. In the context of the latent variable model in Eq. 1, where $y_i$ denotes a single token and $\mathbf{c}$ represents the full latent space, this condition essentially requires that the posterior distribution $p(\mathbf{c} \mid y)$ changes slowly with respect to some token $y$. This seems intuitive. For example, a given token is typically generated by only a small subset of latent concepts, while each latent concept may correspond to many different tokens. Under such a common one-to-many structure, the posterior distributions for different tokens may not fluctuate dramatically.

**Coverage Condition.** While the TV condition controls how much the posterior $p(\mathbf{c} \mid y_i)$ moves across anchor tokens, bounding the remainder additionally requires a mild regularity assumption on the conditional posteriors $p(\mathbf{c} \mid \mathbf{x}, y_i)$ and $p(\mathbf{c} \mid \mathbf{x}, y_0)$. Specifically, we assume that there exists a constant $\delta$ such that

$$\sup_{\mathbf{c}} \big| \log p(\mathbf{c} \mid \mathbf{x}, y_i) - \log p(\mathbf{c} \mid \mathbf{x}, y_0) \big| \ \leq \ \delta.$$

This condition states that, on the latent configurations that are relevant for generating $(\mathbf{x}, y_i)$ and $(\mathbf{x}, y_0)$, the conditional posteriors do not collapse and do not differ by arbitrarily large log-factors. Such a regularity assumption is intuitive: a given token $y_i$ is typically generated by only a small subset of latent factors, and the observed variable $\mathbf{x}$ usually retains sufficient information about these factors. Hence the conditional posteriors $p(\mathbf{c} \mid \mathbf{x}, y_i)$ and $p(\mathbf{c} \mid \mathbf{x}, y_0)$ remain well-behaved and their log-difference shrinks as the generative map becomes more invertible.

**Theorem 3.1.** *Under the diversity condition above, the true latent variables $\mathbf{c}$ are related to LLM representations $\mathbf{f_x}(\mathbf{x})$, which are learned through the next-token prediction framework, by the following relationship:*

$$\mathbf{f_x}(\mathbf{x}) = \mathbf{A}[\log p(\mathbf{c} = \mathbf{c}_i|\mathbf{x})]_i + \mathbf{b} - (\hat{\mathbf{L}}^T)^{-1}\mathbf{h}_y, \tag{5}$$

*where*

- $\mathbf{h}_y = [h_{y_1} - h_{y_0}, ..., h_{y_\ell} - h_{y_0}]$, *with* $h_{y_j} = [p(\mathbf{c} = \mathbf{c}_i|y = y_j)]_i^T [\log p(\mathbf{c} = \mathbf{c}_i|y = y_j, \mathbf{x})]_i$,

- $\mathbf{b} = (\hat{\mathbf{L}}^T)^{-1}[\dots, b(y = y_i) - b(y = y_0), \dots]$, *with* $b(y = y_j) = \mathbb{E}_{p(\mathbf{c}|y=y_j)}[\log p(y = y_j \mid \mathbf{c})]$,

- $\mathbf{A} = (\hat{\mathbf{L}}^T)^{-1}\mathbf{L}$ *with* $\mathbf{L} \in \mathbb{R}^{\ell' \times \ell}$, $L_{j,i} = p(\mathbf{c} = \mathbf{c}_i \mid y = y_j) - p(\mathbf{c} = \mathbf{c}_i \mid y = y_0)$.

*Moreover,*

- *when $\epsilon = 0$, we have:* $\mathbf{f_x}(\mathbf{x}) = \mathbf{A}[\log p(\mathbf{c} = \mathbf{c}_i|\mathbf{x})]_i + \mathbf{b}$,[1]

- *when $\epsilon \to 0$ and $\delta \to 0$, under the TV condition, we have:* $\mathbf{f_x}(\mathbf{x}) \approx \mathbf{A}[\log p(\mathbf{c} = \mathbf{c}_i|\mathbf{x})]_i + \mathbf{b}$.

*Remark* 3.2. This theorem establishes a precise relationship between the LLM representations $\mathbf{f_x}(\mathbf{x})$ and the underlying latent concepts $\mathbf{c}$. This not only provides theoretical grounding for understanding LLM representations, but also offers a unified prospective for the linear representation hypothesis (Sec. 4). Beyond this, it suggests a principled approach for evaluating SAEs (Sec. 5).

## 4 THEORY → INSIGHTS: UNIFYING THE LINEAR HYPOTHESIS

In this section, we demonstrate how the identifiability result presented in Theorem 3.1 supports the linear representation hypothesis in LLMs. To this end, we first briefly introduce the linear representation hypothesis and then explain how it can be understood and unified through the linear matrix $\mathbf{A}$ from our identifiability result.

### 4.1 THE LINEAR REPRESENTATION HYPOTHESIS

The linear representation hypothesis suggests that human-interpretable concepts in LLMs are represented linearly. This idea is supported by empirical evidence in various forms, including:

**Concepts as Directions**: Each concept is represented as a direction determined by the differences (i.e., vector offset) in representations of pairs that vary only in one latent concept of interest, e.g., gender. For instance, Rep("men")-Rep("women") $\approx$ Rep("king")-Rep("queen") (Mikolov et al., 2013; Pennington et al., 2014; Turner et al., 2023).

**Concept Manipulability**: Previous studies have demonstrated that the value of a concept can be altered independently from others by introducing a corresponding steering vector (Li et al., 2024; Wang et al., 2023; Turner et al., 2023). For example, transitioning an output from a false to a truthful

---

[1]Here, $[\cdot]_i$ denotes a vector indexed by all latent configurations $\mathbf{c}_i$, and later we will also use $[\cdot]_{c^k}$ to denote a vector indexed by all possible values of a single concept $c^k$.

answer can be accomplished by adding a vector offset derived from counterfactual pairs that differ solely in the false/truthful concept.

**Linear Probing**: The value of a concept is often measured using a linear probe. For instance, the probability that the output language is French is logit-linear in the representation of the input. In this context, the linear weights can be interpreted as representing the concept of English/French (Park et al., 2023; Marks & Tegmark, 2023).

## 4.2 UNDERSTANDING THE LINEAR REPRESENTATION HYPOTHESIS

The linear representation hypothesis has received growing empirical support in recent years. While recent work has aimed to develop unified frameworks for a deeper understanding of this phenomenon (Park et al., 2023; Marconato et al., 2024; Jiang et al., 2024), our approach seeks to explain it through the lens of identifiability. Before proceeding, we first provide the following definition:

**Definition 4.1.** We define the degree of approximate invertibility of the mapping from the latent variables $\mathbf{c}$ to the observed variables $\mathbf{x}$ by introducing an error term $\epsilon_{\mathbf{x}}$, as follows: $1 - p(\mathbf{c} = \mathbf{c}_{\mathbf{x}}^* | \mathbf{x}) = \epsilon_{\mathbf{x}}$, where $0 < \epsilon_{\mathbf{x}} < 1$, and $\mathbf{c}_{\mathbf{x}}^* = \arg\max_{\mathbf{c}} p(\mathbf{c}|\mathbf{x})$ represents the dominant mode of $p(\mathbf{c}|\mathbf{x})$.

Next, we present two key corollaries derived from our identifiability results.

**Corollary 4.2** (Concepts Are Encoded in the Matrix $\mathbf{A}$). *Suppose that Theorem 3.1 holds, i.e.,* $\mathbf{f}_{\mathbf{x}}(\mathbf{x}) \approx \mathbf{A} \left[\log p(\mathbf{c} = \mathbf{c}_i \mid \mathbf{x})\right]_i + \mathbf{b}$. *Let $\mathbf{x}_0$ and $\mathbf{x}_1$ be a pair that differ only in the $i$-th concept $c^k$.* *Then as $\epsilon_{\mathbf{x}} \to 0$, $\mathbf{f}_{\mathbf{x}}(\mathbf{x}_1) - \mathbf{f}_{\mathbf{x}}(\mathbf{x}_0) \approx \tilde{\mathbf{A}}^k \left(\left[\log p(c^k \mid \mathbf{x}_1) - \log p(c^k \mid \mathbf{x}_0)\right]_{c^k}\right)$, where $\tilde{\mathbf{A}}^k = \mathbf{A}\mathbf{B}^k$,* *where $\mathbf{B}^k \in \{0,1\}^{\ell \times |\mathcal{V}_k|}$ is a binary lifting matrix that broadcasts each entry of $\left[\log p(c^k \mid \mathbf{x})\right]_{c^k}$ to the corresponding index in $\left[\log p(\mathbf{c} = \mathbf{c}_i \mid \mathbf{x})\right]_i$.*

**Understanding Concepts as Directions** The corollary explains why the representation difference Rep("man") − Rep("woman") can be closely approximated by Rep("king") − Rep("queen"). In both the ("man", "woman") and ("king", "queen") pairs, the primary distinguishing factor is the latent concept of gender, while other concepts such as royalty remain largely unchanged. As a result, both Rep("man") − Rep("woman") and Rep("king") − Rep("queen") are driven by the same expression $\tilde{\mathbf{A}}^k \left(\left[\log p(c^k \mid \mathbf{x}_1) - \log p(c^k \mid \mathbf{x}_0)\right]_{c^k}\right)$. In the case of a binary concept $c^k$, the expression is further reduced to a vector direction corresponding to the concept of interest. See Appendix G.1 for details.

**Understanding Concept Manipulability** The corollary also supports the notion that a concept's value can be adjusted by adding a corresponding steering vector, such as Rep("man") − Rep("woman"). Adding the steering vector effectively modifies the original representation to produce a new representation, i.e., $\hat{\mathbf{f}}_{\mathbf{x}} = \mathbf{f}_{\mathbf{x}}(\mathbf{x}) + \alpha\left(\tilde{\mathbf{A}}^k\left(\left[\log p(c^k \mid \mathbf{x}_1) - \log p(c^k \mid \mathbf{x}_0)\right]_{c^k}\right)\right) = \mathbf{A}\left(\left[\log p(\mathbf{c} = \mathbf{c}_i|\mathbf{x})\right]_i + \alpha\mathbf{B}^k\left(\left[\log p(c^k \mid \mathbf{x}_1) - \log p(c^k \mid \mathbf{x}_0)\right]_{c^k}\right)\right) + \mathbf{b}$, where $\alpha$ represents an introduced weight (Wang et al., 2023; Turner et al., 2023). Thus, manipulating a concept via a steering vector is, in essence, equivalent to modifying the posterior distribution of the concept of interest given $\mathbf{x}$, which directly impacts the model's output.

To understand **Linear Probing**, we first introduce the following corollary:

**Corollary 4.3** (Linear Classifiability of Representations). *Suppose that Theorem 3.1 holds, i.e.,* $\mathbf{f}_{\mathbf{x}}(\mathbf{x}) \approx \mathbf{A} \left[\log p(\mathbf{c} = \mathbf{c}_i \mid \mathbf{x})\right]_i + \mathbf{b}$. *Let $\mathbf{x}_0$ and $\mathbf{x}_1$ be pair data that differ only in the $i$-th concept variable $c^k$. Then, as $\epsilon_{\mathbf{x}} \to 0$, the corresponding representations $(\mathbf{f}_{\mathbf{x}}(\mathbf{x}_0), \mathbf{f}_{\mathbf{x}}(\mathbf{x}_1))$ are linearly separable along the variation of $c^k$. In particular, there exists a linear classifier with weight matrix $\mathbf{W}$ such that $\mathbf{W}\tilde{\mathbf{A}}^k \approx \mathbf{I}$, and the corresponding logit is $\left[p(c^k \mid \mathbf{x})\right]_{c^k}$.*

This corollary supports that latent concepts, e.g., English vs. French, can be reliably classified using a linear probing on the model's representations. This linear separability enables alignment between the model's predictive distribution and the true class distribution, achieved via cross-entropy minimization. A specific analysis for the binary case of concept $c^k$ is provided in Appendix G.2.

## 4.3 UNIFYING THE LINEAR REPRESENTATION HYPOTHESIS

Taken together, Corollary 4.2 shows that the linear transformation $\mathbf{A}$ underlies both concepts as directions and concept manipulability, while Corollary 4.3 demonstrates that linear probing is

supported by a classifier $\mathbf{W}$ satisfying $\mathbf{W}\tilde{\mathbf{A}}^k \approx \mathbf{I}$. This establishes a unified view on understanding the linear representation hypothesis: the various forms are all connected through the same underlying linear matrix $\mathbf{A}$, providing a unified theoretical perspective on how LLM representations linearly encode concepts. Importantly, according to Theorem 3.1, the matrix is defined as $\mathbf{A} = (\hat{\mathbf{L}}^T)^{-1}\mathbf{L}$, which, in essence, arises from the data diversity condition outlined in **Diversity Condition**.

## 5 THEORY → PRACTICE: EVALUATING SAES AND STRUCTURED SAES

**Evaluating SAEs**   Broadly speaking, SAEs are designed with two primary objectives. First, they aim to learn a set of latent features $\mathbf{z}$ such that sparse linear combinations $\boldsymbol{\beta}\mathbf{z}$ can accurately reconstruct LLM representations, i.e., $\mathbf{f}_\mathbf{x}(\mathbf{x}) \approx \boldsymbol{\beta}\mathbf{z}$. Second, they seek to ensure that each learned feature $z_i$ corresponds to an monosemantic, human-interpretable concept, thereby enabling a mechanistic understanding of LLMs. While reconstruction loss is commonly used to assess how well representations are reconstructed (Rajamanoharan et al., 2024a;b; Gao et al., 2025; Braun et al., 2024), it is a limited proxy for the second objective. A key challenge lies in the absence of ground truth for the underlying concepts (Kantamneni et al., 2025), making the evaluation of feature disentanglement challenging.

To address this issue, we propose a new evaluation method for SAEs grounded in our theoretical insights. Specifically, based on Theorem 3.1, the LLM representations $\mathbf{f}_\mathbf{x}(\mathbf{x})$ can be approximated as $\mathbf{f}_\mathbf{x}(\mathbf{x}) \approx \mathbf{A}\left[\log p(\mathbf{c} = \mathbf{c}_i \mid \mathbf{x})\right]_i$ [2]. Meanwhile, SAEs are trained to reconstruct $\mathbf{f}_\mathbf{x}(\mathbf{x})$ by $\boldsymbol{\beta}\mathbf{z}$. Combining the two, we arrive at: $\boldsymbol{\beta}\mathbf{z} \approx \mathbf{A}\left[\log p(\mathbf{c} = \mathbf{c}_i \mid \mathbf{x})\right]_i$. This suggests that SAE features $\mathbf{z}$ are linearly related to $\left[\log p(\mathbf{c} = \mathbf{c}_i \mid \mathbf{x})\right]_i$. Therefore, if we expect each latent dimension $z_i$ to encode only a single concept $c^k$, it should depend only on the posterior of that concept $\log p(c^k \mid \mathbf{x})$. Consequently, we can evaluate whether each $z_i$ has successfully learned a monosemantic concept by measuring its linear correlation with $\log p(c^k \mid \mathbf{x})$. The question is: how can we obtain $p(c^k \mid \mathbf{x})$?

Based on Corollary 4.3, this can be achieved using paired data that differ only in the $k$-th concept variable $c^k$. Specifically, we can construct paired data $(\mathbf{x}_0, \mathbf{x}_1)$ that differ in only a single binary concept $c^k$, with labels $c^k = 0$ for $\mathbf{x}_0$ and $c^k = 1$ for $\mathbf{x}_1$. We then train a linear classifier in a supervised manner on the corresponding LLM representations $\mathbf{f}_\mathbf{x}(\mathbf{x}_0)$ and $\mathbf{f}_\mathbf{x}(\mathbf{x}_1)$, with the goal of predicting their labels. Once the classifier is trained, the resulting logit provides a estimate of the posterior probability $p(c^k = 1 \mid \mathbf{x})$. See Appendix I.2 for more rigorous details.

**Structured SAEs**   Building on the evaluation method described above, our experiments (See Figure 4) suggest that sparsity regularization alone may may not be sufficient to fully disentangle the underlying concepts in LLM representations. Revisiting Theorem 3.1, the identifiability result depends on the proposed latent variable model. In this model, the complex dependencies in text data are encoded by the latent variables, whose interdependencies capture the underlying structure of the data. That is, latent variables are likely to exhibit strong interdependencies. Motivated by this, we propose exploring structured SAEs that incorporate additional regularization beyond sparsity to encourage learned features to model potential relationships among concepts.

In particular, we experiment with low-rank structures to complement sparsity, while noting that other forms of structured regularization could also be considered. Formally, structured SAEs minimize the following objective:

$$\mathcal{L} = \mathbb{E}_{\mathbf{x} \sim \mathcal{D}_{\text{train}}} \left[ \|\mathbf{f}_\mathbf{x}(\mathbf{x}) - \bar{\mathbf{f}}_\mathbf{x}(\mathbf{x})\|_2^2 + \lambda_t \left( \|\mathbf{S}\|_{p_t}^{p_t} + \gamma \|\mathbf{R}\|_{\text{nuc}} \right) \right], \tag{6}$$

where $\bar{\mathbf{f}}_\mathbf{x}(\mathbf{x})$ denotes reconstruction, $\lambda_t$ is the dynamically adjusted sparsity coefficient at step $t$, following the $p$-annealing SAE strategy (Karvonen et al., 2024), $\gamma$ is a hyperparameter that balances the sparsity penalty and the low-rank regularization, the learned features $\mathbf{z} = \mathbf{S} + \mathbf{R}$, $\|\mathbf{S}\|_{p_t}^{p_t} = \sum_i |S_i|^{p_t}$ is the adaptive $L_{p_t}$ norm promoting sparsity, $\|\mathbf{R}\|_{\text{nuc}}$ is the nuclear norm, used to enforce a low-rank structure on $\mathbf{R}$. See Appendix I.1 for more implementation details.

## 6 EMPIRICAL EVALUATION ON SIMULATED AND REAL DATA WITH LLMS

**Simulation**   We begin by conducting experiments on synthetic data, which is generated through the following process: First, we create random directed acyclic graphs (DAGs) with $n$ latent variables,

---

[2]For simplicity, we omit the constant term $\mathbf{b}$ in this section, which does not affect the subsequent analysis.

representing concepts. For each random DAG, the conditional probabilities of each variable given its parents are modeled using Bernoulli distributions, where the parameters are sampled uniformly from $[0.2, 0.8]$. To simulate a nonlinear mixture process, we then convert the latent variable samples into one-hot format and randomly apply a permutation matrix to the one-hot encoding, generating one-hot observed samples. These are then transformed into binary observed samples. To simulate next-token prediction, we randomly mask a part of the binary observed data, e.g.,, $x_i$, and predict it by use the remaining portion $\mathbf{x}_{\backslash i}$. Refer to Appendix H.1 for more details.

**Evaluation** In Theorem 3.1, we demonstrate that the representations learned by next-token prediction approximate a linear transformation of $\log p(\mathbf{c}|\mathbf{x})$, and this approximation becomes tighter when the mapping from $\mathbf{c}$ to $\mathbf{x}$ is approximately invertible. Building on this, Corollary 4.3 establishes that for a data pair $(\mathbf{c}, \mathbf{x})$ differing only in the concept of interest, the representations $\mathbf{f_x}(\mathbf{x})$ is linearly separable. Therefore, to validate Theorem 3.1, we can assess the degree to which the learned representations can be classified linearly for data pairs $(\mathbf{c}, \mathbf{x})$.

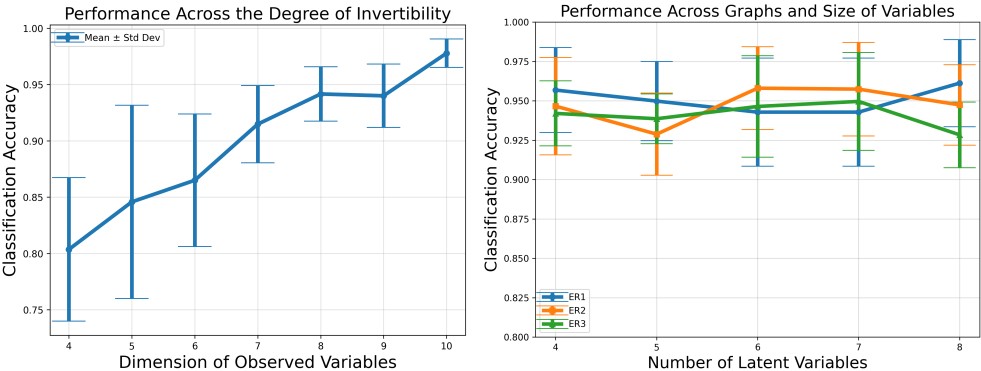

Figure 2: Classification accuracy vs. number of observed variables (left) and graph structures (right).

Our initial experiments investigate the relationship between the degree of invertibility of the mapping from $\mathbf{c}$ to $\mathbf{x}$ and the approximation of the identifiability result in Theorem 3.1. This exploration provides empirical insights into how invertibility influences the recovery of latent variables. To this end, we fix the size of the latent variables and gradually increase the size of the observed variables, thereby enhancing the degree of invertibility of the mapping from $\mathbf{c}$ to $\mathbf{x}$. The left of Figure 2 demonstrates that classification accuracy improves as the size of the observed variables $\mathbf{x}$ increases, aligning with results in Theorem 3.1.

We then examine the impact of latent graph structures on our identifiability results. To this end, we randomly generate DAG structures imposed on $\mathbf{c}$. Specifically, random Erdős-Rényi (ER) graphs (ERDdS & R&wi, 1959) are generated with varying numbers of expected edges. For instance, ER$k$ denotes graphs with $d$ nodes and $kd$ expected edges. The right panel of Figure 2 illustrates the relationship between classification accuracy and the size of the latent variables $\mathbf{c}$ under different settings, including ER1, ER2, and ER3. The results demonstrate that our identifiability findings hold consistently across various graph structures and latent variable sizes, as evidenced by the linear classification accuracy.

**Experiments with LLMs** We now present experiments on pre-trained LLMs. While it is challenging to collect all latent variables from real-world data to directly evaluate our identifiability result in Thorem 3.1, we can instead assess our corollaries to indirectly validate it. For Corollary 4.2, prior studies have already demonstrated the linear representation properties of LLM embeddings using counterfactual pairs that differ only in a single concept of interest (Mikolov et al., 2013; Pennington et al., 2014; Turner et al., 2023; Li et al., 2024; Wang et al., 2023; Park et al., 2023). Therefore, we shift our focus to a new property highlighted in Corollary 4.3, specifically the relationship between $\mathbf{W}$ and $\tilde{\mathbf{A}}^k$, which has not been explored in prior work. To investigate this, we require counterfactual pairs that differ only in a single concept. *Constructing such counterfactual sentences is highly non-trivial, even for human annotators, due to the complexities of semantics and the need for precise control over contextual variations, as noted in prior studies (Park et al., 2023; Jiang et al., 2024).*

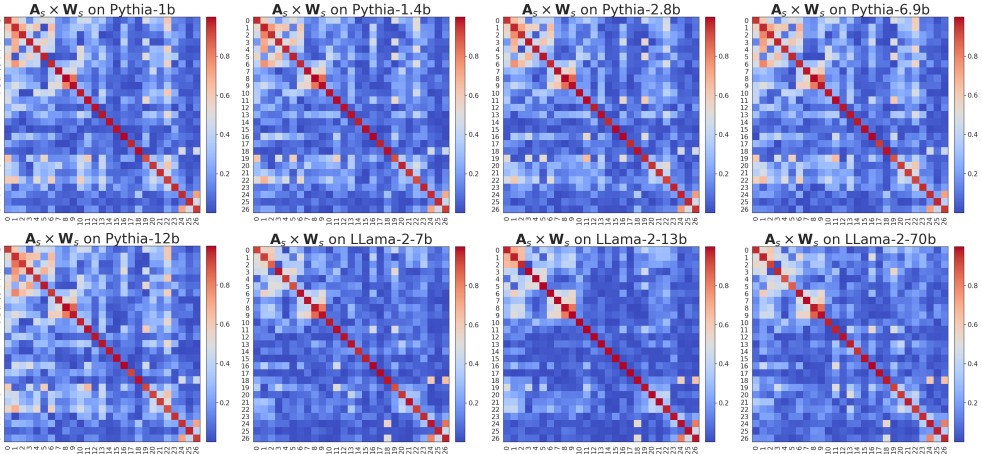

Figure 3: Results of the product $\mathbf{A}_s \times \mathbf{W}_s$ across the LLaMA-2 and Pythia model families. Here, $\mathbf{A}_s$ represents a matrix derived from the feature differences of 27 counterfactual pairs, while $\mathbf{W}_s$ is a weight matrix obtained from a linear classifier trained on these features. The product approximates the identity matrix, supporting the theoretical findings in Corollary 4.3.

For our experiments, we utilize 27 counterfactual pairs from Park et al. (2023) that differ only in a *binary* concept, which are constructed based on the Big Analogy Test dataset (Gladkova et al., 2016). More details can be found in Sec. H.2 in Appendix.

Guided by Corollary 4.2, we use these 27 counterfactual pairs to obtain $\tilde{\mathbf{A}}^k$. Specifically, for each pair differing in a binary concept, we compute the representation difference, and stacking all such vectors yields the matrix $\mathbf{A}_s \in \mathbb{R}^{27 \times \dim}$, where each row corresponds to a concept direction, and *dim* denotes the representation dimension of the pre-trained LLM used. See a binary special case of Corollary 4.2 in Appendix G.1 for further details. To obtain $\mathbf{W}$, motivated by Corollary 4.3, we train a linear classifier for each binary concept using the representations of the counterfactual pairs. The resulting weight vectors are stacked to form a matrix $\mathbf{W}_s \in \mathbb{R}^{\dim \times 27}$, where each column corresponds to the decision boundary for one concept. See a binary special case of Corollary 4.3 in Appendix G.2 for further details. Finally, we first normalize both $\mathbf{A}_s$ and $\mathbf{W}_s$ to remove the effect of scaling, which does not affect the semantics of a direction or decision boundary, and then examine the product $\mathbf{A}_s \mathbf{W}_s$. According to Corollary 4.3, the $(i, j)$-th entry of this product corresponds to the inner product between the representation difference vector for the $i$-th concept and the classifier weight vector for the $j$-th concept. When $i = j$, this inner product should be close to 1, indicating that the classifier is aligned with the concept direction. When $i \neq j$, the inner product should be less than 1, since the classifier for one concept should not be aligned with the concept direction of a different concept. Therefore, the matrix product $\mathbf{A}_s \mathbf{W}_s$ should approximate the identity matrix $\mathbf{I}$. Figure 3 displays the results of this product across the LLaMA-2 and Pythia model families. Refer to Appendix K for more results on LLaMA-3 and DeepSeek-R1. The results show that the product approximates the identity matrix, which is consistent with the theoretical finding in Corollary 4.3.

**Experiments on SAEs** We finally conduct experiment on SAEs, training four sparse variants, including top-$k$ SAE (Gao et al., 2025), batch-top-$k$ SAE (Bussmann et al., 2024), $p$-annealing SAE (Karvonen et al., 2024), and the proposed structured SAE. Following Theorem 3.1, each SAE is trained on representations from the final hidden layer across Pythia-70m, Pythia-1.4b and Pythia-2.8b (Biderman et al., 2023), with *The Pile* corpus (Gao et al., 2020). For evaluation, as mentioned in Sec. 5, we use 27 counterfactual pairs from (Park et al., 2023) again to train a linear classifier using LogisticRegression implementation from the scikit-learn library, which provides logits, i.e., unnormalized $p(c^k = 1|\mathbf{x})$, for each of the 27 pairs. These counterfactual pairs are also passed through the trained SAEs to extract features $\mathbf{z}$. We search for the best-matching feature $z_i$ for each $p(c^k|\mathbf{x})$ according to Pearson correlation between $\exp(z_i)$ and $p(c^k|\mathbf{x})$, to measure linear correlation.

Figure 4 reports the average Pearson correlations across the 27 counterfactual concepts (left, detailed results in Appendix I) and the reconstruction error measured by mean squared error (MSE, right). We

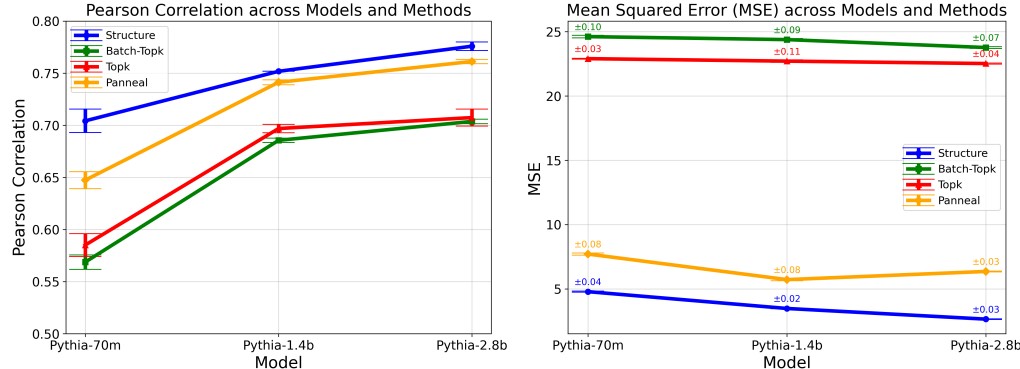

Figure 4: Comparison of SAE models. Pearson correlation scores (left) and MSE on the used counterfactual pairs (right) across different scales of Pythia.

draw two key conclusions. First, the proposed evaluation framework shows that: it differentiates SAE variants with sensitivity and provides interpretable evidence of how the learned features align with binary concepts. The trends are consistent with conventional reconstruction metrics such as MSE, reinforcing the reliability of our approach. Second, the results highlight the advantage of the proposed structured SAEs, which benefits from incorporating structured regularization. This advantage is consistently observed under both our evaluation framework and standard reconstruction metrics.

We emphasize again that obtaining counterfactual pairs is challenging. Even on this compact, high-precision benchmark, all four SAEs achieve Pearson correlations below 0.8 (1.0 indicates perfect recovery). Consequently, this benchmark is effective in differentiating the performance of the four SAEs. We hope it motivates the creation of more such high-quality counterfactual pairs.

## 7 CONCLUSION

In this work, we propose a latent variable model to capture the generative process of text, representing high-level, human-interpretable concepts as latent variables. Under mild assumptions, our analysis provides a key insight into LLMs: a *linear property*, whereby LLM representations approximate a linear transformation of the posterior over latent variables. This establishes a foundational framework for understanding next-token prediction. Furthermore, we show that this linear property offers a unified theoretical perspective on various forms of the linear representation hypothesis, and motivates a principled evaluation strategy for sparse autoencoders. These findings open avenues for deeper exploration of how LLMs learn and represent complex patterns. Building on our results, we suggest the following future direction: develop methods to linearly unmix LLM representations to directly extract the probabilities of individual high-level concepts from the latent posterior. Further details are provided in Appendix L.

## 8 ACKNOWLEDGMENT

This project was partially funded by the Responsible AI Research Centre (Yuhang Liu, Zhen Zhang, Erdun Gao, Anton van den Hengel, and Javen Qinfeng Shi). Dong Gong was partially supported by the Australian Government through the Australian Research Council Discovery Early Career Researcher Award (DECRA)(DE230101591). Mingming Gong was partially supported by the Australian Government through the Australian Research Council Discovery Projects (DP240102088). The authors also thank the anonymous reviewers for their constructive feedback.

**Ethics Statement.**   This study complies with the ethical standards of ICLR. It relies exclusively on public datasets and does not pose foreseeable negative impacts.

**Reproducibility Statement.**   We provide thorough descriptions of the methodology and experiments. The code and instructions will be openly available once the paper is published.

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

# Appendix

## Table of Contents

# A  RELATED WORK

**Linearity of Representation in LLMs**   Recent studies have established the empirical finding that concepts in LLMs are often linearly encoded, a phenomenon known as the linear representation hypothesis (Mikolov et al., 2013; Pennington et al., 2014; Arora et al., 2016; Elhage et al., 2022; Burns et al., 2022; Tigges et al., 2023; Nanda et al., 2023; Moschella et al., 2022; Park et al., 2023; Li et al., 2024; Gurnee et al., 2023; Rajendran et al., 2024). Building on this observation, recent works (Park et al., 2023; Marconato et al., 2024) attempt to unify these findings into a cohesive framework, aiming to deepen our understanding and potentially inspire new insights. However, these works do not address why and how such linear properties emerge. Some previous works (Marconato et al., 2024; Roeder et al., 2021) have demonstrated identifiability results within the inference space, such as establishing connections between features derived from different inference models. However, these results are confined to the inference space and do not connect to the true latent variables in latent variable models. A recent work (Jiang et al., 2024) seeks to explain the origins of these linear properties but employs a latent variable model that differs from ours. From a technical perspective, their explanation is rooted in the implicit bias of gradient descent. In contrast, our work provides an explanation grounded in identifiability theory. This distinction highlights our focus on connecting the observed linear properties directly to the identifiability of true latent variables in latent variable models, offering a more comprehensive and theoretically robust understanding of these phenomena. In addition, recent work (Rajendran et al., 2024) assumes continuous latent and observed variables, while we assume both latent variables and observed variables to be discrete, aligning more closely with modeling natural language.

**Causal Representation Learning**   This work is closely related to causal representation learning (Schölkopf et al., 2021), which aims to identify high-level latent causal variables from low-level observational data. Many prior studies (Brehmer et al., 2022; Von Kügelgen et al., 2021; Massidda et al., 2023; von Kügelgen et al., 2023; Ahuja et al., 2023; Seigal et al., 2022; Shen et al., 2022; Liu et al., 2022; Buchholz et al., 2023; Varici et al., 2023; Liu et al., 2024b; 2025; 2024a; 2026; Hyvarinen & Morioka, 2016; Hyvarinen et al., 2019; Khemakhem et al., 2020) have developed theoretical frameworks supporting the recovery of true latent variables up to simple transformations. However, these works primarily focus on continuous spaces and do not address the next-token prediction framework employed by LLMs. A subset of works (Gu & Dunson, 2023; Kong et al., 2024; Kivva et al., 2021) has explored causal representation learning in discrete spaces for both latent and observed variables, often imposing specific graph structures and assuming invertible mappings from latent to observed spaces. Despite their focus on discrete settings, none of these studies consider the next-token prediction framework in LLMs. In contrast, our work overcomes these limitations. We analyze approximate identifiability without relying on strict invertibility assumptions, which better aligns with the complex and often non-invertible relationships observed in real-world data. While a recent study has examined non-invertible mappings from latent to observed spaces, they rely on additional historical information to effectively restore invertibility. In our approach, we focus on achieving approximate identifiability without requiring such additional information. A very recent work (Rajendran et al., 2024) explores identifiability analysis for LLMs; however, they model both observed text data and latent variables as continuous and still assume invertibility in their analysis. In contrast, our framework explicitly considers discrete latent and observed variables, relaxed invertibility assumptions, and directly aligns with the next-token prediction paradigm, offering a more realistic and generalizable approach to causal representation learning in LLMs. Our work also differs from Zhang et al. (2025), who study identifiability through multi-task learning for the latent variables in the latent variable model of Everett (2013), where the mapping from latent to observation is assumed to be invertible. In contrast, we focus specifically on the next-token prediction framework used in LLMs, and analyze a different latent variable model that explicitly allows the mapping from latent to observed space to be non-invertible.

**Sparse Autoencoders**   Polysemanticity, a phenomenon observed in recent studies, roughly speaking, refers to cases where a single representation encodes multiple distinct, human-interpretable concepts (Arora et al., 2018). Early investigations suggested that neural networks represent features by linear superposition and motivated efforts to disentangle human-interpretable concepts from such linear mixing (Elhage et al., 2022). This can be achieved by using sparse autoencoders (Huben et al., 2024; Rajamanoharan et al., 2024a;b; Gao et al., 2025; Braun et al., 2024; Bricken et al., 2023), a technique

closely related to the well-known framework of dictionary learning (Dumitrescu & Irofti, 2018; Eggert & Korner, 2004; Elad, 2010; Elad & Bruckstein, 2002; Aharon et al., 2006; Arora et al., 2015). In contrast to these works, we propose structured SAEs to model the dependencies among latent concepts. Furthermore, motivated by our theoretical findings, we introduce a new evaluation method for SAEs, grounded in our justified theoretical results.

**Structured Sparsity**   Structured sparsity extends the classical sparse modeling paradigm by enforcing correlations or constraints among groups of features, rather than treating all entries independently. Early formulations include Group lasso (Yuan & Lin, 2006; Bach et al., 2012; Hastie et al., 2015), graph-structured sparsity (Huang et al., 2009), and hierarchical or tree-structured sparsity (Jenatton et al., 2010; Mairal et al., 2010), which allow prior knowledge about the data structure to guide feature selection. Moreover, low-rank plus structured-sparse decompositions further combine shared low-dimensional subspaces with structured sparse deviations, capturing both global and concept-specific components (Chandrasekaran et al., 2009; Sprechmann et al., 2015). Classical optimization-based approaches use nuclear-norm regularization for the low-rank component and structured norms for the sparse component (Candès et al., 2011; Chandrasekaran et al., 2011; Zhou et al., 2010). The proposed Structured SAEs bring structured-sparsity ideas into LLMs by integrating structured-sparsity constraints directly into autoencoder architectures. In doing so, they build a bridge between classical structured-sparsity methods and modern neural representations in large language models. We do not attempt to enumerate all existing work on structured sparsity, but it is expected that exploring possible structured-sparsity techniques can further benefit LLMs.

## B    LIMITATIONS

One of the main contributions of this work is establishing an identifiability result for the next-token prediction framework, a widely used approach for training LLMs. This result hinges on a key assumption:Intuitively, this requires that the posterior distribution $p(\mathbf{c} \mid y)$ changes slowly across tokens. While a given token is typically generated by only a small subset of latent concepts, and each latent concept may correspond to multiple tokens, rigorously verifying this condition in practice is challenging. On the other hand, this TV condition primarily serves as a supplementary requirement to control the influence of non-dominant points arising from the non-invertibility of the mapping from latent to observed space. It is possible that more general conditions could be formulated to replace or relax this assumption, which remains an interesting direction for future work.

Similar to existing works (Park et al., 2023; Jiang et al., 2024; Marconato et al., 2024; Rajendran et al., 2024), our analysis is limited to the last layer in LLMs and does not provide a justification for intermediate layers. Formalizing the relationship between representations in intermediate layers and identifiability analysis presents additional challenges. This complexity arises from the intricate nature of intermediate layer representations. Therefore, extending identifiability results to intermediate layers remains an open and more challenging problem.

Moreover, prior empirical studies have shown that linearity and concept representations often emerge in intermediate layers (Nanda et al., 2023; Gurnee & Tegmark, 2023). In some cases, these intermediate-layer linear representations may even diminish or transform by the time they reach the output layer (Marks & Tegmark, 2023). Consequently, while our theoretical results rigorously address identifiability at the last layer, they do not capture potential linear structures present in intermediate layers. This limitation is particularly relevant for applications such as Structured SAEs, which are sometimes trained on intermediate-layer activations rather than the final output (Gao et al., 2024). Addressing identifiability and linear representation formation across all layers remains an important direction for future work.

## C    FURTHER JUSTIFICATION FOR THE ASSUMPTIONS USED

We emphasize that, while the diversity condition is primarily a theoretical assumption ensuring identifiability, its plausibility can be assessed empirically using pre-trained LLMs on real data. Directly observing all latent variables $\mathbf{c}$ is infeasible, but the corollaries derived from our theory, including Corollary 4.2 and Corollary 4.3, provide predictions that can be indirectly validated. Prior work has shown that LLM embeddings exhibit linear representation properties when comparing

counterfactual pairs differing in a single concept (Mikolov et al., 2013; Pennington et al., 2014; Turner et al., 2023; Li et al., 2024; Wang et al., 2023; Park et al., 2023), supporting the theoretical prediction of sufficient diversity in latent directions (Corollary 4.2). In addition, our experiments show that the alignment between representation differences and classifier weights across multiple LLM families (Figures 3 and 6) closely matches the predictions of Corollary 4.3, suggesting that the diversity condition might plausibly hold in practice.

## D  PROOF OF THEOREM 3.1

*Proof.* Next-token prediction mirrors multinomial logistic regression, where the model approximates the true conditional distribution, i.e., $p(y|\mathbf{x})$, by minimizing the cross-entropy loss. In this framework, given sufficient training data, a sufficiently expressive architecture, and effective optimization, the model's predictions are expected to converge to the true conditional distribution, as follows:

$$p(y|\mathbf{x}) = \frac{\exp(\mathbf{f_x}(\mathbf{x})^T\mathbf{f}_y(y))}{\sum_{y'}\exp(\mathbf{f_x}(\mathbf{x})^T\mathbf{f}_y(y_j))}. \tag{7}$$

On the other hand, for the proposed latent variable model described in Eq. 1), the true conditional distribution $p(y|\mathbf{x})$ can be derived using Bayes' rule:

$$p(y|\mathbf{x}) = \sum_{\mathbf{c}} p(y|\mathbf{c})p(\mathbf{c}|\mathbf{x}). \tag{8}$$

By comparing Eq. 8 and Eq. 7, we arrive at the following relationship:

$$\frac{\exp(\mathbf{f_x}(\mathbf{x})^T\mathbf{f}_y(y))}{\sum_{y'}\exp(\mathbf{f_x}(\mathbf{x})^T\mathbf{f}_y(y_y))} = \sum_{\mathbf{c}} p(y|\mathbf{c})p(\mathbf{c}|\mathbf{x}). \tag{9}$$

Taking the logarithm on both sides of Eq. 9, we obtain:

$$\mathbf{f_x}(\mathbf{x})^T\mathbf{f}_y(y) - \log Z(\mathbf{x}) = \log\Big(\sum_c p(y|\mathbf{c})p(\mathbf{c}|\mathbf{x})\Big) \tag{10}$$

where $Z(\mathbf{x}) = \sum_{y_j}\exp(\mathbf{f_x}(\mathbf{x})^T\mathbf{f}_y(y_j))$ represents the normalization constant. This transformation provides a direct link between the representations $\mathbf{f}(\mathbf{x})$ learned by next-token prediction (Eq. 7) and the true posterior distribution $p(\mathbf{c}|\mathbf{x})$.

Now, let us focus on the right of Eq. 10 , we can obtain that:

$$\log\Big(\sum_c p(y|\mathbf{c})p(\mathbf{c}|\mathbf{x})\Big) = \log p(y|\mathbf{x}), \tag{11}$$

$$= \mathbb{E}_{p(\mathbf{c}|y)}\big[\log p(y|\mathbf{x})\big], \tag{12}$$

$$= \mathbb{E}_{p(\mathbf{c}|y)}\Big[\log\big(p(\mathbf{c}|y,\mathbf{x})\frac{p(y|\mathbf{x})}{p(\mathbf{c}|y,\mathbf{x})}\big)\Big], \tag{13}$$

$$= \mathbb{E}_{p(\mathbf{c}|y)}\big[\log p(y,\mathbf{c}|\mathbf{x})\big] - \mathbb{E}_{p(\mathbf{c}|y)}\big[\log p(\mathbf{c}|y,\mathbf{x})\big], \tag{14}$$

$$= \mathbb{E}_{p(\mathbf{c}|y)}\big[\log p(\mathbf{c}|\mathbf{x})\big] + \mathbb{E}_{p(\mathbf{c}|y)}\big[\log \underbrace{p(y|\mathbf{c},\mathbf{x})}_{p(y|\mathbf{c})}\big] - \mathbb{E}_{p(\mathbf{c}|y)}\big[\log p(\mathbf{c}|y,\mathbf{x})\big], \tag{15}$$

where $p(y|\mathbf{c},\mathbf{x}) = p(y|\mathbf{c})$, due to the conditional independence of $y$ and $\mathbf{x}$ given $\mathbf{c}$, as defined in the proposed latent variable model.

Together with Eq. 15 and Eq. 10, we have:

$$\mathbf{f_x}(\mathbf{x})^T\mathbf{f}_y(y) - \log Z(\mathbf{x}) = \mathbb{E}_{p(\mathbf{c}|y)}[\log p(\mathbf{c}|\mathbf{x})] + \underbrace{\mathbb{E}_{p(\mathbf{c}|y)}[\log p(y|\mathbf{c})]}_{b_y} - \mathbb{E}_{p(\mathbf{c}|y)}[\log p(\mathbf{c}|y,\mathbf{x})]. \tag{16}$$

Define $\mathbb{E}_{p(\mathbf{c}|y)}[\log p(y|\mathbf{c})] = b_y$, and define a vector $[p(\mathbf{c} = \mathbf{c}_i|\mathbf{x})]_i$ as the vector constructed by the probabilities of all possible values of $\mathbf{c}$, conditional on $\mathbf{x}$. As a result, $\mathbb{E}_{p(\mathbf{c}|y)}\log p(\mathbf{c}|\mathbf{x}) =$

$\sum_{\mathbf{c}} p(\mathbf{c}|y) \log p(\mathbf{c}|\mathbf{x}) = [p(\mathbf{c} = \mathbf{c}_i|y)]_i^T [\log p(\mathbf{c} = \mathbf{c}_i|\mathbf{x})]_i$, and similarly $\mathbb{E}_{p(\mathbf{c}|y)} \log p(\mathbf{c}|y, \mathbf{x}) = [p(\mathbf{c} = \mathbf{c}_i|y)]_i^T [\log p(\mathbf{c} = \mathbf{c}_i|y, \mathbf{x})]_i$. Then we can re-write Eq. 16 as follows:

$$\mathbf{f_x}(\mathbf{x})^T \mathbf{f}_y(y) - \log Z(\mathbf{x}) = [p(\mathbf{c} = \mathbf{c}_i|y)]_i^T [\log p(\mathbf{c} = \mathbf{c}_i|\mathbf{x})]_i \tag{17}$$

$$- [p(\mathbf{c} = \mathbf{c}_i|y)]_i^T [\log p(\mathbf{c} = \mathbf{c}_i|y, \mathbf{x})]_i + b_y \tag{18}$$

Let $y_0, \ldots, y_{\ell'}$ be the points provided by the diversity condition outlined in Section 3, for $y = 0$, we have:

$$\mathbf{f_x}(\mathbf{x})^T \mathbf{f}_y(y_0) - \log Z(\mathbf{x}) = [p(\mathbf{c} = \mathbf{c}_i|y = y_0)]_i^T [\log p(\mathbf{c} = \mathbf{c}_i|\mathbf{x})]_i$$
$$- \underbrace{[p(\mathbf{c} = \mathbf{c}_i|y = y_0)]_i^T [\log p(\mathbf{c} = \mathbf{c}_i|y = y_0, \mathbf{x})]_i}_{h_{y_0}} + b_{y_0}, \tag{19}$$

where we define $h_{y_0} = [p(\mathbf{c} = \mathbf{c}_i|y = y_0)]_i^T [\log p(\mathbf{c} = \mathbf{c}_i|y = y_0, \mathbf{x})]_i$. For $y = 1$, we similarly obtain:

$$\mathbf{f_x}(\mathbf{x})^T \mathbf{f}_y(y_1) - \log Z(\mathbf{x}) = [p(\mathbf{c} = \mathbf{c}_i|y = y_1)]_i^T [\log p(\mathbf{c} = \mathbf{c}_i|\mathbf{x})]_i$$
$$- h_{y_1} + b_{y_1}. \tag{20}$$

Subtracting Eq. 19 from Eq. 20, we get the following expression:

$$\left(\mathbf{f}_y(y_1)^T - \mathbf{f}_y(y_0)^T\right)\mathbf{f_x}(\mathbf{x}) = \left([p(\mathbf{c} = \mathbf{c}_i|y = y_1) - p(\mathbf{c} = \mathbf{c}_i|y = y_0)]_i^T\right)[\log p(\mathbf{c} = \mathbf{c}_i|\mathbf{x})]_i$$
$$- (h_{y_1} - h_{y_0}) + b_{y_1} - b_{y_0}. \tag{21}$$

According to the diversity condition, where $y$ can take $\ell' + 1$ values, we can obtain a total of $\ell'$ equations similar to Eq. 21. Collecting all of these equations, we have:

$$\underbrace{\left(\mathbf{f}_y(y_1) - \mathbf{f}_y(y_0), ..., \mathbf{f}_y(y_{\ell'}) - \mathbf{f}_y(y_0)\right)^T}_{\hat{\mathbf{L}}^T} \mathbf{f_x}(\mathbf{x}) \tag{22}$$

$$= \underbrace{\left([p(\mathbf{c} = \mathbf{c}_i|y = y_1) - p(\mathbf{c} = \mathbf{c}_i|y = y_0)]_i, ..., [p(\mathbf{c} = \mathbf{c}_i|y = y_{\ell'}) - p(\mathbf{c} = \mathbf{c}_i|y = y_0)]_i\right)^T}_{\mathbf{L}} \times$$

$$[\log p(\mathbf{c} = \mathbf{c}_i|\mathbf{x})]_i - \underbrace{[h_{y_1} - h_{y_0}, ..., h_{y_{\ell'}} - h_{y_0}]}_{\mathbf{h}_y} + \underbrace{[b_{y_1} - b_{y_0}, ..., b_{y_{\ell'}} - b_{y_0}]}_{\mathbf{b}_y}. \tag{23}$$

According to the diversity condition, the matrix $\hat{\mathbf{L}}$ of size $\ell' \times \ell'$ is invertible, as a result, we arrive:

$$\mathbf{f_x}(\mathbf{x}) = \underbrace{(\hat{\mathbf{L}}^T)^{-1}\mathbf{L}}_{\mathbf{A}}[\log p(\mathbf{c} = \mathbf{c}_i|\mathbf{x})]_i - (\hat{\mathbf{L}}^T)^{-1}\mathbf{h}_y + (\hat{\mathbf{L}}^T)^{-1}\mathbf{b}_y. \tag{24}$$

Now, we focus on the term $\mathbf{b}_y$ on the right-hand side of Eq. 24. Note that $b_{y_i} = \mathbb{E}_{p(\mathbf{c}|y_i)}[\cdot]$ is a constant with respect to $\mathbf{c}$, as the expectation integrates over all possible values of $\mathbf{c}$. As a result, the entire term $(\hat{\mathbf{L}}^T)^{-1}\mathbf{b}_y$, denoted as $\mathbf{b}$, is also a constant.

We next examine the term $\mathbf{h}_y$ in Eq. 24, considering the cases where the mapping from the latent space to the observed space is exactly invertible and where it is only approximately invertible.

**Invertible** According to definition 2.1, when the mapping $\mathbf{g}$ from latent space to observed space is invertible, meaning that for

$$1 - p(\mathbf{c} = \mathbf{c}^* | \mathbf{x}, y) = \epsilon, \tag{25}$$

we have $\epsilon = 0$. Then, for $\mathbf{h}_y$, we analyze each component, i.e., $h_{y_i} - h_{y_0}$, where

$$h_{y_i} = \mathbb{E}_{p(\mathbf{c}|y=y_i)} [\log p(\mathbf{c} | \mathbf{x}, y = y_i)], \quad h_{y_0} = \mathbb{E}_{p(\mathbf{c}|y=y_0)} [\log p(\mathbf{c} | \mathbf{x}, y = y_0)]. \tag{26}$$

When $\epsilon = 0$, the posterior distribution $p(\mathbf{c} \mid \mathbf{x}, y)$ becomes a delta distribution centered at $\mathbf{c}^*$, i.e.,

$$p(\mathbf{c} \mid \mathbf{x}, y) = \delta(\mathbf{c} - \mathbf{c}^*), \tag{27}$$

which implies that the posterior is concentrated at a single point $\mathbf{c}^*$, satisfying $\log p(\mathbf{c}^* \mid \mathbf{x}, y) = 0$. In this case, we have

$$h_{y_i} - h_{y_0} = 0. \tag{28}$$

**Approximately invertible.** Assume the generative map $\mathbf{g} : \mathbf{c} \mapsto (\mathbf{x}, y)$ is approximately invertible so that, for observed pair $(\mathbf{x}, y)$, the posterior concentrates on a single mode $\mathbf{c}^* = \arg\max_{\mathbf{c}} p(\mathbf{c} \mid \mathbf{x}, y)$ with

$$\epsilon := 1 - p(\mathbf{c}^* \mid \mathbf{x}, y) \to 0. \tag{29}$$

Let

$$q_\bullet(\mathbf{c}) := p(\mathbf{c} \mid y_\bullet), \qquad r_\bullet(\mathbf{c}) := p(\mathbf{c} \mid \mathbf{x}, y_\bullet), \tag{30}$$

and consider the remainder

$$\Delta h := h_{y_i} - h_{y_0} = \sum_c q_i(\mathbf{c}) \log r_i(\mathbf{c}) - \sum_c q_0(\mathbf{c}) \log r_0(\mathbf{c}). \tag{31}$$

Using the decomposition

$$\Delta h = \sum_{\mathbf{c}} \big(q_i(\mathbf{c}) - q_0(\mathbf{c})\big) \log r_i(\mathbf{c}) + \sum_c q_0(\mathbf{c})\big(\log r_i(\mathbf{c}) - \log r_0(\mathbf{c})\big), \tag{32}$$

we bound each term under the approximate-invertibility assumption.

For the first term, split the sum at $\mathbf{c}^*$:

$$\sum_{\mathbf{c}} \big(q_i(\mathbf{c}) - q_0(\mathbf{c})\big) \log r_i(\mathbf{c}) = \big(q_i(\mathbf{c}^*) - q_0(\mathbf{c}^*)\big) \log r_i(\mathbf{c}^*) \tag{33}$$

$$+ \sum_{\mathbf{c} \neq \mathbf{c}^*} \big(q_i(\mathbf{c}) - q_0(\mathbf{c})\big) \log r_i(\mathbf{c}). \tag{34}$$

By approximate concentration, $r_i(\mathbf{c}^*) \approx 1 - \epsilon$ and for $\mathbf{c} \neq \mathbf{c}^*$ we have $r_i(\mathbf{c}) \lesssim \epsilon$, so $\log r_i(\mathbf{c}^*) \to 0$ and $\log r_i(\mathbf{c}) \approx \log \epsilon$ for $\mathbf{c} \neq \mathbf{c}^*$. Hence

$$\left| \sum_{\mathbf{c}} \big(q_i(\mathbf{c}) - q_0(\mathbf{c})\big) \log r_i(\mathbf{c}) \right| \lesssim \left( \sum_{c \neq c^*} |q_i(\mathbf{c}) - q_0(\mathbf{c})| + |q_i(\mathbf{c}^*) - q_0(\mathbf{c}^*)| \right) |\log \epsilon|. \tag{35}$$

The prefactor is exactly $2\mathrm{TV}(p(\mathbf{c} \mid y_i), p(\mathbf{c} \mid y_0))$, so

$$\left| \sum_{\mathbf{c}} \big(q_i(\mathbf{c}) - q_0(\mathbf{c})\big) \log r_i(\mathbf{c}) \right| \lesssim 2\,\mathrm{TV}\big(p(\mathbf{c} \mid y_i), p(\mathbf{c} \mid y_0)\big) |\log \epsilon|. \tag{36}$$

For the second term,

$$\sum_{\mathbf{c}} q_0(\mathbf{c})\big(\log r_i(\mathbf{c}) - \log r_0(\mathbf{c})\big), \tag{37}$$

By the Coverage Condition, we have:

$$|\log r_i(\mathbf{c}) - \log r_0(\mathbf{c})| \leq \delta, , \tag{38}$$

thus,

$$\left| \sum_{\mathbf{c}} q_0(\mathbf{c})\big(\log r_i(\mathbf{c}) - \log r_0(\mathbf{c})\big) \right| \leq \delta. \tag{39}$$

Combining the two parts, we obtain

$$|\Delta h| \lesssim 2\,\mathrm{TV}\big(p(\mathbf{c} \mid y_i), p(\mathbf{c} \mid y_0)\big) |\log \epsilon| + \delta. \tag{40}$$

Thus, if the TV condition

$$\mathrm{TV}\big(p(\mathbf{c} \mid y_i), p(\mathbf{c} \mid y_0)\big) = o\Big(\frac{1}{|\log \epsilon|}\Big) \tag{41}$$

holds and $\delta \to 0$, then $|\Delta h| \to 0$ as $\epsilon \to 0$.

Therefore, under approximate invertibility together with the TV and mild coverage conditions above, we have $h_{y_i} - h_{y_0} \to 0$, completing the proof for the approximately-invertible case.

$$\square$$

# E    PROOF OF COROLLARY 4.2

*Proof.* We first prove that: when $\epsilon_{\mathbf{x}} \to 0$, i.e., $p(\mathbf{c} \mid \mathbf{x})$ becomes sharply peaked at $\mathbf{c}_{\mathbf{x}}^*$, we can approximate:

$$p(\mathbf{c}) \approx p(c^k \mid \mathbf{x}) \cdot p(\mathbf{c}^{-k} \mid \mathbf{x}), \tag{42}$$

where $\mathbf{c}^{-k}$ denotes all concepts except $c^k$. To this end, we analysis their KL (Kullback–Leibler) residual term:

$$\text{Residual} := D_{\text{KL}}\left(p(\mathbf{c} \mid \mathbf{x}) \,\big\|\, p(c^k \mid \mathbf{x})p(\mathbf{c}^{-k} \mid \mathbf{x})\right). \tag{43}$$

Since when $\epsilon_{\mathbf{x}} \to 0$, $p(\mathbf{c} \mid \mathbf{x})$ is sharply peaked at a particular configuration $\mathbf{c}_x^* = (c^{k*}, \mathbf{c}^{-k*})$, i.e.,:

$$p(\mathbf{c} \mid \mathbf{x}) = \begin{cases} 1 - \epsilon_{\mathbf{x}}, & \text{if } \mathbf{c} = \mathbf{c}_x^*, \\ \epsilon_{\mathbf{x}} \cdot r(\mathbf{c}), & \text{otherwise,} \end{cases} \tag{44}$$

where $\sum_{\mathbf{c} \neq \mathbf{c}_x^*} r(\mathbf{c}) = 1$.

**Main Term in KL**    The KL divergence is given by:

$$D_{\text{KL}}(p(\mathbf{c} \mid \mathbf{x})\|p(c^k \mid \mathbf{x})p(\mathbf{c}^{-k} \mid \mathbf{x})) = \sum_{\mathbf{c}} p(\mathbf{c} \mid \mathbf{x}) \log \frac{p(\mathbf{c} \mid \mathbf{x})}{p(c^k \mid \mathbf{x})p(\mathbf{c}^{-k} \mid \mathbf{x})}. \tag{45}$$

The main contribution comes from $\mathbf{c} = \mathbf{c}_x^*$:

$$(1 - \epsilon_{\mathbf{x}}) \log \frac{1 - \epsilon_{\mathbf{x}}}{p(c^{k*} \mid \mathbf{x}) \cdot p(\mathbf{c}^{-k*} \mid \mathbf{x})}. \tag{46}$$

Note that:

$$p(c^{k*} \mid \mathbf{x}) = \sum_{\mathbf{c}^{-k}} p(c^{k*}, \mathbf{c}^{-k} \mid \mathbf{x}) \geq p(c^{k*}, \mathbf{c}^{-k*} \mid \mathbf{x}) = 1 - \epsilon_{\mathbf{x}}, \tag{47}$$

and similarly:

$$p(\mathbf{c}^{-k*} \mid \mathbf{x}) \geq 1 - \epsilon_{\mathbf{x}}. \tag{48}$$

Hence:

$$p(c^{k*} \mid \mathbf{x}) \cdot p(\mathbf{c}^{-k*} \mid \mathbf{x}) \geq (1 - \epsilon_{\mathbf{x}})^2, \tag{49}$$

$$\Rightarrow \quad \frac{1 - \epsilon_{\mathbf{x}}}{p(c^{k*} \mid \mathbf{x}) \cdot p(\mathbf{c}^{-k*} \mid \mathbf{x})} \leq \frac{1}{1 - \epsilon_{\mathbf{x}}}. \tag{50}$$

Thus, the main term becomes:

$$(1 - \epsilon_{\mathbf{x}}) \log \frac{1 - \epsilon_{\mathbf{x}}}{p(c^{k*} \mid \mathbf{x}) \cdot p(\mathbf{c}^{-k*} \mid \mathbf{x})} \leq (1 - \epsilon_{\mathbf{x}}) \log \left(\frac{1}{1 - \epsilon_{\mathbf{x}}}\right) = -(1 - \epsilon_{\mathbf{x}}) \log(1 - \epsilon_{\mathbf{x}}). \tag{51}$$

**Tail Term**    For $\mathbf{c} \neq \mathbf{c}_x^*$, the contribution to the KL divergence is:

$$\epsilon_{\mathbf{x}} r(\mathbf{c}) \log \left(\frac{\epsilon_{\mathbf{x}} r(\mathbf{c})}{\epsilon_{\mathbf{x}}^2 r(c^k)r(\mathbf{c}^{-k})}\right) = \epsilon_{\mathbf{x}} r(\mathbf{c}) \log \left(\frac{1}{\epsilon_{\mathbf{x}}} \cdot \frac{r(\mathbf{c})}{r(c^k)r(\mathbf{c}^{-k})}\right). \tag{52}$$

Summing over all $\mathbf{c} \neq \mathbf{c}_x^*$, we obtain:

$$\epsilon_{\mathbf{x}} \sum_{\mathbf{c} \neq \mathbf{c}_x^*} r(\mathbf{c}) \log \left(\frac{1}{\epsilon_{\mathbf{x}}} \cdot \frac{r(\mathbf{c})}{r(c^k)r(\mathbf{c}^{-k})}\right) = \epsilon_{\mathbf{x}} \log \frac{1}{\epsilon_{\mathbf{x}}} \sum_{\mathbf{c} \neq \mathbf{c}_x^*} r(\mathbf{c}) + \epsilon_{\mathbf{x}} \sum_{\mathbf{c} \neq \mathbf{c}_x^*} r(\mathbf{c}) \log \left(\frac{r(\mathbf{c})}{r(c^k)r(\mathbf{c}^{-k})}\right). \tag{53}$$

Since the term $r(\mathbf{c})$ is bounded, we conclude:

$$\text{Tail Term} = o(\epsilon_{\mathbf{x}} \log \epsilon_{\mathbf{x}}). \tag{54}$$

which vanishes faster than the main term as $\epsilon_{\mathbf{x}} \to 0$.

**Final Bound** Combining both contributions in Main Term and Tail Term, we get:

$$D_{\mathrm{KL}}\left(p(\mathbf{c} \mid \mathbf{x}) \,\big\|\, p(c^k \mid \mathbf{x}) \cdot p(\mathbf{c}^{-k} \mid \mathbf{x})\right) \leq -(1 - \epsilon_{\mathbf{x}}) \log(1 - \epsilon_{\mathbf{x}}) + o(\epsilon_{\mathbf{x}} \log \epsilon_{\mathbf{x}}). \tag{55}$$

Here when $\epsilon_{\mathbf{x}} \to 0$, $D_{\mathrm{KL}}\left(p(\mathbf{c} \mid \mathbf{x}) \,\big\|\, p(c^k \mid \mathbf{x}) \cdot p(\mathbf{c}^{-k} \mid \mathbf{x})\right) \to 0$.

Now that we have shown that Eq. 42 holds, we can rewrite the term $[\log p(\mathbf{c}_i \mid \mathbf{x})]_i$ as:

$$[\log p(\mathbf{c}_i \mid \mathbf{x})]_i \approx \mathbf{B}^k \left[\log p(c^k \mid \mathbf{x})\right]_{c^k} + \mathbf{B}^{-i} \left[\log p(\mathbf{c}^{-k} \mid \mathbf{x})\right]_{\mathbf{c}^{-k}}. \tag{56}$$

Here, $\mathbf{B}^k$ and $\mathbf{B}^{-k}$ are binary broadcasting matrices that expand the marginal log-probability vectors $\left[\log p(c^k \mid \mathbf{x})\right]_{c^k}$ and $\left[\log p(\mathbf{c}^{-k} \mid \mathbf{x})\right]_{\mathbf{c}^{-k}}$ to the full configuration space $[\log p(\mathbf{c}_i \mid \mathbf{x})]_i$, respectively.

As a result, for pair $\mathbf{x}_0$ and $\mathbf{x}_1$ that differ only in the $i$-th concept variable $c^k$, their difference in posterior distribution is:

$$[\log p(\mathbf{c}_i \mid \mathbf{x}_1) - \log p(\mathbf{c}_i \mid \mathbf{x}_0)]_i$$
$$\approx \mathbf{B}^k \left[\log p(c^k \mid \mathbf{x}_1) - \log p(c^k \mid \mathbf{x}_0)\right]_{c^k} + \mathbf{B}^{-k} \left[\log p(\mathbf{c}^{-k} \mid \mathbf{x}_1) - \log p(\mathbf{c}^{-k} \mid \mathbf{x}_0)\right]_{\mathbf{c}^{-k}}. \tag{57}$$

We now show that:

$$p(\mathbf{c}^{-k} \mid \mathbf{x}_1) - p(\mathbf{c}^{-k} \mid \mathbf{x}_0) \to 0, \quad \text{as } \epsilon_{\mathbf{x}} \to 0 \tag{58}$$

so the term $\mathbf{B}^{-i} \left[\log p(\mathbf{c}^{-i} \mid \mathbf{x}_1) - \log p(\mathbf{c}^{-i} \mid \mathbf{x}_0)\right]_{\mathbf{c}^{-i}}$ in Eq. 57 vanishes.

Recall Eq. 44, we have

$$p(\mathbf{c} \mid \mathbf{x}) = \begin{cases} 1 - \epsilon_{\mathbf{x}}, & \text{if } \mathbf{c} = \mathbf{c}_x^* = (c^{k*}, \mathbf{c}^{-k*}), \\ \epsilon_{\mathbf{x}} \cdot r(\mathbf{c}), & \text{otherwise}, \end{cases} \tag{59}$$

Then, for $\mathbf{c}^{-i}$, we have:

$$p(\mathbf{c}^{-k} \mid \mathbf{x}) = \sum_{c^k} p(c^k, \mathbf{c}^{-k} \mid \mathbf{x}). \tag{60}$$

Combining this with Eq. 59, we have:

$$p(\mathbf{c}^{-k*} \mid \mathbf{x}) = 1 - \epsilon_{\mathbf{x}} + \epsilon_{\mathbf{x}} \sum_{c^k \neq c^{k*}} r(c^k, \mathbf{c}^{-k*}) = 1 - \epsilon_{\mathbf{x}} + o(\epsilon_{\mathbf{x}}), \tag{61}$$

$$p(\mathbf{c}^{-k} \mid \mathbf{x}) = \epsilon_{\mathbf{x}} \sum_{c^k} r(c^k, \mathbf{c}^{-k}) = o(\epsilon_{\mathbf{x}}), \quad \text{for } \mathbf{c}^{-k} \neq \mathbf{c}^{-k*}. \tag{62}$$

Both Eq. 61 and Eq. 61 only depends on $\epsilon$. As a result, if $\mathbf{x}_0$ and $\mathbf{x}_1$ have the same values on the components relevant to $\mathbf{c}^{-k}$, the difference between $p(\mathbf{c}^{-k} \mid \mathbf{x}_1)$ and $p(\mathbf{c}^{-k} \mid \mathbf{x}_0)$ is of order $o(\epsilon_{\mathbf{x}})$, as:

$$p(\mathbf{c}^{-i} \mid \mathbf{x}_1) = p(\mathbf{c}^{-i} \mid \mathbf{x}_0) + o(\epsilon_{\mathbf{x}}), \tag{63}$$

which implies:

$$\log p(\mathbf{c}^{-k} \mid \mathbf{x}_1) - \log p(\mathbf{c}^{-k} \mid \mathbf{x}_0) = \log\left(1 + \frac{o(\epsilon_{\mathbf{x}})}{p(\mathbf{c}^{-k} \mid \mathbf{x}_0)}\right) = o(\epsilon_{\mathbf{x}}). \tag{64}$$

Consequently,

$$\mathbf{B}^{-k} \left[\log p(\mathbf{c}^{-k} \mid \mathbf{x}_1) - \log p(\mathbf{c}^{-k} \mid \mathbf{x}_0)\right]_{\mathbf{c}^{-k}} = o(\epsilon_{\mathbf{x}}) \to 0 \quad \text{as } \epsilon_{\mathbf{x}} \to 0. \tag{65}$$

Together with Eq. 57 and the result from Theorem 3.1, we get:

$$\mathbf{f}_{\mathbf{x}}(\mathbf{x}_1) - \mathbf{f}_{\mathbf{x}}(\mathbf{x}_0) \approx \mathbf{A}\mathbf{B}^k \left(\left[\log p(c^k \mid \mathbf{x}_1)\right]_{c^k} - \left[\log p(c^k \mid \mathbf{x}_0)\right]_{c^k}\right). \tag{66}$$

$\square$

# F   PROOF OF COROLLARY 4.3

*Proof.* Again, when $\epsilon_{\mathbf{x}} \to 0$, i.e., when $p(\mathbf{c} \mid \mathbf{x})$ becomes sharply peaked at $\mathbf{c}_{\mathbf{x}}^*$, we can approximate:

$$p(\mathbf{c} \mid \mathbf{x}) \approx p(c^k \mid \mathbf{x}) \cdot p(\mathbf{c}^{-k} \mid \mathbf{x}), \tag{67}$$

Given the above, neglecting the constant term in the result in Theorem 3.1,

$$\mathbf{f}_{\mathbf{x}}(\mathbf{x}) \approx \mathbf{A} \left[ \log p(\mathbf{c} = \mathbf{c}_i \mid \mathbf{x}) \right]_i, \tag{68}$$

which can be rewritten as

$$\mathbf{f}_{\mathbf{x}}(\mathbf{x}) \approx \mathbf{A} \left[ \log p(\mathbf{c} = \mathbf{c}_i \mid \mathbf{x}) \right]_i \approx \mathbf{A} (\mathbf{B}^k \left[ \log p(c^k \mid \mathbf{x}) \right]_{c^k} + \mathbf{B}^{-k} \left[ \log p(\mathbf{c}^{-k} \mid \mathbf{x}) \right]_{\mathbf{c}^{-k}}). \tag{69}$$

In this case, for a data pair $(\mathbf{x}_0, \mathbf{x}_1)$ that differ only in the latent variable $c^k$, the representations $\mathbf{f}_{\mathbf{x}}(\mathbf{x})$ are passed to a linear classifier with weights $\mathbf{W}$.

The classifier produces the logits:

$$\mathbf{logits} \approx \mathbf{W} \big( \mathbf{A} (\mathbf{B}^k \left[ \log p(c^k \mid \mathbf{x}) \right]_{c^k} + \mathbf{B}^{-k} \left[ \log p(\mathbf{c}^{-k} \mid \mathbf{x}) \right]_{\mathbf{c}^{-k}}) \big). \tag{70}$$

For correct classification under cross entropy loss, the logits must match the true probabilities:

$$\mathbf{logits} = \left[ p(c^k \mid \mathbf{x}) \right]_{c^k}, \tag{71}$$

where we omit constant scaling factors for simplicity, corresponding to the normalization applied prior to the softmax operation in the cross-entropy loss.

Combining Eq. 70 and Eq. 71, the weight matrix $\mathbf{W}$ must satisfy the condition:

$$\mathbf{W}(\mathbf{A}\mathbf{B}^k) \approx \mathbf{I}, \tag{72}$$

which ensures that the classifier produces the correct logits. Here $\mathbf{I}$ denotes the identify matrix.   □

## G    EXTENSION OF COROLLARIES 4.2 AND 4.3 FOR A BINARY CONCEPT

When considering pair that differ only in a concept of interest, i.e., $c^k$, and assuming that the concept is binary, both Corollaries 4.2 and 4.3 can be further refined, yielding the following results.

### G.1    EXTENSION OF COROLLARY 4.2 FOR A BINARY CONCEPT

**Corollary G.1** (Binary Concept Direction). *Suppose that Theorem 3.1 holds, and let $c^k$ be a binary concept variable, i.e., $c^k \in \{0, 1\}$. Let $\mathbf{x}_0$ and $\mathbf{x}_1$ be a pair of inputs that differ only in the $i$-th **binary** concept $c^k$, with $c^k = 0$ for $\mathbf{x}_0$ and $c^k = 1$ for $\mathbf{x}_1$. Then, as $\epsilon_{\mathbf{x}} \to 0$, the representation difference simplifies as:*

$$\mathbf{f}_{\mathbf{x}}(\mathbf{x}_1) - \mathbf{f}_{\mathbf{x}}(\mathbf{x}_0) \approx \tilde{\mathbf{A}}^k \left( \left[ \log p(c^k \mid \mathbf{x}_1) - \log p(c^k \mid \mathbf{x}_0) \right]_{c^k} \right) \approx \log p(c^k = 0 \mid \mathbf{x}_1) \cdot \tilde{\mathbf{A}}^k \begin{bmatrix} 1 \\ -1 \end{bmatrix},$$
(73)

$$\text{or,} \approx \log p(c^k = 1 \mid \mathbf{x}_0) \cdot \tilde{\mathbf{A}}^k \begin{bmatrix} 1 \\ -1 \end{bmatrix},$$
(74)

*where $\tilde{\mathbf{A}}^k = \mathbf{A}\mathbf{B}^k$, $\mathbf{B}^k$ is a binary lifting matrix that broadcasts each entry of $[\log p(c^k \mid \mathbf{x})]_{c^k}$ to the corresponding index in $[\log p(\mathbf{c} = \mathbf{c}_i \mid \mathbf{x})]_i$. This shows that changes in a binary concept are encoded in a specific direction in the representation space defined by $\tilde{\mathbf{A}}^k$.*

*Proof.* Recall Eq. 44, the joint concept distribution conditioned on input $\mathbf{x}$ follows a peaked structure:

$$p(\mathbf{c} \mid \mathbf{x}) = \begin{cases} 1 - \epsilon_{\mathbf{x}}, & \text{if } \mathbf{c} = \mathbf{c}_{\mathbf{x}}^*, \\ \epsilon_{\mathbf{x}} \cdot r(\mathbf{c}), & \text{otherwise,} \end{cases}$$
(75)

where $\mathbf{c}_{\mathbf{x}}^* = (c^{k*}, \mathbf{c}^{-k*})$ is the dominant concept configuration for $\mathbf{x}$, and $r(\mathbf{c})$ is a normalized residual distribution over all non-dominant $\mathbf{c}$.

Let $\mathbf{x}_0$ and $\mathbf{x}_1$ differ only in the $i$-th binary concept variable $c^k$, with:

$$\mathbf{c}_{\mathbf{x}_0}^* = (0, \mathbf{c}^{-k*}), \quad \mathbf{c}_{\mathbf{x}_1}^* = (1, \mathbf{c}^{-k*}).$$
(76)

Then the marginal probabilities for $c^k$ are:

For $\mathbf{x}_1$:

$$p(c^k = 1 \mid \mathbf{x}_1) = (1 - \epsilon_{\mathbf{x}_1}) + \epsilon_{\mathbf{x}_1} \cdot \sum_{\mathbf{c}^{-k} \neq \mathbf{c}^{-k*}} r(1, \mathbf{c}^{-i}) = 1 - \epsilon_{\mathbf{x}_1} \cdot \alpha_1,$$
(77)

where $\alpha_1 := 1 - \sum_{\mathbf{c}^{-k} \neq \mathbf{c}^{-k*}} r(1, \mathbf{c}^{-k})$.

$$p(c^k = 0 \mid \mathbf{x}_1) = 1 - p(c^k = 1 \mid \mathbf{x}_1) = \epsilon_{\mathbf{x}_1} \cdot \alpha_1.$$
(78)

For $\mathbf{x}_0$:

$$p(c^k = 0 \mid \mathbf{x}_0) = (1 - \epsilon_{\mathbf{x}_0}) + \epsilon_{\mathbf{x}_0} \cdot \sum_{\mathbf{c}^{-k} \neq \mathbf{c}^{-k*}} r(0, \mathbf{c}^{-k}) = 1 - \epsilon_{\mathbf{x}_0} \cdot \alpha_0,$$
(79)

where $\alpha_0 := 1 - \sum_{\mathbf{c}^{-k} \neq \mathbf{c}^{-k*}} r(0, \mathbf{c}^{-k})$.

$$p(c^k = 1 \mid \mathbf{x}_0) = 1 - p(c^k = 1 \mid \mathbf{x}_0) = \epsilon_{\mathbf{x}_0} \cdot \alpha_0.$$
(80)

Taking logarithmic, we have:

$$\log p(c^k = 1 \mid \mathbf{x}_1) = \log(1 - \epsilon_{\mathbf{x}_1} \cdot \alpha_1) \approx 0, \quad \text{as } \epsilon_{\mathbf{x}} \to 0 \tag{81}$$

$$\log p(c^k = 0 \mid \mathbf{x}_1) = \log(\epsilon_{\mathbf{x}_1} \cdot \alpha_1). \tag{82}$$

$$\log p(c^k = 0 \mid \mathbf{x}_0) = \log(1 - \epsilon_{\mathbf{x}_0} \cdot \alpha_0) \approx 0, \quad \text{as } \epsilon_{\mathbf{x}} \to 0 \tag{83}$$

$$\log p(c^k = 1 \mid \mathbf{x}_0) = \log(\epsilon_{\mathbf{x}_0} \cdot \alpha_0) \approx \log(\epsilon_{\mathbf{x}_1} \cdot \alpha_1) = \log p(c^k = 0 \mid \mathbf{x}_1), \quad \text{as } \epsilon_{\mathbf{x}} \to 0. \tag{84}$$

Then the vector difference:

$$\left[\log p(c^k \mid \mathbf{x}_1) - \log p(c^k \mid \mathbf{x}_0)\right]_{c^k} = \begin{bmatrix} \log p(c^k = 0 \mid \mathbf{x}_1) - \log p(c^k = 0 \mid \mathbf{x}_0) \\ \log p(c^k = 1 \mid \mathbf{x}_1) - \log p(c^k = 1 \mid \mathbf{x}_0) \end{bmatrix} \tag{85}$$

$$\approx \begin{bmatrix} \log p(c^k = 0 \mid \mathbf{x}_1) \\ -\log p(c^k = 1 \mid \mathbf{x}_0) \end{bmatrix} \tag{86}$$

$$\approx \log p(c^k = 0 \mid \mathbf{x}_1) \begin{bmatrix} 1 \\ -1 \end{bmatrix} \tag{87}$$

Finally, the representation difference becomes:

$$\mathbf{f_x}(\mathbf{x}_1) - \mathbf{f_x}(\mathbf{x}_0) \approx \tilde{\mathbf{A}}^k \left(\left[\log p(c^k \mid \mathbf{x}_1) - \log p(c^k \mid \mathbf{x}_0)\right]_{c^k}\right) \approx \log p(c^k = 0 \mid \mathbf{x}_1) \cdot \tilde{\mathbf{A}}^k \begin{bmatrix} 1 \\ -1 \end{bmatrix}. \tag{88}$$

Here, note that: $\log p(c^k = 0 \mid \mathbf{x}_1) \approx \log p(c^k = 1 \mid \mathbf{x}_0)$ as shown in Eq. 84. $\qquad\square$

### G.2 EXTENSION OF COROLLARY 4.3 FOR A BINARY CONCEPT

**Corollary G.2** (Binary Concept Classification). *Suppose that Theorem 3.1 holds, i.e., $\mathbf{f_x}(\mathbf{x}) \approx \mathbf{A}\left[\log p(\mathbf{c} = \mathbf{c}_i \mid \mathbf{x})\right]_i + \mathbf{b}$. Let $\mathbf{x}_0$ and $\mathbf{x}_1$ be pair data that differ only in the $i$-th binary concept variable $c^k$, with labels $c^k$, where $c^k = 0$ for $\mathbf{x}_0$ and $c^k = 1$ for $\mathbf{x}_1$. Then when $\epsilon_{\mathbf{x}} \to 0$, the corresponding representations $(\mathbf{f}(\mathbf{x}_0), \mathbf{f}(\mathbf{x}_1))$ are linearly separable with a weight **vector** $\mathbf{w}$ satisfying $\mathbf{w}^\top \tilde{\mathbf{A}}^k \begin{bmatrix} -1 \\ 1 \end{bmatrix} \approx 1$. The corresponding logit is the (unnormalized) $p(c^k = 1 \mid \mathbf{x})$.*

*Proof.* When $\epsilon_{\mathbf{x}} \to 0$, the posterior $p(\mathbf{c} \mid \mathbf{x})$ becomes sharply peaked at a unique mode $\mathbf{c}_{\mathbf{x}}^*$. This implies a near-independence of $c^k$ and $\mathbf{c}^{-i}$ given $\mathbf{x}$, allowing us to write:

$$p(\mathbf{c} \mid \mathbf{x}) \approx p(c^k \mid \mathbf{x}) \cdot p(\mathbf{c}^{-k} \mid \mathbf{x}). \tag{89}$$

Taking logs and substituting into Theorem 3.1 (neglecting the bias term $\mathbf{b}$), we obtain:

$$\mathbf{f_x}(\mathbf{x}) \approx \mathbf{A}\left[\log p(\mathbf{c}_i \mid \mathbf{x})\right]_i \approx \mathbf{A}\left(\mathbf{B}^k[\log p(c^k \mid \mathbf{x})]_{c^k} + \mathbf{B}^{-i}[\log p(\mathbf{c}^{-k} \mid \mathbf{x})]_{\mathbf{c}^{-k}}\right), \tag{90}$$

where $\mathbf{B}^k$ and $\mathbf{B}^{-k}$ denote the lifting operators that map the marginal log-probabilities of $c^k$ and $\mathbf{c}^{-k}$ into the joint log-probability vector space.

Define $\tilde{\mathbf{A}}^k := \mathbf{A}\mathbf{B}^k$. Then:

$$\mathbf{f_x}(\mathbf{x}) \approx \tilde{\mathbf{A}}^k[\log p(c^k \mid \mathbf{x})]_{c^k} + (\text{terms involving only } \mathbf{c}^{-k}). \tag{91}$$

Consider a linear classifier with weight vector $\mathbf{w}$ applied to $\mathbf{f_x}(\mathbf{x})$:

$$\text{logit} = \mathbf{w} \cdot \mathbf{f_x}(\mathbf{x}) \approx \mathbf{w}\tilde{\mathbf{A}}^k \begin{bmatrix} p(c^k = 0 \mid \mathbf{x}) \\ 1 - p(c^k = 0 \mid \mathbf{x}) \end{bmatrix} + \text{const.} \tag{92}$$

Let $\mathbf{w}^\top \tilde{\mathbf{A}}^k = [s_0, s_1]$. Then:

$$\text{logit} \approx s_0 \log p(c^k = 0 \mid \mathbf{x}) + s_1 \log p(c^k = 1 \mid \mathbf{x}) + \text{const} = (s_1 - s_0) \log p(c^k = 1 \mid \mathbf{x}) + \text{const.} \tag{93}$$

For the classifier to correctly separate $\mathbf{x}_0$ and $\mathbf{x}_1$ under cross-entropy loss, we require:

$$\text{logit} = \log p(c^k = 1 \mid \mathbf{x}), \tag{94}$$

where we omit the normalization constant prior to the softmax, since it does not affect the analysis. This is equivalent to:

$$\mathbf{w}^\top \tilde{\mathbf{A}}^k \begin{bmatrix} -1 \\ 1 \end{bmatrix} \approx 1, \tag{95}$$

This completes the proof. $\qquad\square$

# H EXPERIMENTAL DETAILS SUPPORTING THEORETICAL RESULTS

## H.1 SIMULATION DETAILS

For the left side of Figure 2, which investigates the relationship between the degree of invertibility in the mapping from $\mathbf{c}$ to $\mathbf{x}$ and the approximation of the identifiability result in Theorem 3.1, we aim to exclude other uncertain factors that might affect the result. To achieve this, we keep the number of latent variables constant (i.e., 3) and ensure that the graph structure follows a chain structure. Based on this structure, we model the conditional probabilities of each variable, given its parents, using Bernoulli distributions. The parameters of these distributions are uniformly sampled from the interval [0.2, 0.8], which are then used to generate the latent variables. Subsequently, we apply a one-hot encoding to these samples to obtain one-hot formal representations. These one-hot samples are then randomly permuted. For 3 latent variables, there are $2^3$ possible permutations, each corresponding to 3 observed binary variables, resulting in a total of $2^3 \times 3$ different observed binary variables. We then randomly sample from these observed variables, varying the sample size. For example, as shown on the left in Figure 2, we can select different variables as observed variables. Clearly, as the number of observed variables increases, the mutual information between observed and latent variables also increases. As a result, the degree of invertibility from latent to observed variables increases.

For the right side of Figure 2, we explore the robustness of our identifiability result in Theorem 3.1 with respect to both the graph structure and the size of the latent variables. To this end, we randomly generate DAG structures in the latent space using Erdős-Rényi (ER) graphs (ERDdS & R&wi, 1959), where ER$k$ denotes graphs with $d$ nodes and $kd$ expected edges. For each ER$k$ configuration, we also vary the size of the latent variables from 4 to 8, allowing us to examine how the size of latent variables influences the identifiability results. In terms of the observed variables, we adapt the experimental setup from the left side of Figure 2 to determine the appropriate observed variable size for different latent variable sizes. This ensures that the degree of invertibility from the latent space to the observed space remains sufficiently high, a crucial factor for the accuracy of our identifiability analysis.

Throughout the simulation, we use the following: In each experiment, we randomly mask one observed variable $x_i$, and use the remaining observed variables to predict it. Specifically, the remaining variables and the corresponding mask matrix are used as inputs to an embedding layer. This embedding layer transforms the input into a high-dimensional feature representation. The generated embeddings are then passed through a Multi-Layer Perceptron (MLP)-based architecture to extract meaningful features, e.g., $\mathbf{f_x}(\mathbf{x})$. The MLP model consists of three layers, each with 256 hidden units. After ach layer, we apply Batch Normalization unit to stabilize training and a ReLU nonlinear activation function to introduce nonlinearity. The final output of the MLP is used to predict the masked variable through a linear classification layer. This allows us to assess how well the model can predict missing or masked values based on the remaining observed variables. We employ the Adam optimizer with a learning rate of $1e - 4$. To ensure robustness and account for potential variability in the results, we conduct each experimental setting with five different runs, each initialized with a different random seed. This procedure helps mitigate the effects of random initialization and provides a more reliable evaluation of the model's performance.

For evaluation, we use the LogisticRegression classifier from the scikit-learn library, which operates on the features extracted from the output of the MLP-based architecture described above.

## H.2 EXPERIMENTAL DETAILS ON LLMS

Unlike in simulation studies, where we have access to the complete set of latent variables, in real-world scenarios, their true values remain inherently unknown. This limitation arises from the nature of latent variables, they are unobserved and must be inferred indirectly from the data. As a consequence, we cannot directly validate the linear identifiability results established in Theorem 3.1, since such validation would require explicit knowledge of these latent variables.

However, we can instead verify Corollary 4.3, which is a direct consequence of Theorem 3.1. By doing so, we provide indirect empirical evidence supporting the theoretical identifiability results. To achieve this, we need collect counterfactual pairs of data instances that differ in controlled and specific ways. These counterfactual pairs are essential for testing the implications of our theory in the context of real-world data.

Generating such counterfactual pairs, however, presents significant challenges. First, the inherent complexity and nuances of natural language make it difficult to create pairs that differ in precisely the intended contexts while leaving other aspects unchanged. Second, as highlighted in prior works (Park et al., 2023; Jiang et al., 2024), constructing such counterfactual sentences is a highly non-trivial task, even for human annotators, due to the intricacies of semantics and the need for precise control over contextual variations.

We utilize the 27 counterfactual pairs introduced in (Park et al., 2023), which provide a structured and well-curated set of counterfactual pairs. These concepts encompass a wide range of semantic and morphological transformations, as detailed in Table 1. By leveraging this established dataset, we ensure consistency with previous research while facilitating a robust and meaningful evaluation of Corollary 4.3.

We first use these 27 counterfactual pairs to construct a $\mathbf{A}_s$ with size $27 \times dim$ by using the differences in the representations of these 27 counterfactual pairs, where $dim$ corresponds to the feature dimension of the used LLM, such as 4096 for the Llama-27B model. To construct the corresponding matrix $\mathbf{W}_s$, we train a linear classifier using these 27 counterfactual pairs. Specifically, we use the representations of the counterfactual pairs as input and the corresponding values of the latent variables as output. As a result, the corresponding linear weights for the 27 counterfactual pairs are be used to create $\mathbf{W}_s$. In our experiments, we employ the LogisticRegression classifier from the scikit-learn library.

Note that, since the 27 counterfactual pairs involve only binary concepts, each row of $\mathbf{A}_s$ corresponds to the direction of a concept associated with a counterfactual pair, as stated in Corollary G.1. Similarly, as discussed in Corollary G.2, each column of $\mathbf{W}_s$ denotes the classifier direction for a specific pair and aligns with the corresponding row of $\mathbf{A}_s$ for that concept. As a result, after normalizing $\mathbf{A}_s$ and $\mathbf{W}_s$ to remove the arbitrary logit scaling (which is irrelevant before the softmax in cross-entropy loss), we expect to observe that $\mathbf{A}_s \mathbf{W}_s \approx \mathbf{I}$. Therefore, we apply normalization to both $\mathbf{A}_s$ and $\mathbf{W}_s$ prior to computing their product.

## I  EXPERIMENT ON SPARSE AUTOENCODERS

### I.1  IMPLEMENTATION OF THE PROPOSED STRUCTURED SAE

The proposed structured SAE employs two regularization terms: a structured regularization to model the dependence among latent concepts, and a sparsity regularization based on the assumption that latent concepts may be sparsely activated. We implement it as follows:

$$\mathbf{S} = \text{ReLU}(\mathbf{w}_s(\mathbf{f_x}(\mathbf{x}) - \mathbf{b}_d) + \mathbf{b}_s), \tag{96}$$

$$\mathbf{R} = \text{ReLU}(\mathbf{w}_l(\mathbf{f_x}(\mathbf{x}) - \mathbf{b}_d) + \mathbf{b}_l), \tag{97}$$

$$\mathbf{z} = \mathbf{S} + \mathbf{R}, \tag{98}$$

$$\bar{\mathbf{f}}_\mathbf{x}(\mathbf{x}) = \mathbf{w}_d \mathbf{z} + \mathbf{b}_d. \tag{99}$$

Here:

- $\mathbf{S}$ denotes the sparse representations from the sparse encoder (with parameters $\mathbf{w}_s, \mathbf{b}_s$);
- $\mathbf{R}$ denotes the structured representation from the structured encoder (with parameters $\mathbf{w}_l, \mathbf{b}_l$);
- $\mathbf{z}$ is the combined representations used for reconstruction;
- $\bar{\mathbf{f}}_\mathbf{x}(\mathbf{x})$ denote the reconstruction of $\mathbf{f_x}(\mathbf{x})$.

The loss function for training is:

$$\mathcal{L} = \mathbb{E}_{\mathbf{x} \sim \mathcal{D}_{\text{train}}} \left[ \|\mathbf{f_x}(\mathbf{x}) - \bar{\mathbf{f}}_\mathbf{x}(\mathbf{x})\|_2^2 + \lambda_t \left( \|\mathbf{S}\|_{p_t}^{p_t} + \gamma \|\mathbf{R}\|_{\text{nuc}} \right) \right], \tag{100}$$

where:

Table 1: Concept names, one example of the counterfactual pairs, and the number of used pairs, taken from (Park et al., 2023).

| # | Concept | Example | Word Pair Counts |
|---|---------|---------|------------------|
| 1 | verb $\Rightarrow$ 3pSg | (accept, accepts) | 50 |
| 2 | verb $\Rightarrow$ Ving | (add, adding) | 50 |
| 3 | verb $\Rightarrow$ Ved | (accept, accepted) | 50 |
| 4 | Ving $\Rightarrow$ 3pSg | (adding, adds) | 50 |
| 5 | Ving $\Rightarrow$ Ved | (adding, added) | 50 |
| 6 | 3pSg $\Rightarrow$ Ved | (adds, added) | 50 |
| 7 | verb $\Rightarrow$ V + able | (accept, acceptable) | 50 |
| 8 | verb $\Rightarrow$ V + er | (begin, beginner) | 50 |
| 9 | verb $\Rightarrow$ V + tion | (compile, compilation) | 50 |
| 10 | verb $\Rightarrow$ V + ment | (agree, agreement) | 50 |
| 11 | adj $\Rightarrow$ un + adj | (able, unable) | 50 |
| 12 | adj $\Rightarrow$ adj + ly | (according, accordingly) | 50 |
| 13 | small $\Rightarrow$ big | (brief, long) | 25 |
| 14 | thing $\Rightarrow$ color | (ant, black) | 50 |
| 15 | thing $\Rightarrow$ part | (bus, seats) | 50 |
| 16 | country $\Rightarrow$ capital | (Austria, Vienna) | 158 |
| 17 | pronoun $\Rightarrow$ possessive | (he, his) | 4 |
| 18 | male $\Rightarrow$ female | (actor, actress) | 52 |
| 19 | lower $\Rightarrow$ upper | (always, Always) | 73 |
| 20 | noun $\Rightarrow$ plural | (album, albums) | 100 |
| 21 | adj $\Rightarrow$ comparative | (bad, worse) | 87 |
| 22 | adj $\Rightarrow$ superlative | (bad, worst) | 87 |
| 23 | frequent $\Rightarrow$ infrequent | (bad, terrible) | 86 |
| 24 | English $\Rightarrow$ French | (April, avril) | 116 |
| 25 | French $\Rightarrow$ German | (ami, Freund) | 128 |
| 26 | French $\Rightarrow$ Spanish | (annee, año) | 180 |
| 27 | German $\Rightarrow$ Spanish | (Arbeit, trabajo) | 228 |

- $\lambda_t$ is the dynamically adjusted sparsity coefficient at step $t$, following $p$-annealing SAE (Karvonen et al., 2024);

- $\gamma$ is a hyperparameter that balances the sparsity penalty and the low-rank regularization;

- $\|\mathbf{S}\|_{p_t}^{p_t} = \sum_i |s_i|^{p_t}$ is the adaptive $L_{p_t}$ norm promoting sparsity, following $p$-annealing SAE (Karvonen et al., 2024);

- $\|\mathbf{R}\|_{\text{nuc}}$ is the nuclear norm, used to encourage low-rank structure.

We estimate the nuclear norm using the top-$k_{svd}$ singular values:

$$\|\mathbf{R}\|_{\text{nuc}} \approx \sum_{i=1}^{k_{svd}} \sigma_i, \tag{101}$$

where $\{\sigma_i\}_{i=1}^{k_{svd}}$ are the largest $k_{svd}$ singular values, obtained via low-rank SVD (e.g., PyTorch's `svd_lowrank`). The value of $k_{svd}$ is a tunable parameter (e.g., $k_{svd} = 64$) that trades off approximation accuracy and computational cost.

## I.2 DETAILS OF EVALUATION METRIC

**Obtaining $p(c^k = 1|\mathbf{x})$ from Supervised Linear Classification.** For evaluation, we first use 27 counterfactual pairs from (Park et al., 2023), also see Table 1, to train a linear classification using LogisticRegression classifier from the scikit-learn library, to obtain logits, i.e., unnormalized $p(c^k = 1|\mathbf{x})$, for each of the 27 pairs. As a result, for each concept in these 27 concepts, we can obtain the corresponding logit. Figure 5 shows classification accuracy of LogisticRegression classifier.

Stacking these 27 logits yields the logit vector

$$\mathbf{u} = (u_1, u_2, \ldots, u_{27}). \tag{102}$$

**Extracting $z_i$ from trained SAEs.** We use the representations $\mathbf{f_x}(\mathbf{x})$ of the same 27 counterfactual pairs from (Park et al., 2023) as input to a trained SAE, extracting the corresponding latent features $\mathbf{z}$. Let $\tilde{\mathbf{z}}$ denote the element-wise exponentiation of $\mathbf{z}$, i.e., $\tilde{\mathbf{z}} = \exp(\mathbf{z})$. This yields a feature matrix of size $27 \times D$, where $D$ is the dimensionality of $\mathbf{z}$:

$$\tilde{\mathbf{z}} = (\tilde{\mathbf{z}}_1, \tilde{\mathbf{z}}_2, \ldots, \tilde{\mathbf{z}}_{27})^T. \tag{103}$$

**Correlation Matrix and Assignment.** For the logit vector $\mathbf{u}$, and the feature matrix $\tilde{\mathbf{z}}$, we compute the Pearson correlation

$$\mathbf{R}_d = \text{corr}(\mathbf{u}, \tilde{\mathbf{z}}_{:,d}), \qquad d = 1, \ldots, D \tag{104}$$

where $\tilde{\mathbf{z}}_{:,d}$ denotes the $d$-th column of $\tilde{\mathbf{z}}$.

Note that the estimated features $\mathbf{z}$ from the SAE are subject to permutation indeterminacy. To address this, we apply the Hungarian algorithm to solve the assignment problem on $\mathbf{R}_d$. This yields the optimal assignment for each concept, allowing us to compute the assigned Pearson correlation. We report the mean Pearson correlation across the 27 concepts.

## I.3 EXPERIMENTS AND RESULTS

We train four SAE variants—top-$k$ SAE (Gao et al., 2025), batch-top-$k$ SAE (Bussmann et al., 2024), $p$-annealing SAE (Karvonen et al., 2024), and our proposed structured SAE. Each SAE is trained three times on activations from the final hidden layer of the pretrained Pythia 70m, Pythia 1.4b, and Pythia 2.8b (Biderman et al., 2023) (download from `https://huggingface.co/EleutherAI`), using training data from the first 200 million tokens of the Pile corpus (Gao et al., 2020) (download from `https://huggingface.co/datasets/EleutherAI/the_pile_deduplicated`).

**Experimental setup.** We set the feature dimension of all SAE variants to be the same ($D = 32{,}768$) and are trained for $20\,000$ optimization steps with a batch size of $10\,000$. We employ the Adam optimizer with an initial learning rate of $1 \times 10^{-4}$ and linearly warm up the learning rate during the first 200 steps. For the top-$k$ and batch-top-$k$ SAEs, we set $k = 32$. For $p$-annealing SAEs, we apply a sparsity warm-up of 400 steps and an initial sparsity penalty coefficient $\lambda_s = 0.1$. The $p$-annealing-LoRa SAE uses the same $p$-annealing settings and additionally applies a low-rank scaling factor $\gamma = 0.1$.

**Compute resources.** All experiments are conducted on a server equipped with four NVIDIA A100 GPUs (40 GB each).

**Pearson Correlation Coefficient.** We report Pearson Correlation Coefficient (PCC) at the $20\,000$-step checkpoint. As summarized in Tables 3-5, the two $p$-annealing variants markedly outperform the fixed-$k$ baselines. Across the 27 concepts, the proposed structured SAE achieves better PCC across different scales of Pythia family, confirming that sparsity and the low-rank adaptation yield features that align more cleanly with human-interpretable concepts.

Table 2: Ablation results for the low-rank term scaling factor $\gamma$. PCC is reported as mean $\pm$ standard deviation over multiple runs.

| $\gamma$ | PCC (mean $\pm$ std) |
|---|---|
| $10^{-3}$ | $0.691 \pm 0.023$ |
| $10^{-2}$ | $0.695 \pm 0.019$ |
| $10^{-1}$ | $0.704 \pm 0.013$ |
| $1$ | $0.685 \pm 0.021$ |

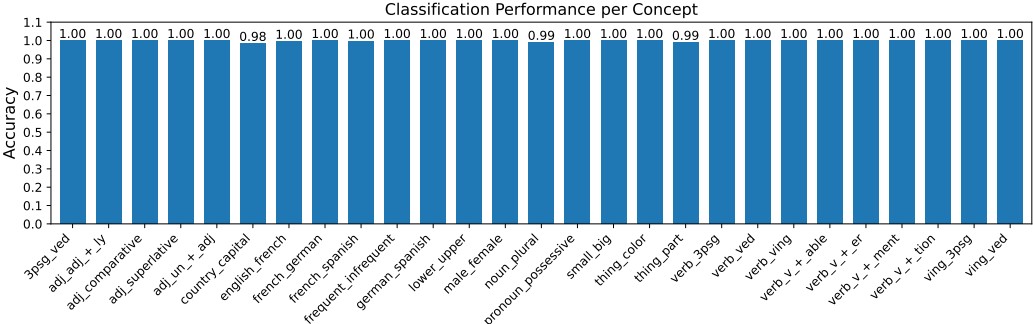

Figure 5: Classification accuracy of logistic probes across various concepts. Each bar represents the performance for a given concept.

**Ablation on the low-rank coefficient.** We investigate the effect of the low-rank term scaling factor $\gamma \in \{10^{-3}, 10^{-2}, 10^{-1}, 1\}$ while keeping all other hyperparameters fixed. As shown in Table 2, the PCC peaks at $\gamma = 10^{-1}$, indicating that a moderate weighting of the low-rank term is beneficial. Smaller or larger values of $\gamma$ lead to slightly worse performance, suggesting that both under- and over-emphasis on the low-rank term can hurt the model's ability to capture the underlying structure.

Table 3: PCC results at training step 20,000 on Pythia-70m. Bold values denote the best.

| Concepts | PCC across different SAEs (↑) | | | |
| --- | --- | --- | --- | --- |
| | top-$k$ | batch-top-$k$ | $p$-annealing | Our structured SAE |
| 3psg_ved | $0.533 \pm 0.069$ | $0.466 \pm 0.012$ | $0.660 \pm 0.011$ | $\mathbf{0.684 \pm 0.008}$ |
| adj_adj_+_ly | $0.828 \pm 0.054$ | $\mathbf{0.918 \pm 0.005}$ | $0.848 \pm 0.004$ | $0.884 \pm 0.005$ |
| adj_comparative | $0.640 \pm 0.108$ | $0.646 \pm 0.087$ | $0.536 \pm 0.077$ | $\mathbf{0.695 \pm 0.070}$ |
| adj_superlative | $0.743 \pm 0.017$ | $0.757 \pm 0.028$ | $0.731 \pm 0.021$ | $\mathbf{0.792 \pm 0.033}$ |
| adj_un_+_adj | $0.362 \pm 0.041$ | $0.353 \pm 0.030$ | $0.510 \pm 0.031$ | $\mathbf{0.771 \pm 0.033}$ |
| country_capital | $0.237 \pm 0.008$ | $0.247 \pm 0.021$ | $0.356 \pm 0.005$ | $\mathbf{0.494 \pm 0.003}$ |
| english_french | $0.501 \pm 0.076$ | $0.448 \pm 0.006$ | $0.629 \pm 0.008$ | $\mathbf{0.639 \pm 0.005}$ |
| french_german | $0.615 \pm 0.003$ | $0.617 \pm 0.023$ | $0.608 \pm 0.020$ | $\mathbf{0.685 \pm 0.050}$ |
| french_spanish | $0.627 \pm 0.069$ | $0.520 \pm 0.054$ | $\mathbf{0.643 \pm 0.033}$ | $0.613 \pm 0.036$ |
| frequent_infrequent | $0.279 \pm 0.014$ | $0.292 \pm 0.017$ | $0.490 \pm 0.011$ | $\mathbf{0.509 \pm 0.018}$ |
| german_spanish | $0.583 \pm 0.077$ | $\mathbf{0.730 \pm 0.010}$ | $0.688 \pm 0.021$ | $0.713 \pm 0.010$ |
| lower_upper | $0.474 \pm 0.080$ | $0.442 \pm 0.016$ | $\mathbf{0.631 \pm 0.010}$ | $0.614 \pm 0.008$ |
| male_female | $0.378 \pm 0.040$ | $0.326 \pm 0.008$ | $0.585 \pm 0.030$ | $\mathbf{0.625 \pm 0.006}$ |
| noun_plural | $0.616 \pm 0.100$ | $0.501 \pm 0.038$ | $\mathbf{0.662 \pm 0.024}$ | $0.640 \pm 0.011$ |
| pronoun_possessive | $0.897 \pm 0.035$ | $0.885 \pm 0.037$ | $0.967 \pm 0.022$ | $\mathbf{0.974 \pm 0.020}$ |
| small_big | $0.403 \pm 0.005$ | $0.412 \pm 0.040$ | $0.603 \pm 0.023$ | $\mathbf{0.668 \pm 0.010}$ |
| thing_color | $\mathbf{0.950 \pm 0.008}$ | $0.899 \pm 0.042$ | $0.918 \pm 0.020$ | $0.915 \pm 0.016$ |
| thing_part | $0.421 \pm 0.013$ | $0.345 \pm 0.016$ | $\mathbf{0.494 \pm 0.031}$ | $0.506 \pm 0.008$ |
| verb_3psg | $0.658 \pm 0.035$ | $0.588 \pm 0.061$ | $0.673 \pm 0.033$ | $\mathbf{0.752 \pm 0.027}$ |
| verb_v_+_able | $0.723 \pm 0.068$ | $0.706 \pm 0.033$ | $0.787 \pm 0.031$ | $\mathbf{0.825 \pm 0.018}$ |
| verb_v_+_er | $0.581 \pm 0.039$ | $0.548 \pm 0.021$ | $\mathbf{0.772 \pm 0.025}$ | $0.748 \pm 0.011$ |
| verb_v_+_ment | $0.750 \pm 0.013$ | $\mathbf{0.778 \pm 0.017}$ | $0.622 \pm 0.019$ | $0.696 \pm 0.017$ |
| verb_v_+_tion | $0.764 \pm 0.054$ | $0.681 \pm 0.079$ | $0.722 \pm 0.048$ | $\mathbf{0.794 \pm 0.035}$ |
| verb_ved | $0.522 \pm 0.064$ | $0.583 \pm 0.087$ | $0.537 \pm 0.061$ | $\mathbf{0.668 \pm 0.052}$ |
| verb_ving | $0.689 \pm 0.066$ | $0.717 \pm 0.035$ | $0.638 \pm 0.026$ | $\mathbf{0.737 \pm 0.034}$ |
| ving_3psg | $0.467 \pm 0.057$ | $0.403 \pm 0.029$ | $\mathbf{0.704 \pm 0.029}$ | $0.704 \pm 0.010$ |
| ving_ved | $0.557 \pm 0.043$ | $0.538 \pm 0.067$ | $0.691 \pm 0.053$ | $\mathbf{0.688 \pm 0.022}$ |

Table 4: PCC results at training step 20,000 on Pythia-1.4b. Bold values denote the best.

| Concepts | PCC across different SAEs (↑) | | | |
|---|---|---|---|---|
| | top-$k$ | batch-top-$k$ | $p$-annealing | Our structured SAE |
| 3psg_ved | $0.577 \pm 0.015$ | $0.536 \pm 0.013$ | $0.629 \pm 0.012$ | $\mathbf{0.727 \pm 0.009}$ |
| adj_adj_+_ly | $0.870 \pm 0.013$ | $0.841 \pm 0.009$ | $0.897 \pm 0.020$ | $\mathbf{0.940 \pm 0.011}$ |
| adj_comparative | $\mathbf{0.760 \pm 0.018}$ | $0.738 \pm 0.013$ | $0.640 \pm 0.011$ | $0.660 \pm 0.010$ |
| adj_superlative | $0.849 \pm 0.022$ | $\mathbf{0.871 \pm 0.017}$ | $0.767 \pm 0.013$ | $0.790 \pm 0.009$ |
| adj_un_+_adj | $0.563 \pm 0.076$ | $0.556 \pm 0.107$ | $0.654 \pm 0.021$ | $\mathbf{0.749 \pm 0.012}$ |
| average | $0.697 \pm 0.004$ | $0.686 \pm 0.002$ | $0.742 \pm 0.019$ | $\mathbf{0.752 \pm 0.010}$ |
| country_capital | $0.802 \pm 0.032$ | $0.795 \pm 0.006$ | $0.788 \pm 0.007$ | $\mathbf{0.823 \pm 0.012}$ |
| english_french | $\mathbf{0.836 \pm 0.014}$ | $0.810 \pm 0.066$ | $0.813 \pm 0.008$ | $0.814 \pm 0.011$ |
| french_german | $0.701 \pm 0.030$ | $0.676 \pm 0.081$ | $0.790 \pm 0.016$ | $\mathbf{0.799 \pm 0.012}$ |
| french_spanish | $0.737 \pm 0.006$ | $0.706 \pm 0.041$ | $\mathbf{0.823 \pm 0.010}$ | $0.762 \pm 0.013$ |
| frequent_infrequent | $0.522 \pm 0.029$ | $0.569 \pm 0.003$ | $0.588 \pm 0.012$ | $\mathbf{0.646 \pm 0.011}$ |
| german_spanish | $0.687 \pm 0.023$ | $0.673 \pm 0.038$ | $\mathbf{0.796 \pm 0.009}$ | $0.783 \pm 0.014$ |
| lower_upper | $\mathbf{0.765 \pm 0.007}$ | $0.746 \pm 0.007$ | $0.716 \pm 0.010$ | $0.726 \pm 0.012$ |
| male_female | $0.584 \pm 0.047$ | $\mathbf{0.617 \pm 0.011}$ | $0.593 \pm 0.006$ | $0.546 \pm 0.007$ |
| noun_plural | $0.688 \pm 0.006$ | $0.703 \pm 0.029$ | $\mathbf{0.851 \pm 0.014}$ | $0.814 \pm 0.010$ |
| pronoun_possessive | $0.937 \pm 0.018$ | $0.869 \pm 0.031$ | $0.980 \pm 0.012$ | $\mathbf{0.989 \pm 0.008}$ |
| small_big | $0.287 \pm 0.007$ | $0.300 \pm 0.008$ | $\mathbf{0.536 \pm 0.013}$ | $0.533 \pm 0.010$ |
| thing_color | $0.913 \pm 0.019$ | $\mathbf{0.937 \pm 0.009}$ | $0.928 \pm 0.012$ | $0.904 \pm 0.011$ |
| thing_part | $0.397 \pm 0.005$ | $0.402 \pm 0.013$ | $0.440 \pm 0.010$ | $\mathbf{0.531 \pm 0.014}$ |
| verb_3psg | $0.675 \pm 0.090$ | $0.619 \pm 0.011$ | $0.709 \pm 0.008$ | $\mathbf{0.716 \pm 0.012}$ |
| verb_v_+_able | $0.785 \pm 0.048$ | $0.743 \pm 0.026$ | $\mathbf{0.899 \pm 0.015}$ | $0.798 \pm 0.013$ |
| verb_v_+_er | $0.707 \pm 0.012$ | $0.745 \pm 0.019$ | $0.752 \pm 0.011$ | $\mathbf{0.781 \pm 0.012}$ |
| verb_v_+_ment | $0.769 \pm 0.014$ | $0.603 \pm 0.033$ | $\mathbf{0.831 \pm 0.010}$ | $0.741 \pm 0.009$ |
| verb_v_+_tion | $0.857 \pm 0.025$ | $0.879 \pm 0.013$ | $\mathbf{0.841 \pm 0.012}$ | $0.818 \pm 0.010$ |
| verb_ved | $0.599 \pm 0.027$ | $0.674 \pm 0.032$ | $0.682 \pm 0.009$ | $\mathbf{0.710 \pm 0.008}$ |
| verb_ving | $0.730 \pm 0.004$ | $0.721 \pm 0.053$ | $\mathbf{0.800 \pm 0.013}$ | $0.794 \pm 0.011$ |
| ving_3psg | $0.612 \pm 0.064$ | $0.625 \pm 0.036$ | $0.642 \pm 0.010$ | $\mathbf{0.745 \pm 0.014}$ |
| ving_ved | $0.605 \pm 0.019$ | $0.558 \pm 0.008$ | $0.645 \pm 0.011$ | $\mathbf{0.660 \pm 0.012}$ |

## J  FURTHER DISCUSSION: OBSERVATIONS ON LLMS AND WORLD MODELS

### J.1  LLMS MIMIC THE HUMAN WORLD MODEL, NOT THE WORLD ITSELF

As we explore the implications of our linear identifiability result, it is important to situate it within the broader context of human cognition—specifically, how humans develop and interact with an internal world model. In this subsection, we introduce the concept that LLMs Mimic the Human World Model, Not the World Itself, emphasizing the distinction between the vast physical world and the compressed abstraction humans use for reasoning and decision-making. Our analysis shows that LLMs replicate human-like abstractions through latent variable models. We further argue that LLMs aim to emulate this internal, compressed world model—rather than the physical world itself—by learning from human-generated text. This distinction provides critical insight into the success of LLMs in tasks aligned with human conceptualization and deepens our understanding of the relationship between language models and human cognition.

Humans gradually develop their understanding of the environment through learning from others and interacting with the world. This internal representation of our external environment is known as a world model. A key observation is that this model is not a direct reflection of the physical world but rather a highly compressed abstraction of it. For instance, numbers, such as 1, 2, and 3, are abstract tools created by the human mind for reasoning and problem-solving. They do not exist as tangible entities in the natural world. Similarly, many aspects of reality that escape human senses or even the most sophisticated scientific instruments are absent from our mental representations.

Interestingly, our texts reflect our mental activities and emotions, providing a window into this compressed world model. A recent study reveals a striking disparity between the limited information throughput of human behavior (approximately 10 bits/s) and the vast sensory input available (around $10^9$ bits/s) (Zheng & Meister, 2024). This suggests that the human world model is an efficient,

Table 5: PCC results at training step 20,000 on Pythia-2.8b. Bold values denote the best.

| Concepts | PCC across different SAEs (↑) | | | |
| | top-$k$ | batch-top-$k$ | $p$-annealing | Our structured SAE |
| --- | --- | --- | --- | --- |
| 3psg_ved | $0.589 \pm 0.025$ | $0.513 \pm 0.024$ | $0.645 \pm 0.012$ | $\mathbf{0.734 \pm 0.027}$ |
| adj_adj_+_ly | $0.869 \pm 0.036$ | $0.797 \pm 0.050$ | $0.919 \pm 0.020$ | $\mathbf{0.941 \pm 0.013}$ |
| adj_comparative | $0.847 \pm 0.020$ | $\mathbf{0.859 \pm 0.010}$ | $0.817 \pm 0.015$ | $0.757 \pm 0.028$ |
| adj_superlative | $0.885 \pm 0.011$ | $0.897 \pm 0.010$ | $\mathbf{0.901 \pm 0.017}$ | $0.787 \pm 0.009$ |
| adj_un_+_adj | $0.631 \pm 0.086$ | $0.690 \pm 0.033$ | $0.590 \pm 0.012$ | $\mathbf{0.693 \pm 0.021}$ |
| average | $0.707 \pm 0.008$ | $0.704 \pm 0.002$ | $0.762 \pm 0.011$ | $\mathbf{0.771 \pm 0.022}$ |
| country_capital | $0.810 \pm 0.040$ | $0.791 \pm 0.034$ | $\mathbf{0.892 \pm 0.014}$ | $0.832 \pm 0.020$ |
| english_french | $\mathbf{0.854 \pm 0.010}$ | $0.834 \pm 0.010$ | $0.842 \pm 0.015$ | $0.847 \pm 0.027$ |
| french_german | $0.659 \pm 0.060$ | $0.710 \pm 0.035$ | $0.743 \pm 0.016$ | $\mathbf{0.799 \pm 0.022}$ |
| french_spanish | $0.741 \pm 0.027$ | $0.711 \pm 0.012$ | $\mathbf{0.828 \pm 0.011}$ | $0.794 \pm 0.019$ |
| frequent_infrequent | $0.523 \pm 0.066$ | $0.576 \pm 0.036$ | $0.572 \pm 0.018$ | $\mathbf{0.670 \pm 0.028}$ |
| german_spanish | $0.643 \pm 0.040$ | $0.646 \pm 0.063$ | $\mathbf{0.816 \pm 0.021}$ | $0.805 \pm 0.030$ |
| lower_upper | $\mathbf{0.772 \pm 0.027}$ | $0.758 \pm 0.010$ | $0.704 \pm 0.012$ | $0.739 \pm 0.025$ |
| male_female | $0.640 \pm 0.040$ | $0.570 \pm 0.014$ | $\mathbf{0.721 \pm 0.018}$ | $0.640 \pm 0.020$ |
| noun_plural | $0.761 \pm 0.024$ | $0.737 \pm 0.014$ | $\mathbf{0.879 \pm 0.023}$ | $0.830 \pm 0.027$ |
| pronoun_possessive | $0.918 \pm 0.049$ | $0.914 \pm 0.011$ | $0.980 \pm 0.015$ | $\mathbf{0.995 \pm 0.021}$ |
| small_big | $0.347 \pm 0.007$ | $0.400 \pm 0.043$ | $0.481 \pm 0.014$ | $\mathbf{0.602 \pm 0.032}$ |
| thing_color | $0.918 \pm 0.019$ | $\mathbf{0.932 \pm 0.012}$ | $0.923 \pm 0.007$ | $0.901 \pm 0.015$ |
| thing_part | $0.373 \pm 0.033$ | $0.319 \pm 0.021$ | $0.457 \pm 0.010$ | $\mathbf{0.506 \pm 0.025}$ |
| verb_3psg | $0.523 \pm 0.071$ | $0.578 \pm 0.025$ | $0.643 \pm 0.012$ | $\mathbf{0.739 \pm 0.021}$ |
| verb_v_+_able | $0.751 \pm 0.022$ | $0.736 \pm 0.023$ | $\mathbf{0.832 \pm 0.018}$ | $0.828 \pm 0.026$ |
| verb_v_+_er | $0.715 \pm 0.052$ | $0.790 \pm 0.011$ | $\mathbf{0.832 \pm 0.025}$ | $0.806 \pm 0.022$ |
| verb_v_+_ment | $0.784 \pm 0.036$ | $0.762 \pm 0.032$ | $\mathbf{0.860 \pm 0.015}$ | $0.854 \pm 0.028$ |
| verb_v_+_tion | $0.805 \pm 0.053$ | $0.845 \pm 0.085$ | $\mathbf{0.864 \pm 0.023}$ | $0.861 \pm 0.030$ |
| verb_ved | $0.720 \pm 0.041$ | $0.728 \pm 0.039$ | $0.664 \pm 0.012$ | $\mathbf{0.752 \pm 0.018}$ |
| verb_ving | $0.732 \pm 0.060$ | $0.755 \pm 0.035$ | $\mathbf{0.845 \pm 0.020}$ | $0.819 \pm 0.027$ |
| ving_3psg | $\mathbf{0.728 \pm 0.082}$ | $0.559 \pm 0.117$ | $0.690 \pm 0.015$ | $0.675 \pm 0.022$ |
| ving_ved | $0.562 \pm 0.022$ | $0.589 \pm 0.009$ | $\mathbf{0.624 \pm 0.017}$ | $0.620 \pm 0.019$ |

compressed abstraction, enabling us to reason, predict, and make decisions effectively. Despite this compression, humans have flourished as the dominant species on Earth, demonstrating the power of such a streamlined model.

LLMs aim to mimic not the vast and unbounded world but this human-compressed world model, which is significantly smaller and more manageable. By learning from human text, which encodes this abstraction, LLMs effectively replicate the patterns, reasoning, and abstractions that have proven successful for humans. This explains the impressive performance of LLMs in tasks that align with human understanding. Furthermore, the overlap between the latent space of LLMs (representing human concepts) and the observed space (human text) provides a powerful mechanism for aligning human-like abstractions with model predictions. For instance, modifying words in the observed space often corresponds to predictable changes in latent concepts.

### J.2 LLMs Versus Pure Vision Models: A Fundamental Difference

While LLMs model human language, which has already undergone significant compression through human cognition, vision models face a fundamentally different challenge. In the previous subsection, we discussed how LLMs mimic the human world model, leveraging compressed abstractions derived from human-generated text. In contrast, vision models operate on raw, high-dimensional data from visual inputs. Moreover, the training data for vision models represents only a tiny fraction of the universe, constrained by the limitations of capturing devices and datasets. This vastness and lack of compression make the task of building generalizable representations in vision models inherently more complex than in NLP-based LLMs.

This difference in the nature of their training data and observed space might explain the behavioral differences between LLMs and pure vision models. While LLMs benefit from the inherent abstraction and compression of human language, vision models must contend with raw, unprocessed inputs that

## K    MORE RESULTS ON LLAMA-3 AND DEEPSEEK-R1

We conduct additional experiments on recent LLMs, including Llama-3 and DeepSeek-R1, to further evaluate our findings. The experimental setup strictly adheres to the settings described in Section H.2. This enables a comprehensive investigation of our findings, ensuring that the results are thoroughly evaluated and validated across diverse LLM architectures and experimental conditions. Overall, we can see, the product $\mathbf{A}_s \times \mathbf{W}_s$ approximates the identity matrix, supporting the theoretical findings outlined in Corollary G.2.

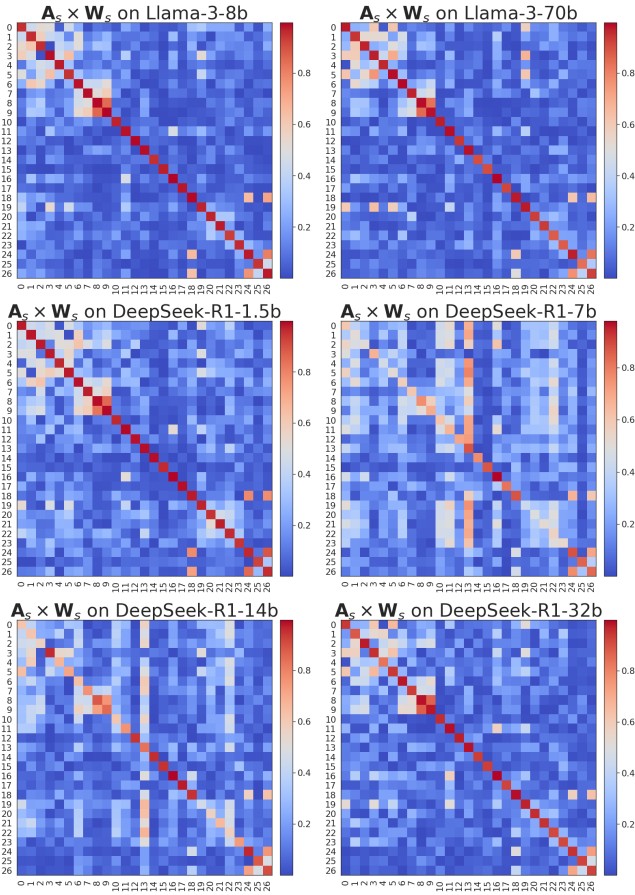

Figure 6: Results of the product $\mathbf{A}_s \times \mathbf{W}_s$ across the LLaMA-3 and DeepSeek-R1 model families. Here, $\mathbf{A}_s$ represents a matrix derived from the feature differences of 27 counterfactual pairs, while $\mathbf{W}_s$ is a weight matrix obtained from a linear classifier trained on these features.

## L    FUTURE DIRECTIONS

**Rethinking Invertibility Assumptions in Causal Representation Learning**    Our identifiability analysis is closely related to the concept of identifiability in causal representation learning. Notably, to the best of our knowledge, this is the first work to explore approximate identifiability in the context of *non-invertible* mappings from latent space to observed space—a departure from the commonly upheld *invertibility* assumption in the causal representation learning community (Brehmer et al., 2022; Von Kügelgen et al., 2021; Massidda et al., 2023; von Kügelgen et al., 2023; Ahuja et al., 2023; Seigal et al., 2022; Shen et al., 2022; Liu et al., 2022). We hope that our work will inspire future research aimed at overcoming the limitations imposed by invertibility assumptions in causal representation learning.

**Embedding Causal Reasoning in LLMs Through Linear Unmixing**    Our linear identifiability result lays a foundation for uncovering latent causal relationships among concepts, especially when these variables exhibit causal dependencies within the proposed latent variable model. By showing that the representations in LLMs are linear mixtures of latent causal variables, our analysis shows that linear unmixing of these representations may allow for the identification of underlying latent causal structures. This approach not only opens up the possibility of understanding causal dynamics within LLMs but also suggests that causal reasoning, particularly in latent spaces, could be achievable through the exploration of these linear unmixing techniques. We believe this work marks a pivotal step toward embedding robust causal reasoning capabilities into LLMs.

## M    ACKNOWLEDGMENT OF LLMS USAGE

We disclose that large language models (LLMs) were used only to correct typos, improve grammar, and refine phrasing. All scientific contributions, including problem formulation, methodology design, theoretical proofs, experiments, and analysis of results, are exclusively the work of the authors.

## N    CROSS-VALIDATION AND NULL BASELINE ANALYSIS

**Motivation.** In previous experimental results in Figure 3, we demonstrate the results of the product $\mathbf{A}_s \times \mathbf{W}_s$ across the LLaMA-2 and Pythia model families. However, the concept directions $\mathbf{A}_s$ and probing weights $\mathbf{W}_s$ were computed using the same set of word pairs, which could artificially inflate diagonal alignment due to repeated supervision. To rigorously test the theoretical finding in Corollary 4.3, we introduce two complementary safeguards: (i) **50%/50% split cross-validation**, and (ii) **null baseline with randomly permuted labels**.

Regarding **50%/50% Cross-Validation**, we split each word pair into two halves. The first half is used to compute the concept direction (mean difference vector), and the second half is used exclusively for training the linear probing classifier, to ensures that the evaluation is performed on held-out data.

For **Null Baseline (Permutation Labels)**, We use randomly permuted labels of the word pairs to train the linear probing classifier. The resulting distribution of diagonal alignments serves as a null model, reflecting the alignment expected by chance. The results are shown in Figures 7 and 8. These findings confirm that the high diagonal alignment observed experimental results in Figure 3 is not a trivial consequence of reusing the same supervision, but rather reflects a meaningful association between the concept directions and the learned probe weights, aligning with the theoretical finding in Corollary 4.3. We also report Frobenius norm, which measures how far the obtained operator deviates from the ideal projection in its overall action, with smaller values indicating closer alignment to the identity on the concept subspace.

We further evaluate the setup under multiple random seeds that control the 50%/50% cross-validation split. This allows us to verify that the observed performance is not an artifact of a particular partition and that the conclusions hold consistently across different random splits. The results are shown in Figure 9–13, which demonstrate that the finding is robust across different random seeds.

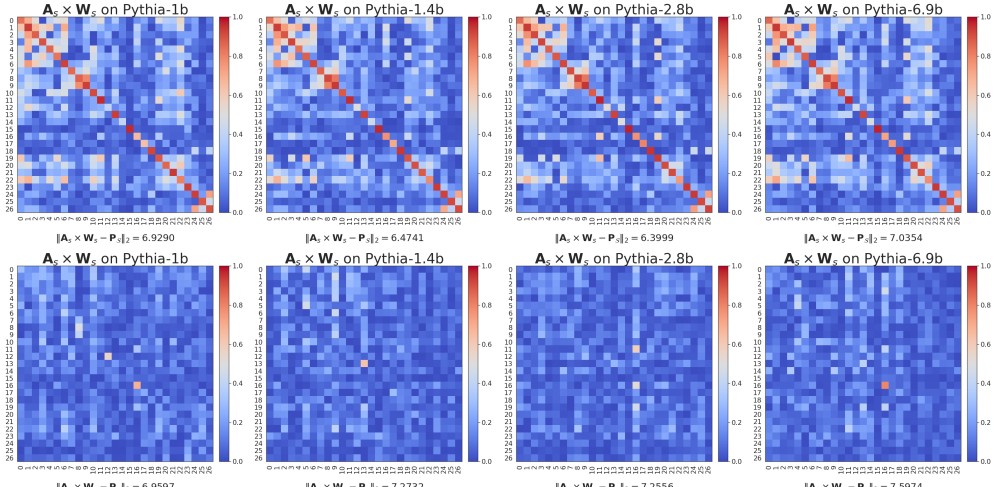

Figure 7: Results of the product $\mathbf{A}_s \times \mathbf{W}_s$ under Cross-Validation (top) and Null Baseline (bottom). The cross-validation ensures that the observed effect generalizes to held-out embeddings, while the null baseline provides a quantitative reference to rule out spurious alignment. Both results are consistent with the theoretical finding in Corollary 4.3.

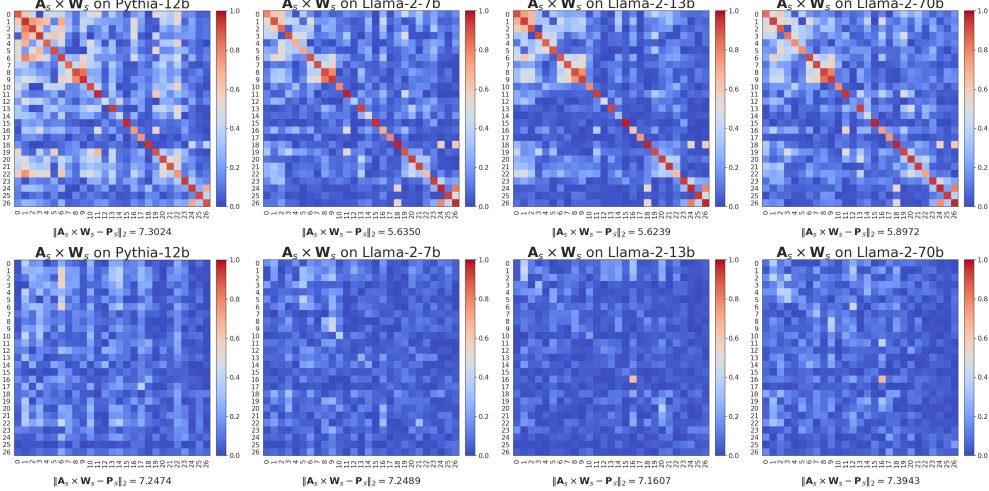

Figure 8: Results of the product $\mathbf{A}_s \times \mathbf{W}_s$ under Cross-Validation (top) and Null Baseline (bottom). The cross-validation ensures that the observed effect generalizes to held-out embeddings, while the null baseline provides a quantitative reference to rule out spurious alignment. Both results are consistent with the theoretical finding in Corollary 4.3.

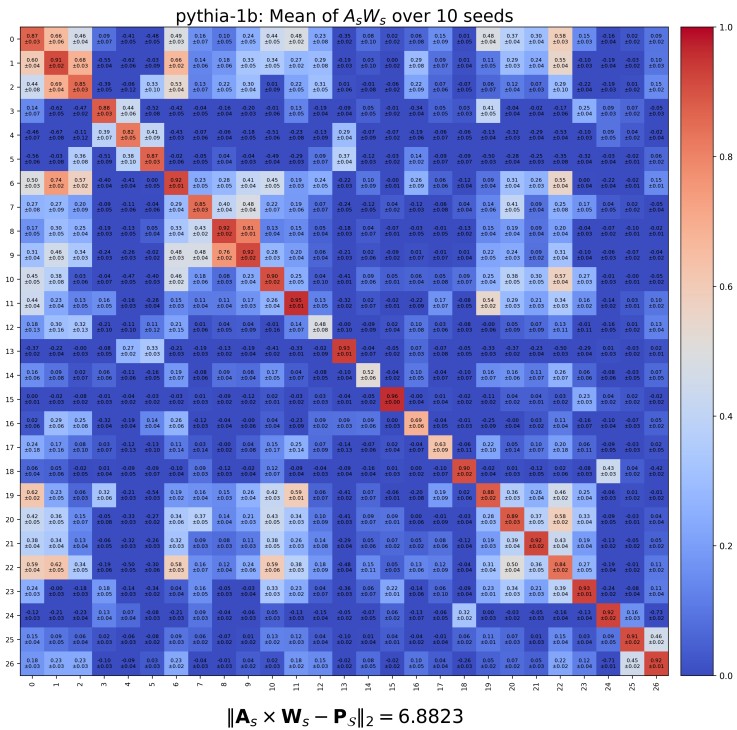

Figure 9: Results of the product $\mathbf{A}_s \times \mathbf{W}_s$ on Pythia-1b across different seeds.

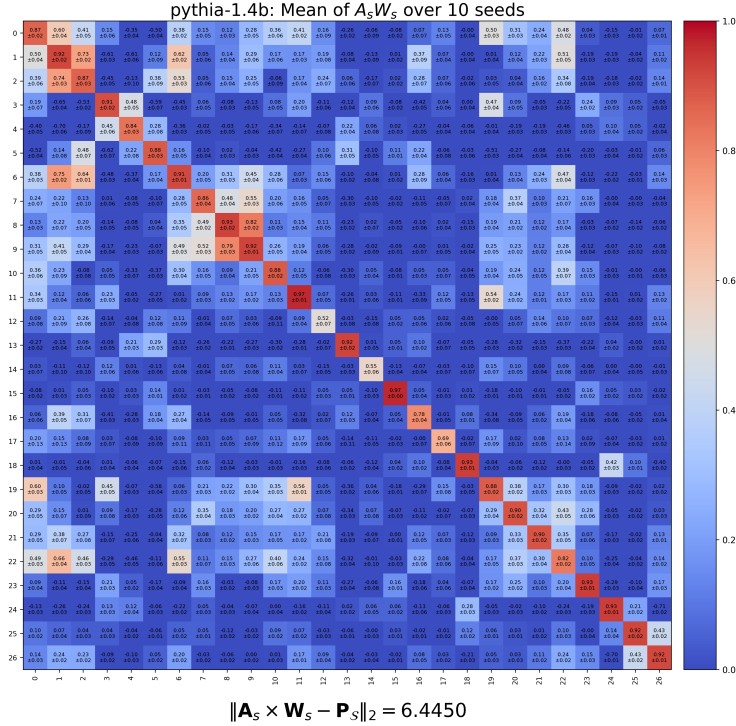

Figure 10: Results of the product $\mathbf{A}_s \times \mathbf{W}_s$ on Pythia-1.4b across different seeds.

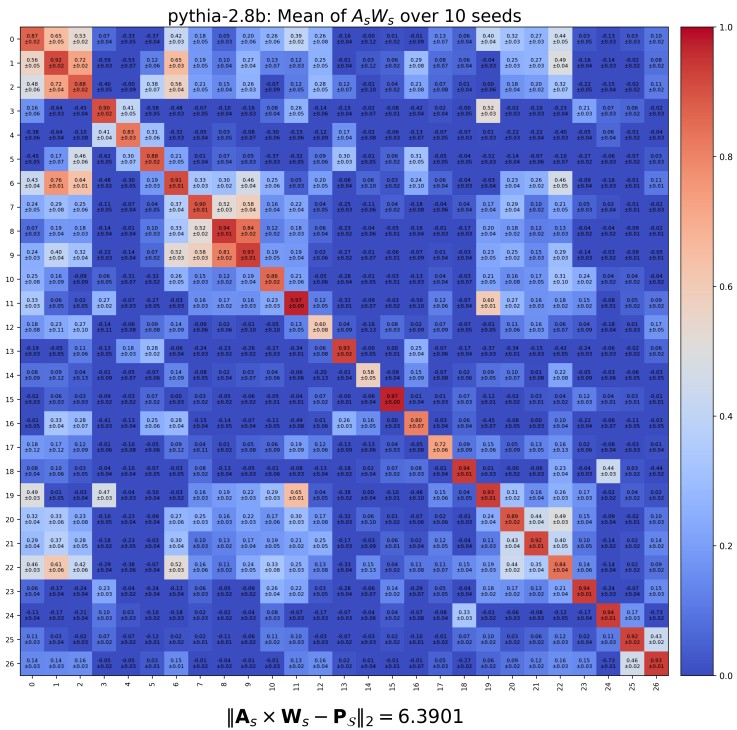

Figure 11: Results of the product $\mathbf{A}_s \times \mathbf{W}_s$ on Pythia-2.8b across different seeds.

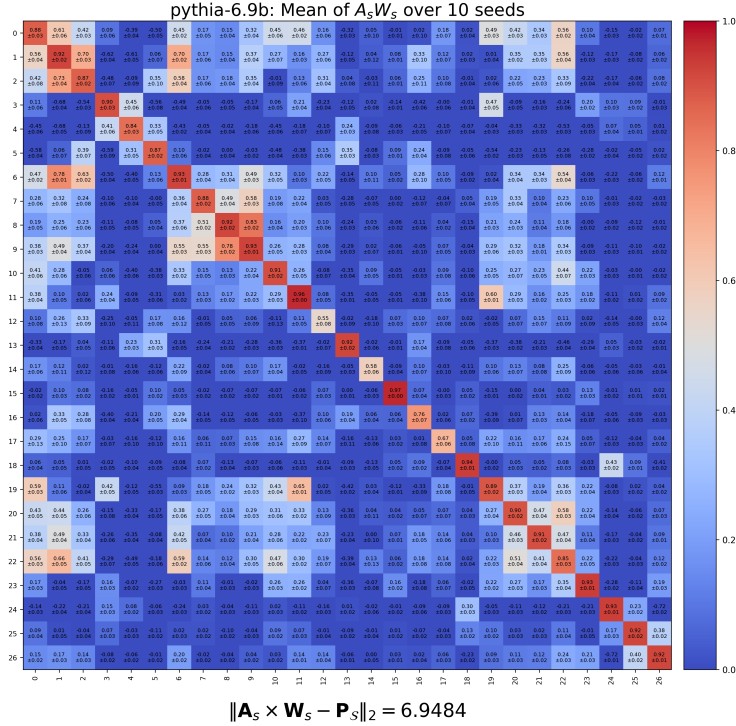

Figure 12: Results of the product $\mathbf{A}_s \times \mathbf{W}_s$ on Pythia-6.9b across different seeds.

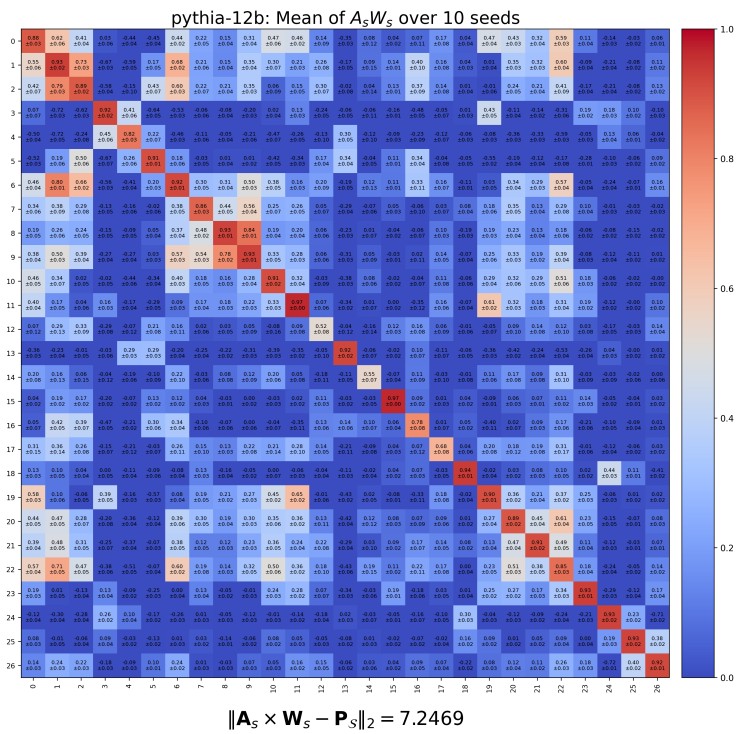

Figure 13: Results of the product $\mathbf{A}_s \times \mathbf{W}_s$ on Pythia-12b across different seeds.

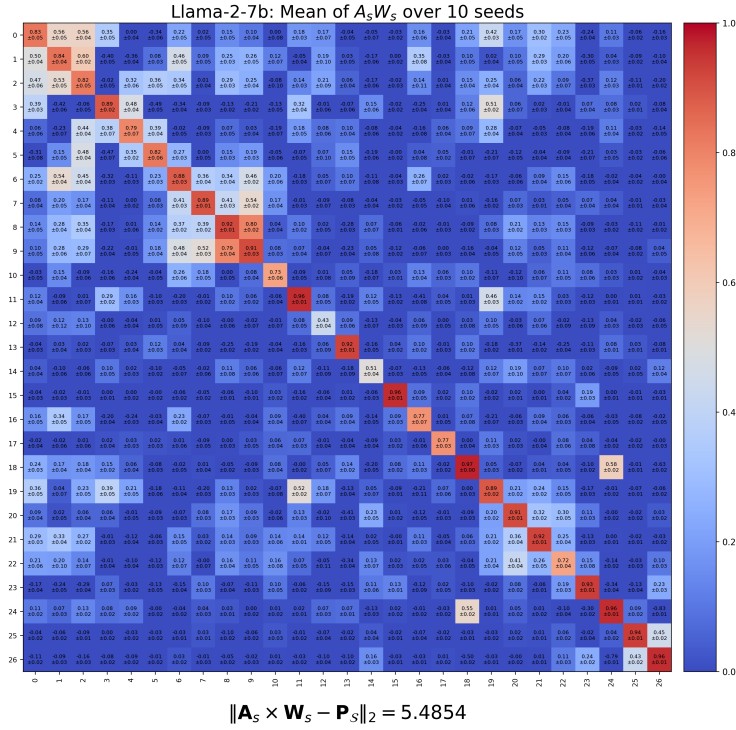

Figure 14: Results of the product $\mathbf{A}_s \times \mathbf{W}_s$ on Llama-2-7b across different seeds.

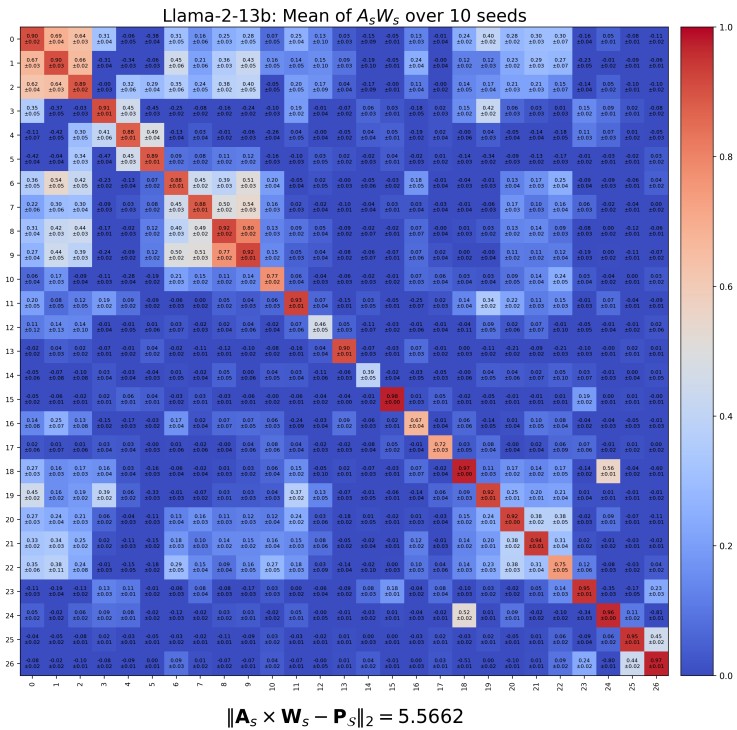

Figure 15: Results of the product $\mathbf{A}_s \times \mathbf{W}_s$ on Llama-2-13b across different seeds.

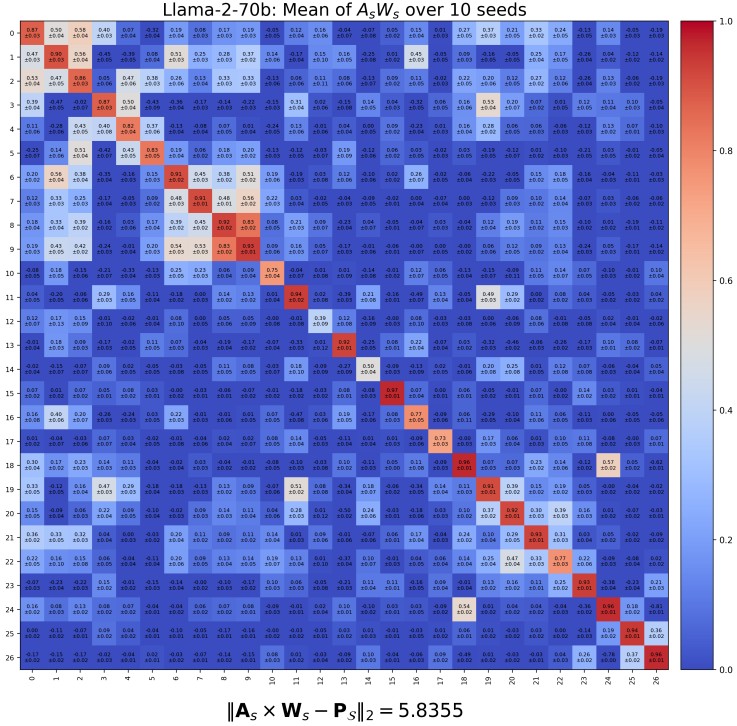

Figure 16: Results of the product $\mathbf{A}_s \times \mathbf{W}_s$ on Llama-2-70b across different seeds.

## O    ASSESSING ROBUSTNESS THE LINEAR RESULT IN THEOREM 3.1

To assess the practical impact of the theoretical correction term $\mathbf{h}_y$ in Theorem 3.1, which appears as the residual perturbation to the ideal linear result in Theorem 3.1. We design an experiment that uses linear-probe performance as an empirical proxy for the magnitude of this term. Intuitively, if $\mathbf{h}_y$ is negligible for a given pretrained model, the representation should closely follow the ideal linear form and a linear probe trained on counterfactual concept pairs will achieve high test accuracy; conversely, a large $\mathbf{h}_y$ should degrade probe performance. Concretely, we use the 27 concept counterfactual pairs to train a logistic linear probe to discriminate the pair. To quantify variability, we repeat the train/test (70%/30%) split three times with different random seeds and report the mean and standard deviation of test accuracy. The majority of concepts show high mean accuracy and small standard deviation, indicating that the $\mathbf{h}_y$ correction is empirically small and that the linear approximation in Theorem 3.1 holds well in practice. The results are shown in Figures 17-21. We note that some concepts, such as the "pronoun - possessive" category, have very limited sample sizes, which lead to poor and unstable test performance; consequently, the results for these concepts should be interpreted with caution. Overall, the repeated-run analysis demonstrates that our main empirical findings are robust and not an artifact of a particular random split, and it provides quantitative evidence that the $\mathbf{h}_y$ term has limited influence for most concepts under study.

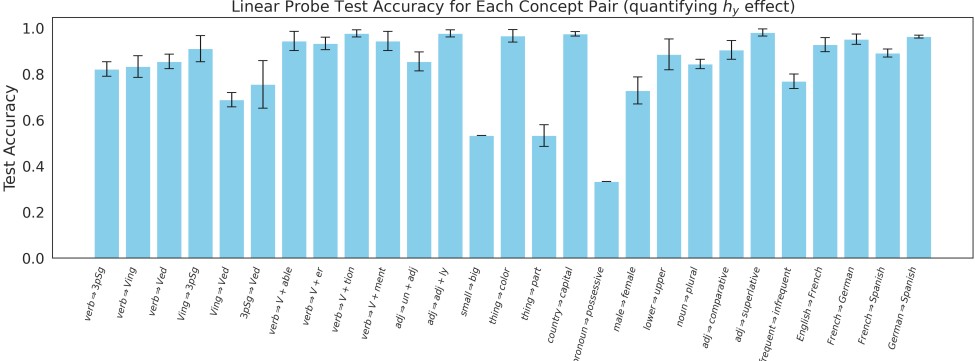

Figure 17: Testing performance of linear probing on Pythia-1b model.

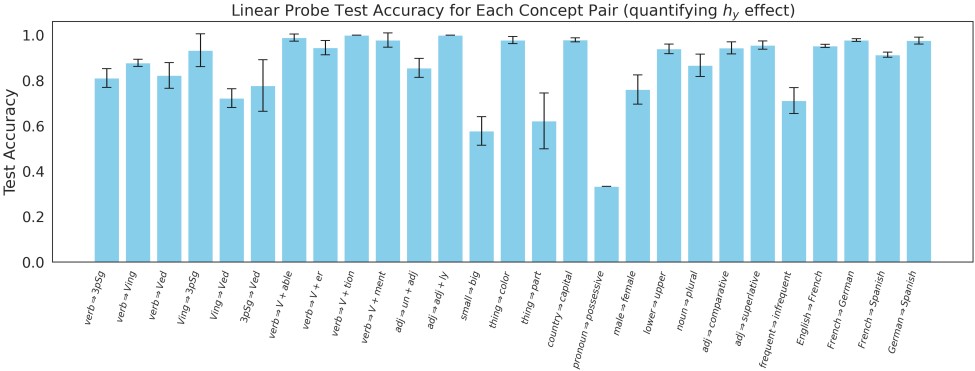

Figure 18: Testing performance of linear probing on Pythia-1.4b model.

## P    SENSITIVITY OF SAE EVALUATION TO LINEAR PROBE QUALITY

In principle, training a linear probe to approximate the ideal posterior $p(c|\mathbf{x})$ for evaluating SAE features requires ideal counterfactual samples. In practice, however, we often face several challenges, such as the choice of regularization parameters in logistic regression, imbalanced classes, and label

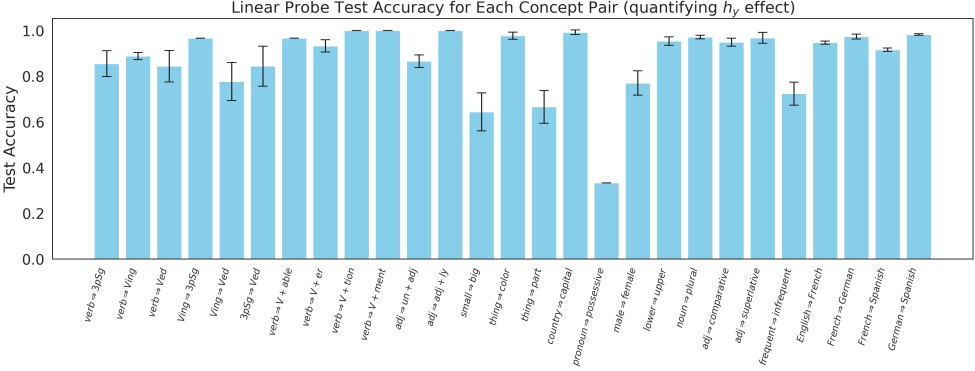

Figure 19: Testing performance of linear probing on Pythia-2.8b model.

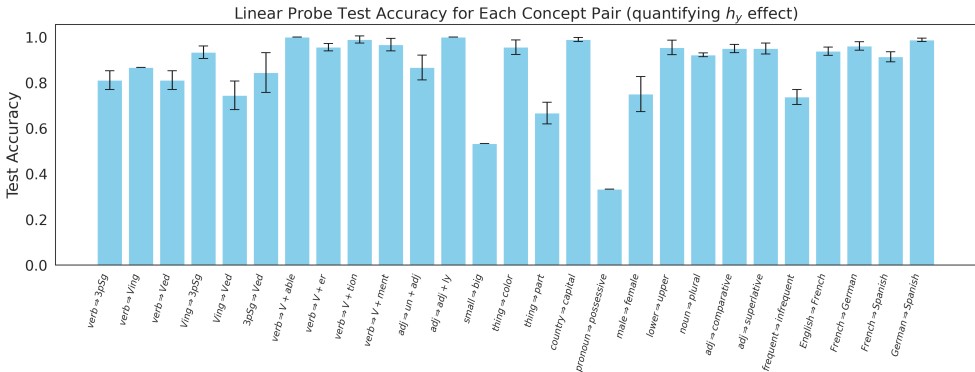

Figure 20: Testing performance of linear probing on Pythia-6.9b model.

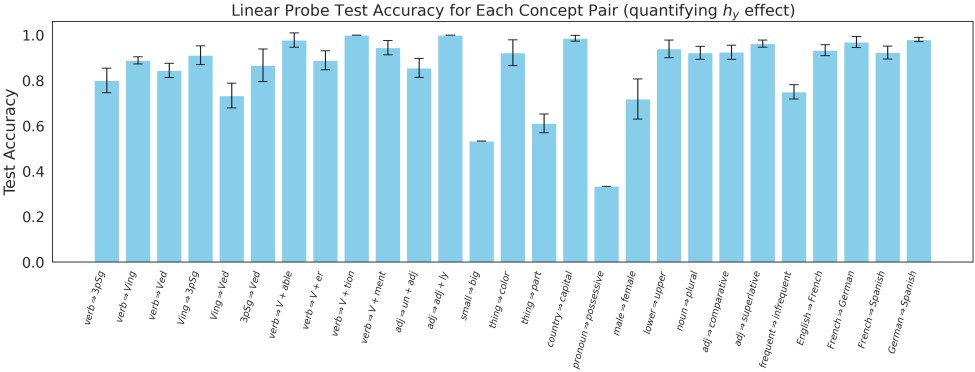

Figure 21: Testing performance of linear probing on Pythia-12b model.

noise. To assess the sensitivity of SAE evaluation to these factors, we conducted experiments varying the probe's regularization strength (LogisticRegression in SKlearn, C=0.1,0.01,0.001, respectively, baseline is C=1), class imbalance (proportion of positive labels $y = 1$) set to 0.7, 0.8, 0.9, implemented by 'class_weight' parameter in LogisticRegression in scikit-learn, baseline balanced at 1), and label noise (5%, 10%, 15% random label flips; baseline 0%). The results are shown in Figure 22. We observe that the Pearson correlation remains stable across different regularization strengths, which is expected since a linear classifier with $l_2$ regularization typically exhibits robustness to variations in strength. In contrast, for varying class imbalance and label noise, there is a slight decrease in Pearson

correlation. This behavior may be reasonable, as both class imbalance and label noise may introduce additional variability in the learned logits, affecting the evaluation.

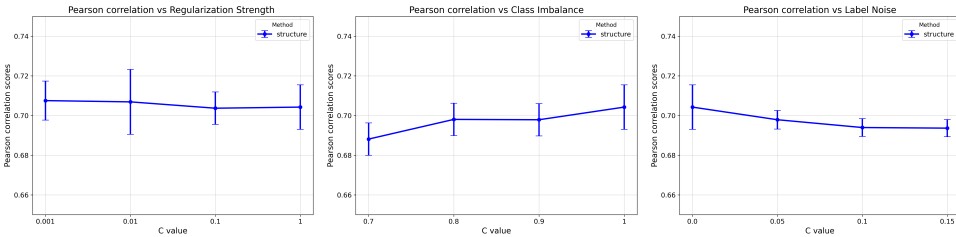

Figure 22: Pearson correlation across regularization strength (left), class imbalance (middle), and Noise Levels (right).

## Q  EMPIRICAL VERIFICATION OF THE TV AND APPROXIMATE INVERTIBILITY

Our theoretical analysis in Section 3 shows that the key remainder term in the decomposition $\Delta h = h_{y_i} - h_{y_0}$ is governed primarily by two quantities: (i) the total-variation distance between the posteriors $p(\mathbf{c} \mid y_i)$ and $p(\mathbf{c} \mid y_0)$, and (ii) the concentration of the conditional posterior $p(\mathbf{c} \mid \mathbf{x}, y)$, captured by the parameter $\epsilon$ in the approximate-invertibility assumption. To assess the strength of these assumptions in practice, we conduct an empirical study on real LLM representations.

**Empirical proxy for the posterior.**  We train 27 linear probes on the counterfactual concept pairs introduced earlier, and stack their prediction probabilities to obtain a normalized vector over the 27 concepts. This provides a *pseudo-posterior* over the probed concept subspace, which we treat purely as a practical proxy for $p(\mathbf{c} \mid \mathbf{x})$. Because the next-token $y$ depends solely on the input $\mathbf{x}$ in the autoregressive setting, the variation of this pseudo-posterior across input contexts serves as an empirical surrogate for the variation of $p(\mathbf{c} \mid y)$. Across all 27 concepts, the resulting TV distances are small (Table 6), supporting the required TV condition in our remainder bound.

**Posterior concentration proxy.**  For each instance $\mathbf{x}$, we measure how concentrated the pseudo-posterior is around its dominant concept via

$$\widehat{\epsilon}(\mathbf{x}) := 1 - \max_{\mathbf{c}} \widehat{p}(\mathbf{c} \mid \mathbf{x}).$$

This quantity is not intended to estimate the true global invertibility parameter, but rather to provide a conservative subspace-based proxy. Following the reviewer's suggestion, we report the 5% quantile

$$\widehat{\epsilon}_{\mathrm{q05}} := \mathrm{Quantile}_{0.05}\big(\widehat{\epsilon}(\mathbf{x})\big),$$

which captures concentration on the high-mass portion of the distribution. As shown in Table 6, this proxy remains small for all concepts.

**Remainder control.**  The theoretical remainder term takes the form

$$\mathrm{TV}\big(p(\mathbf{c} \mid y_i), p(\mathbf{c} \mid y_0)\big) \cdot |\log \epsilon|.$$

Using the empirical TV values and the conservative estimate $|\log \widehat{\epsilon}_{\mathrm{q05}}|$, we compute this product for each of the 27 concepts. Across the board (Table 6), the product remains small, indicating that both factors in the theoretical remainder are simultaneously well-behaved. This suggests that the remainder term is negligible for real LLM representations, reinforcing the practical relevance of our identifiability analysis.

## R  ADDITIONAL COUNTERFACTUAL PAIR ANALYSIS ON PYTHIA MODELS

To further strengthen the generality of our findings, we extend our evaluation, beyond morphology- and translation-based contrasts in the used 27 pairs before, by constructing 8 new counterfactual pairs.

Table 6: Empirical estimates of the TV distance to the anchor concept and the combined remainder proxy TV $\cdot |\log \widehat{\epsilon}_{q05}|$ for all 27 concepts. The global $\widehat{\epsilon}$ proxy is $\widehat{\epsilon}_{q05} = 0.7473$ with $|\log \widehat{\epsilon}_{q05}| = 0.2913$. Note: Concept 21 is selected as the anchor concept (corresponding to $y_0$).

| \multicolumn{3}{c}{Concepts 0–13} | | \multicolumn{3}{c}{Concepts 14–26} | | |
| ID | TV | TV$\cdot|\log\widehat{\epsilon}|$ | ID | TV | TV$\cdot|\log\widehat{\epsilon}|$ |
|---|---|---|---|---|---|
| 0 | 0.1336 | 0.03893 | 14 | 0.1989 | 0.05794 |
| 1 | 0.1721 | 0.05014 | 15 | 0.2073 | 0.06040 |
| 2 | 0.2259 | 0.06581 | 16 | 0.1778 | 0.05181 |
| 3 | 0.2398 | 0.06985 | 17 | 0.1430 | 0.04165 |
| 4 | 0.2389 | 0.06958 | 18 | 0.1821 | 0.05305 |
| 5 | 0.1206 | 0.03513 | 19 | 0.1386 | 0.04037 |
| 6 | 0.1005 | 0.02928 | 20 | 0.1427 | 0.04157 |
| 7 | 0.1721 | 0.05013 | 21 | 0.0000 | 0.00000 |
| 8 | 0.1358 | 0.03957 | 22 | 0.1158 | 0.03374 |
| 9 | 0.1774 | 0.05168 | 23 | 0.1034 | 0.03011 |
| 10 | 0.1455 | 0.04238 | 24 | 0.1124 | 0.03273 |
| 11 | 0.2940 | 0.08565 | 25 | 0.0857 | 0.02497 |
| 12 | 0.0933 | 0.02717 | 26 | 0.0754 | 0.02196 |
| 13 | 0.1548 | 0.04510 | | | |

These include positive vs negative (e.g., good vs bad), truth vs falsity (authentic, fake), and toxic vs neutral (e.g., dangerous, harmless). Constructing such counterfactual pairs is, however, nontrivial and inherently challenging. This constructing process may often introduce additional variability and noise. As a result, the pairs may deviate from the idealized assumption of the case that a single latent concept changing while all other factors remain fixed.

Importantly, the presence of such noise does not undermine our analysis, rather, it provides an opportunity to assess the robustness of our results under realistic, noisy conditions.

We therefore run the evaluation pipeline, including the cross-validation procedure and the null baseline protocol (similar to Section N), on these newly constructed 8 pairs. Overall, the results, shown in Figure 23, exhibit the same qualitative patterns as the original experiments, indicating that our findings remain robust even when additional noise may be introduced. This further supports the broader claim that our framework provides a unified perspective on concept directions, probing, and representation steering.

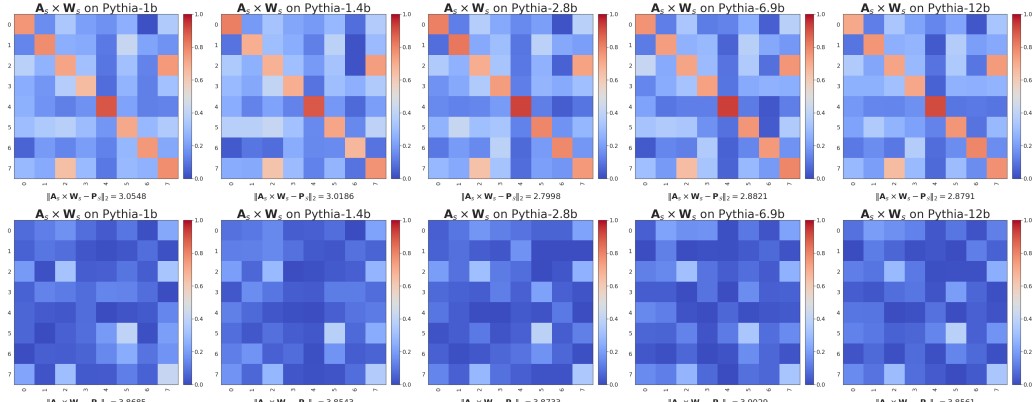

Figure 23: Results of the product $\mathbf{A}_s \times \mathbf{W}_s$ under Cross-Validation (top) and Null Baseline (bottom) on new pair data. The cross-validation ensures that the observed effect generalizes to held-out embeddings, while the null baseline provides a quantitative reference to rule out spurious alignment.

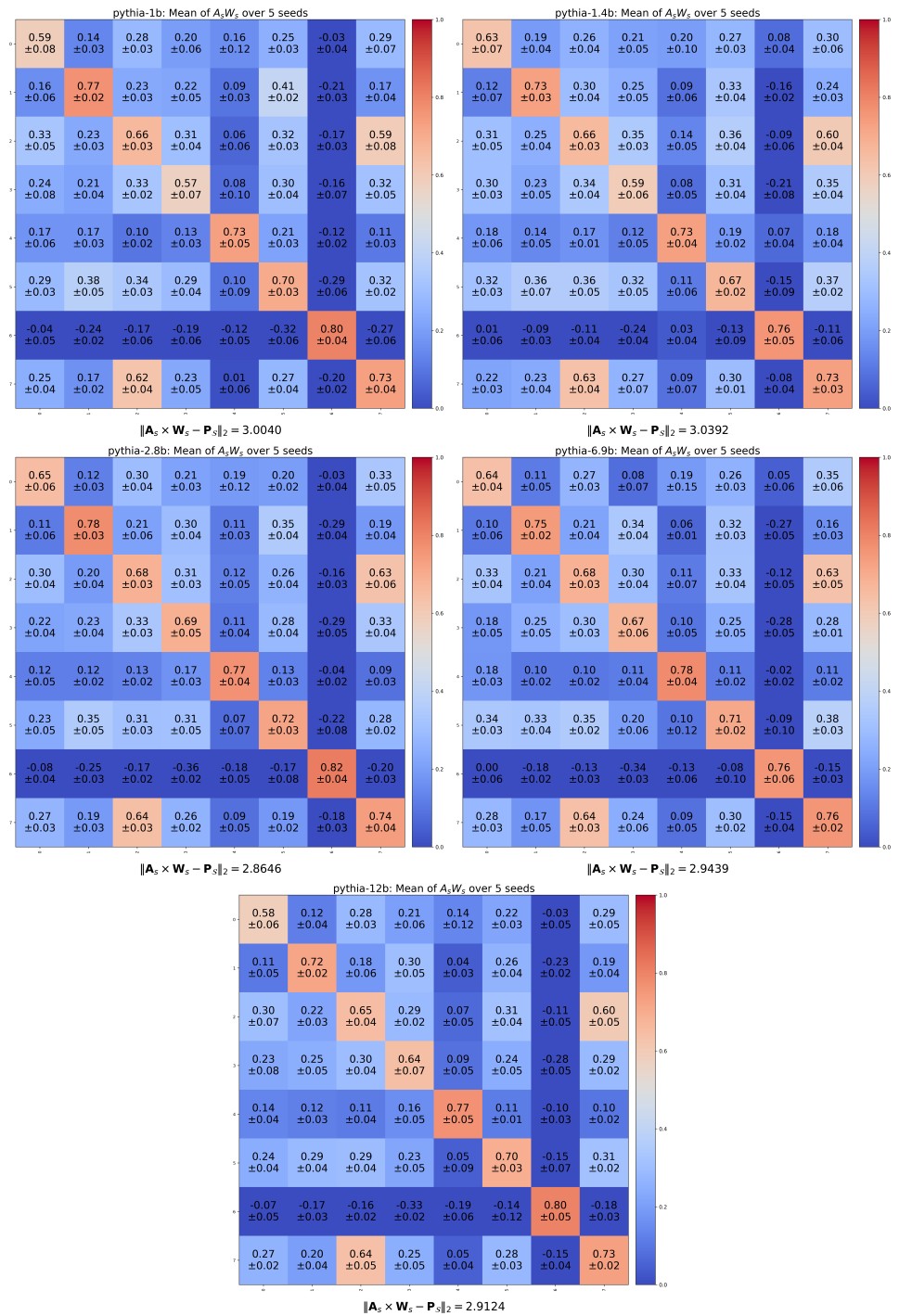

Figure 24: Results of the product $\mathbf{A}_s \times \mathbf{W}_s$ on new pair data across different seeds.

