# OpenReview forum: "I Predict Therefore I Am: Is Next Token Prediction Enough to Learn Human-Interpretable Concepts from Data?"
_ICLR.cc/2026/Conference — ICLR 2026 Poster_

### Official Review · Reviewer_YUaf · 2025-10-27

**Soundness:** 3
**Presentation:** 3
**Contribution:** 3
**Rating:** 6
**Confidence:** 4

**Summary:**

The authors study why next-token prediction (NTP) models appear to learn human‑interpretable concepts. The authors posit a discrete latent-variable generative model in which concepts c (discrete, possibly with arbitrary causal structure) generate input x and target y. Under a diversity condition and without requiring invertibility of the mapping from c to observations, they prove an identifiability result: the model’s representation f_x(x) is (approximately) an invertible linear transform of \log p(c \mid x) up to additive terms. This provides a unified account of “linear representation” phenomena (steering vectors, linear probing) and motivates a principled evaluation of SAEs by comparing their features to linear-probe estimates of concept posteriors. The empirical section includes synthetic validations and tests across Pythia, Llama, and DeepSeek families.

**Strengths:**

* Clear theoretical link from NTP to concept posteriors via a linear transform (Theorem 3.1), with corollaries explaining steering/probing phenomena.
* Actionable implication for evaluating SAEs by benchmarking against linear-probe posterior estimates.
* Good validation practices: synthetic and multiple model families (Pythia, Llama, DeepSeek).

**Weaknesses:**

* Assumptions strength: The diversity condition is nontrivial and may not hold; more intuition or empirical checks would help.
* Residual terms: The correction term in Theorem 3.1 (involving h_y) complicates the neat linear story; Can you quantify when is it negligible?
* Empirical depth: SAE evaluation is promising but would benefit from a broader battery (multiple SAE variants, structured sparsity, stability across seeds).

**Questions:**

* Can you provide sufficient conditions (in practice) under which the diversity condition holds for token distributions in large corpora?
* How large is the h_y correction term in typical NTP models? Can you report empirical magnitudes and the impact on linearity fits?
* In the SAE evaluation recipe, how sensitive are conclusions to the quality of linear probes (regularization, class imbalance, definition of concept labels)?

---

> ### Author Response · Authors · 2025-11-22
>
> We sincerely appreciate you taking the time to review our work.
>
> ----
>
> **Q1: Assumptions strength: The diversity condition is nontrivial and may not hold**
>
> **R1:** Thanks for your comments! This concern has also been mentioned by reviewers TJva and 2RWt. In the new diversity condition, we only need $\ell'+1$ different $y_i$ to enable the matrix $\mathbf{\hat h}$ to be invertible, where $\ell'$ denote the representation dimension of LLMs. Further, as we highlighted in the new version, this new diversity assumption is generally mild: as noted by Roeder et al. (2021), the probability that randomly initialized and stochastically updated parameters of $\mathbf{f}_y$ produce linearly dependent vectors is effectively zero.
>
> **Q2: large is the $\mathbf{h}_y$ correction term in typical NTP models? Can you report empirical magnitudes and the impact on linearity fits?**
>
> **R2:** Thank you for the insightful question. We agree that the correction term $\mathbf{h}_y$ plays an important role in our linear result, and that its empirical magnitude should be quantified. Ideally, one would compare two LLMs that differ only in the presence or absence of the term in order to conduct a clean ablation. However, given a pretrained model, it is not feasible to construct a comparable model in which only the hy information is removed while all others remain unchanged.
>
> Nevertheless, we can adopt a surrogate method that captures the practical effect of the term. According to our theoretical result (e.g., Eq. 5), if $\mathbf{h}_y=\mathbf{0}$, then the representation should satisfy the ideal linear relation, and therefore a linear probe should achieve strong performance on concept discrimination (e.g., Corollary 4.3). Conversely, if the term $\mathbf{h}_y$ significantly perturbs the representation, the linear probe’s accuracy should degrade correspondingly.
>
> Based on this insight, we design an experiment in which the test accuracy of a linear probe serves as an empirical indicator of the magnitude of the term correction. Strong linear-probe performance  may suggest that the term has limited influence, while lower accuracy implies that the correction term contributes noticeable deviation from the ideal linear structure. Please see Section O in Appendix in the new version. Note that for concept 'pronoun - possessive', the sample size is extremely small, so the resulting accuracy may not be informative. Overall, the repeated-run analysis demonstrates that our main empirical findings are
> robust, and provides quantitative evidence that the term has limited influence.
>
> Thanks again for this comment, which was very helpful in guiding us to design an experiment that strengthens our theoretical results.
>
>
> **Q3: In the SAE evaluation recipe, how sensitive are conclusions to the quality of linear probes (regularization, class imbalance, definition of concept labels)?**
>
> **R3:** Thank you for the suggestion. To evaluate the sensitivity of our SAE evaluation to the quality of linear probes, we conducted experiments varying three aspects of the probes: Regularization: baseline (C=1 model default), and perturbed C=0.1,0.01,0.001. Class imbalance: baseline (C=1 balanced classes), and perturbed with positive label proportion set to c=0.7, 0.8, and 0.9 Label noise: baseline (C=0 no label noise), and perturbed with noise rates C=0.05,0.1,0.15. These experiments allow us to assess how conclusions about SAE features depend on probe quality. The results can be found in Figure 22 in Section P in the new version. We found: Pearson correlation is stable across different regularization strengths, indicating robustness of the linear $l_2$-regularized classifier. Slight decreases are observed under class imbalance and label noise, which is expected due to the added variability in the data.

---

### Official Review · Reviewer_2RWt · 2025-10-28

**Soundness:** 2
**Presentation:** 3
**Contribution:** 2
**Rating:** 4
**Confidence:** 3

**Summary:**

This paper proposes a theoretical framework connecting next-token prediction with latent variable identifiability. It argues that the representation in large language models approximates a linear transformation of the log posterior of latent concepts. The authors derive this relationship analytically and validate it through small-scale experiments showing linear alignment between counterfactual representation differences and classifier weights.

**Strengths:**

* The paper provides an interesting conceptual link between next-token prediction, linear probing, and identifiability theory.
* It provides a single, unified mathematical explanation for a diverse set of empirical phenomena (linear probing, concept steering, and vector arithmetic).
* They test their theory on synthetic data where the ground truth $c$ is known, and then successfully test its corollaries (e.g., $A_s W_s \approx I$) across multiple LLM families and sizes.

**Weaknesses:**

The paper presents an interesting finding that LLM representations could be linear latent concept posteriors. However, some assumptions may be too strong and make the analysis flawed.
*  The main theorem (Theorem 3.1) critically depends on the invertibility of the matrix $\hat{L}=[f_{y}(y_{1})-f_{y}(y_{0}), ..., f_{y}(y_{l})-f_{y}(y_{0})]$, which implies that the representation dimension d must equal the total number of latent configurations $l$. In practice, $l$ grows combinatorially explosive (e.g., $2^m$ for m binary concepts), while d (e.g., 4096) is a fixed hyperparameter. This d<<l seems to invalidate the proof.
* Also, the diversity condition, requires the existence of $l+1$ distinct values of y for the matrices L and $\hat{L}$ to be constructed.  $l$ is combinatorially explosive (e.g., $2^m$ for m binary concepts) and will be much larger than the vocabulary size |V| for any non-trivial concept space. And it is impossible to find $l+1$ if $l$ > |V|.
* The latent variable model is  $p(x,y)=\sum_{c}p(x|c)p(y|c)p(c)$. This assumes that the context $x$ and the next token $y$ are conditionally independent given the latent concept $c$. This is a simplification that breaks the autoregressive nature of language, where $x$ (past tokens) is the direct causal parent of $y$ (the next token).

**Questions:**

* Could the authors please clarify how the theory (theorem 3.1) can hold in the realistic setting where $d \ll l$?
* The theory requires $l+1$ distinct $y$ tokens. How can this condition possibly be satisfied in practice, where $l$ is combinatorially large and $l \gg |V|$?
* The latent variable model assumes $x$ and $y$ are conditionally independent given $c$. How does this simplification, which breaks the autoregressive nature of language, where $x$ (past tokens) is the direct causal parent of $y$ (the next token), affect the main claim? Would the linear identifiability result still hold under a more standard model?

---

> ### Author Response · Authors · 2025-11-22
>
> Before responding, we would like to express our sincere appreciation for your comments. Your concerns (similar to those raised by Reviewer TJva) regarding the strength of our assumptions have truly motivated us to relax them.
>
> ----
>
> **Q1: Given that the representation dimension, total number of latent configurations, and vocabulary size are denoted by $d$, $l$, and |V|, repsectively, in the realistic setting, how can the conditions $d<\ell$, and $\ell>|V|$ hold?**
>
> **R1:** Thanks for your comments. Motivated by your feedback, we have relaxed this assumption, please see the diversity condition in the new draft. In the new diversity condition, we only need $\ell'+1$ different $y_i$ to enable the matrix $\mathbf{\hat h}$ to be invertible, where $\ell'$ denote the representation dimension of LLMs. This relaxation has two important consequences: 1) We no longer require the representation dimension to match the total number of latent configurations. 2) We no longer rely on constructing the matrix $\mathbf{L}$ in the original assumption (This requirement is now replaced by a new TV (total variation) condition introduced after the Diversity Condition in the revised version. If you are interested in this condition, please refer to our response to Reviewer TJva.).
>
> After reconsidering your concerns, we re-examined the broader picture and realized that the subsequent corollaries and analysis do not actually require the matrix $\mathbf{A}$ to be square and invertible. It now becomes an $\ell' \times \ell$ matrix, where $\ell$ is the number of latent configurations. Thanks a lot for pushing us toward this improvement!
>
> **Q2: The latent variable model assumes $\mathbf{x}$ and $y$ are conditionally independent given $\mathbf{c}$. How does this simplification, which breaks the autoregressive nature of language, where (past tokens) is the direct causal parent of
>  (the next token), affect the main claim? Would the linear identifiability result still hold under a more standard model?**
>
> **R2:** This is an interesting question.
>
> First, we acknowledge that there exist two mainstream perspectives for modeling text generation: one based on latent variable models, as adopted in this work, and the other considering the autoregressive dependency between past tokens and the next token. Since the true underlying data generative model (DGP) for text is unknown (the autoregressive property may not necessarily reflect the DGP), both perspectives provide valid but different modeling assumptions, and thus we do not claim that either is more “standard.” Moreover, our latent variable model does not break autoregressive property: correlations across tokens can still be captured through the latent variables, meaning that the next-token prediction learning framework remains compatible with our formulation.
>
> Second, our work is task-driven: we aim to investigate the relationship between human-interpretable concepts and LLM representations. From this perspective, latent variable models are a natural choice, as they allow concepts to be formulated explicitly as latent variables. This approach aligns with several previous works, as highlighted in the Introduction, and latent variable models have a long-standing history in the machine learning community.
>
> Third, regarding whether the linear identifiability result would still hold under an autoregressive generative model for text, we currently do not have a formal answer. A key challenge is how to properly formulate human-interpretable concepts within such a model, which remains an interesting direction for future work.

---

> > ### Comment · Reviewer_2RWt · 2025-11-26
> >
> > Thank the author for the response and relax the assumption on Q1. But it also brings a new concern regarding Q1, if the matrix A is now $l' \times l$ with $l' << l$, the system is underdetermined. While this fixes the feasibility issue, does it not fundamentally undermine the 'identifiability' claim? Because it seems to represent a huge info loss and mathematically one cannot uniquely recover the high-dimensional posterior from the low-dimensional representation.

---

> ### Author Response · Authors · 2025-11-26
>
> Thanks for your further comments!
>
> **Q1: Does it not fundamentally undermine the 'identifiability' claim?**
>
> **R1:** Based on the context you provided, we assume that the term “identifiability” you are referring to corresponds to component-wise identifiability of latent concepts. That is, whether latent variables can uniquely recovered up to a trivial element-wise transformation. Please let us know if we have misunderstood.
>
> We would like to clarify that throughout this work, we emphasize the following:
> the representations learned by LLMs through next-token prediction can be approximately modeled as the logarithm of the posterior probabilities of latent discrete concepts given the input context, **up to a linear transformation**. Also, in our formulation, identifiability is formally defined in Theorem 3.1.
>
> ----
>
> Best,

---

### Official Review · Reviewer_wuUB · 2025-10-29

**Soundness:** 3
**Presentation:** 3
**Contribution:** 2
**Rating:** 6
**Confidence:** 4

**Summary:**

This paper theoretically shows that next-token prediction leads to linear identifiability of data-generating latents under a fairly general data model. In particular, such identifiability could be obtained when the generation function is non-linear and non-invertible, and when the latents are discrete and non-Boolean. Motivated by this result, the paper proposes improved techniques for evaluating and training sparse autoencoders and empirically evaluates the proposed method both in synthetic settings and on practical LLMs.

**Strengths:**

- Compared to existing analysis, the identifiability result proved by this paper is more general; the introduced diversity condition makes sense to me and is likely to hold in practical pre-training of LLMs. Showing approximated identifiability in the non-invertible case is also a decent complement to the results.
- The paper translates the theoretical results to practical algorithms on SAEs.
- The paper is well-written and easy to follow.

**Weaknesses:**

- A major drawback of the results, like existing ones, is that linearity is defined only for the output activations (i.e., in the unembedding space) of LLMs. Yet, prior empirical work on the linear representation hypothesis often shows that linearity is usually formed in the _intermediate_ layers [1, 2]. For some concepts, such intermediate-layer linearity could even fade away at the output layer [3], which cannot be captured by the proved results. Although I agree with the authors that rigorously analyzing linearity for intermediate layers is (much) more difficult, I feel it is still necessary to mention it here since it may be of more interest for both theorists and practitioners (e.g., SAEs are often trained on intermediate-layer outputs rather than last-layer outputs), in my opinion.
- The relation between theoretical results and the proposed structured SAE is rather loose (see Questions for more details).
- (Minor) Missing related work: [4] also proves linear identifiability of latent variables under a general setting and discusses the implications for the linear representation hypothesis.

---

[1] Language models represent space and time. ICLR, 2024.

[2] Emergent linear representations in world models of self-supervised sequence models. arXiv preprint arXiv:2309.00941, 2023.

[3] The geometry of truth: Emergent linear structure in large language model representations of true/false datasets. COLM, 2024.

[4] When do neural networks learn world models? ICML, 2025.

**Questions:**

- While discrete concepts are beneficial for modeling text semantics such as topics, there may also be continuous concepts in the text, such as those relevant to numbers. Can the identifiability result be extended to modeling both discrete and continuous latents?
- What is the difference between the proposed SAE evaluation approach and existing probing approaches, e.g., the one used in [5]?
- Why does low-rank regularization help model the interdependence between latent variables/concepts in structured SAE?

---

[5] Scaling and evaluating sparse autoencoders. arXiv preprint arXiv:2406.04093, 2024.

---

> ### Author Response · Authors · 2025-11-22
>
> Thanks so much for taking the time to review our work.
>
> ----
>
>
> **Q1: Mention prior studies showing linear structure in intermediate layers**
>
> **R1:** Thanks for pointing this out. Indeed, as we highlighted in the Limitations section (Section B in the Appendix), one of the main limitations of this work is that it is “limited to the last layer in LLMs and does not provide a justification for intermediate layers.” Following your suggestion, we have added additional discussion in this section to acknowledge this point.
>
> **Q2: Missing related work: [4]**
>
> **R2:** We have added a discussion with [4] in related work. Thanks a lot. Our work also differs from [4], which studies identifiability through multi-task learning for the latent variables of latent variable models, where the mapping from latent to observsation to be invertible. In contrast, we focus specifically on the next-token prediction framework in LLMs, for a different latent variable, which allow the mapping to be non-invertible.
>
> **Q3: Can the identifiability result be extended to modeling both discrete and continuous latents?**
>
> **R3:** Good question. We view this as a promising avenue for future work. We believe that the next-token prediction objective has a potential be compatible with continuous or hybrid latent structures, but establishing identifiability in such settings would likely require substantial new assumptions and proof techniques beyond the scope of the present paper. In fact, we are exploring a possible solution:  continuous latent variables could sometimes be approximated or discretized (e.g., via quantization or binning). However, such discretization must be handled with care: Coarse discretization may collapse distinct continuous states and destroy identifiability. Fine discretization may approximate the continuous model but introduces an effectively large discrete latent space, which may requires stronger conditions.
>
> **Q4: What is the difference between the proposed SAE evaluation approach and existing probing approaches, e.g., the one used in [5]?**
>
> **R4:** The key differences are twofold. First, their method is largely heuristic, evaluating latent features based on their correlation with task labels without formal theoretical grounding. In contrast, our approach is theory-motivated, leveraging the identifiability results established in this work. Second, their evaluation relies on a strong assumption about which features are considered `natural', as mentioned in their work. This means that features merely linearly correlated with task labels may achieve good classification performance, yet the evaluation may fail to determine whether the features truly encode the intended concepts. In our method, we use counterfactual paired data to train linear probes, directly measuring whether each SAE feature captures the target concept.
>
> **Q5: Why does low-rank regularization help model the interdependence between latent variables/concepts in structured SAE?**
>
> **R5:** Extracting features from LLM representations is typically an inverse problem. In general, without any prior regularization, there may exist many possible solutions. While sparsity is a commonly used regularizer, motivated primarily by the superposition hypothesis. Using sparsity alone may be insufficient to constrain the solution space. Therefore, we consider additional regularization. As we mentioned in the paper, the complex dependencies in text data are encoded by the latent variables, making it natural to introduce a structured prior to model these dependencies. Low-rank regularization serves as a structural prior, encouraging the model to recognize and capture relationships among latent concepts, which may reduce the solution space further.

---

### Official Review · Reviewer_TJva · 2025-11-01

**Soundness:** 2
**Presentation:** 3
**Contribution:** 3
**Rating:** 4
**Confidence:** 3

**Summary:**

The paper models text generation via discrete, human-interpretable concepts and shows, under a diversity/invertibility condition, that accurate next-token predictors' internal features are (up to an invertible linear map and bias) equal to the log posterior over those concepts. This yields a unified explanation for linear probing, steering, and "concept directions," and motivates a practical test (alignment between probe weights and concept directions). The authors add small-scale simulations, apply the alignment test to several LLMs with hand-crafted counterfactual pairs, and propose a structured sparse-autoencoder (SAE) plus a probe-based evaluation that outperform common SAE baselines on correlation/MSE.

**Strengths:**

**Unifying lens on ``linear representations.”:** The same matrix $A$ explains concept directions, steering, and probe linearity, which clarifies why these tricks often work together.

**Simple checks in evals:** The $A_s W_s \approx I$ is a neat diagnostic. I haven’t seen exactly this identity‑product visualization before.

**Practical tie‑in to SAEs:** The theory suggests a supervised reference for evaluating SAE features and motivates a structured (sparse+low‑rank) variant that shows consistent gains

**Cross‑model qualitative replication of concept–probe alignment**

**Weaknesses:**

**The diversity condition seems very strong and under-discussed**
From what I understand, in Appendix D, the paper implicitly assumes $f_x(x), f_y(y) \in \mathbb{R}^\ell$ so that $\hat{L} \in \mathbb{R}^{\ell \times \ell}$ exists with columns  $f_y(y_j) - f_y(y_0).$

That means the representation dimension must be at least  $\ell = \prod_k n_k$  (the number of joint latent configurations). This seems like a very strong and unrealistic assumption and the manuscript never states this dimensionality constraint explicitly.

To illustrate, even for a toy model with just 20 binary concepts ($k=20, n_k=2$), $\ell = 2^{20} \approx 1 \text{ million}$. The representation dimension $d$ of a real LLM is large, but fixed (e.g., $d=4096$ or $d=8192$). In any realistic scenario, the representation dimension $d$ is *far smaller* than the number of possible joint concepts $\ell$ ($d \ll \ell$).

If $d = \mathrm{dim}\, f_x \neq \ell,$ you need to formulate everything on the span of  $\{ f_y(y_j) - f_y(y_0) \}$ and use for example the Moore–Penrose pseudoinverse on a $d \times \ell$ matrix; otherwise the shapes don’t line up (you would have a non-square matrix). As written, the proof seems to treat $\hat{L}$ as square without justifying $d = \ell.$ (Same comment for $L$ as well.) The authors state in Appendix C that this is a ``weak assumption" so please correct me if I'm not understanding something here. I think the proof should go through with a pseudo-inverse but I haven't worked out the details.

**Again in appendix D, on the approximately invertible part:**
I am not sure if the argument that the remainder term $\mathbf{h}_y$ vanishes (Pg 20) is correct.

The term is $h_{y_j} = \mathbb{E}_{p(\mathbf{c} \mid y_j)} [\log p(\mathbf{c} \mid \mathbf{x}, y_j)]$. The proof argues this vanishes because $p(\mathbf{c} \mid \mathbf{x}, y_j)$ concentrates at a single $\mathbf{c}^*$ (the $\epsilon$-invertibility assumption).

However, the expectation is taken with respect to $p(\mathbf{c} \mid y_j)$, not $p(\mathbf{c} \mid \mathbf{x}, y_j)$. If I split the expectation:
$$
h_{y_j} = p(\mathbf{c}^* \mid y_j) \log p(\mathbf{c}^* \mid \mathbf{x}, y_j) +  \sum^{\mathbf{c} \neq \mathbf{c}^*} p(\mathbf{c} \mid y_j) \log p(\mathbf{c} \mid \mathbf{x}, y_j)
$$

As $\epsilon \to 0$, we have $\log p(\mathbf{c}^* \mid \mathbf{x}, y_j) \approx \log(1-\epsilon) \approx 0$, but $\log p(\mathbf{c} \neq \mathbf{c}^* \mid \mathbf{x}, y_j) \approx \log \epsilon$.

This seems to imply that the expression for $h_{y_j}$ becomes:
$$
h_{y_j} \approx \sum^{\mathbf{c} \neq \mathbf{c}^*} p(\mathbf{c} \mid y_j) (\log \epsilon)
$$

$$ = (1 - p(\mathbf{c}^* \mid y_j)) \log \epsilon $$

This term clearly depends on $y_j$ (via $p(\mathbf{c}^* \mid y_j)$) and diverges to $-\infty$ as $\epsilon \to 0$.

Therefore, the difference $h_{y_j} - h_{y_0}$ seems to be:
$$
h_{y_j} - h_{y_0} \approx \big( p(\mathbf{c}^* \mid y_0) - p(\mathbf{c}^* \mid y_j) \big) \cdot \log \epsilon
$$

Instead of vanishing, this term seems to diverge, given that the diversity condition requires $p(\mathbf{c} \mid y_0) \neq p(\mathbf{c} \mid y_j)$. The argument on Pg 20 that $\mathbb{E}[\log p(\mathbf{c} \mid \mathbf{x}, y)] \approx \log \epsilon$ "not depending on $y$" appears to miss the $(1 - p(\mathbf{c}^* \mid y_j))$ factor.

This feels like a significant error in the proof, so I wanted to check if I'm missing a different assumption. To make this term vanish, it seems you would need to assume $p(\mathbf{c}^* \mid y_j)$ is constant for all $j=0...\ell$, which seems like a very strong constraint that might conflict with the diversity condition.


**Some concerns about experimental evals**
Using the same pairs to build both $A_s$ (directions) and $W_s$ (probes) can inflate diagonal alignment. There should at least be some reporting over held-out concepts or perform cross-validation, and include null baselines (say, randomly permuted labels) to show that the effect isn’t a trivial consequence of using the same supervision twice.

Also, quantifying ``approximate identity” by reporting, layer-wise and across seeds, something like diagonal/off-diagonal gap might help (just an idea).

**Contextualize Structured SAEs**
I like the Structured SAEs idea but low‑rank‑plus‑sparse decompositions with a nuclear‑norm regularizer have a deep literature (e.g., Candes et al., ``Robust PCA”; many variants since). Bringing L+S to SAEs is reasonable but should be contextualized and credited; please cite this prior line of work explicitly (no relation to me) and clarify what is new here (architecture? training recipe? results on LM activations?)

**Questions:**

For the "Structured SAE," the authors chose nuclear norm regularization to model concept dependencies. What was the intuition behind this specific choice (implying a low-rank structure on concept activations) versus other forms of structured regularization, such as imposing a graphical model prior on the features $\mathbf{z}$ or using a group-lasso penalty? Just a curiosity rather than a weakness of the paper.

---

> ### Author Response · Authors · 2025-11-22
>
> Before addressing your comments, we would like to sincerely thank you for your careful review and valuable feedback. We greatly appreciate the time and effort you took to examine our proofs and point out potential weaknesses. Your comments have motivated us to reconsider our initial draft from a higher-level, big-picture perspective, which has substantially helped us improve the quality of this work. In particular, your feedback has guided us to relax certain assumptions (the fact is that we does not need the matrix $\mathbf{A}$ to be square or invertible to establish the subsequent results following Theorem 3.1), strengthen our proofs, and enhance our experimental evidence.
>
> **Q1:the representation dimension d is far smaller than the number of possible joint concepts.**
>
> **R1:** We have relaxed this assumption to the case where the dimension (row or column) of the square matrix $\mathbf{\hat L}$ could be the representation dimension of LLMs, see Diversity Condition in the new version. In this context, the matrix $\mathbf{A}$ is no longer square (pleasse see Eqs. (20)–(24) in the new version). Importantly, even though $\mathbf{A}$ is non-square and hence non-invertible, we verified that all subsequent corollaries still hold. This is because these results rely only on (i) extracting sub-components of via vector differences, or (ii) left-multiplying $\mathbf{A}$ by a weight matrix in linear probing to extract a concept. We greatly appreciate your comment for helping us clarify this point.
>
> **Q2: The construction of the matrix $\mathbf{{h}}$ with dimension $\ell$.**
>
> **R2:** In the revised draft, we have replaced the original requirement with an alternative condition—the TV (total variation) condition introduced after the Diversity Condition. This new condition essentially requires that the posterior distribution $p(\mathbf{c}\mid y)$ changes slowly with respect to some token $y$. We further elaborate on this condition in our response to your comment regarding the “approximately invertible’’ part.
>
> We have go through the proof in these two new condition, and found that we does not need the matrix $\mathbf{A}$ to be square or invertible to establish the subsequent results following Theorem 3.1. Therefore, no modifications are needed for the following part. Again, thanks a lot!
>
> **Q3: The approximately invertible part**
>
> **R3:** Thanks for pointing out this! We have re-worked the proof in the revised draft (see Page 23). In essence, the new TV condition allows us to bound the diverge caused by the non-invertibility of the mapping from the latent to the observed space (See Eqs (36)-(40) in Page 23). Importantly, when the mapping is assumed to be invertible, the TV condition is not needed. However, when the mapping is non-invertible, the condition is required to suppress the non-dominant points that arise due to non-invertibility (there is no free lunch).
>
> **Q4: Some concerns about experimental evals**
>
> > a) Some reporting over held-out concepts or perform cross-validation, and include null baselines.
>
> **R4(a):** We have added cross-validation results, including null baselines, please see Sec. N in the appendix. Thank you again!. we had not realized the issue before. You are correct, using the same pairs can inflate diagonal alignment, which is aligned with the cross-validation results shown at the top of Figures 7–8 (compared with the result in Figure 3). Additionally, we observe that the null baselines show little to no diagonal alignment, as shown at the bottom of Figures 7–8, which helps strengthen our experimental evidence.
>
> > b) quantifying "approximate identity" by reporting, layer-wise and across seeds
>
> **R4(b):** We have added results across seeds, shown in Figures 9–13. The seeds control how we split the data, and the results appear robust across different random seeds. We do not report layer-wise results, because, in the current draft, our analysis is limited to the last layer, as clarified in the limitations section. It would be interesting to generalize these results to intermediate layers in future work.
>
> **Q5: Contextualize Structured SAEs**
>
> **R5:** We have added citations to prior work on structured sparse decompositions in the Related Work section (Section A of the Appendix), and we agree that this strengthens the context of our work. Importantly, we believe that the proposed Structured SAEs provide a potential bridge between traditional structured-sparsity techniques and the analysis of LLM representations.

---

> > ### Comment · Reviewer_TJva · 2025-11-22
> > **Improved draft but adds a new assumption + needs improvements in some places in clarity/rigor**
> >
> > Thanks for the response. The paper is clearly stronger: the dimension/shape issue is fixed by working on the appropriate subspace; the remainder term is addressed via a TV‑based argument; and the evaluation now includes CV, null baselines, and multiple seeds. I appreciate the clearer scope (last layer) and the improved SAE context.
> >
> > What would still help me get all the way there is to make the key assumption measurable and to tighten a couple claims so they match the non‑square setting. Given the short response window, I’m suggesting changes that are small and mostly should re‑use code you already have.
> >
> > **Remainder term: make the bound explicit and checkable:** The new proof makes the remainder $\Delta h = h_{y_j} - h_{y_0}$ small under (i) near-invertibility of the latent map and (ii) a Total-Variation (TV) condition saying $p(c | y)$ doesn’t move much across the selected anchor tokens.
> >
> > Intuitively this is plausible; however, TV controls movement of $p(c | y)$ while $\Delta h$ involves $\log p(c | x, y)$. If $p(c | x, y)$ is very small in regions where $p(c | y)$ still places non-negligible mass, the log can dominate. In other words, small TV alone doesn’t preclude large contributions from low-probability regions.
> >
> > A simple way to make this explicit is to spell out the decomposition
> > $$ \Delta h = \sum_c \bigl(q_j(c) - q_0(c)\bigr)\log r_j(c) + \sum_c q_0(c)\bigl(\log r_j(c) - \log r_0(c)\bigr) $$
> > with $q_\bullet = p(c \mid y_\bullet)$, $r_\bullet = p(c \mid x, y_\bullet)$.
> >
> > This shows where TV enters (the first sum) and where one also needs a mild coverage idea: "on the parts of $c$-space where $q_\bullet$ has most of its mass, $r_\bullet$ is not vanishing."
> >
> > Empirically, thhere needs to be some changes to make sure the assumption makes sense. For example, one would gain confidence from:
> >
> > - Use your trained probe as a proxy for $p(c\mid y)$ to report an estimated TV across the diversity tokens (a small table).
> > - Report a conservative proxy for $\epsilon$ (e.g., the minimum predicted $p(c\mid x,y)$ on the high‑mass set you actually use; or a quantile). Even a coarse proxy is fine but the goal is to show that, for your token choices, the product “TV $\times \log(1/\epsilon)$” is small.
> >
> > If you can also add one small synthetic plot showing that as you crank the invertibility knob in your toy model, the measured TV proxy shrinks and a remainder proxy declines, that would make the story complete ( but thats optional).
> >
> > **“$AW \approx I \rightarrow$”  identity on the relevant subspace:** Now that this is a non square matrix, results have to be adjusted to be in the relevant **subspace**. The mathematically right statement is "$AW \approx P_S$", i.e., identity on the concept subspace. Concretely, you would need to evaluate against $P_{\mathcal S}$ (identity on the span you actually use) and report a single number like $|AW - P_{\mathcal S}|_F$ (or operator norm), alongside the existing heatmaps. This should just be just a post‑processing change I think.
> >
> > **Concept set is still narrow:** Most tests are morphology/translation. If the claim is broad ("unifies directions, probing, steering"), show at least a few semantic concept families (e.g., sentiment, toxicity, factuality/truthfulness) so it’s not just an artifact of morphological pairs (running the existing CV + null protocol on one representative model should be enough I think)
> >
> > Given the short rebuttal period, I have tried to be realistic with the ask so if these things are done, I thhink the paper becomes more complete and I would be willing to raise my score.

---

> > > ### Author Response · Authors · 2025-11-28
> > >
> > > Again, before response, we would like to express our sincere appreciation for the exceptionally constructive nature of your feedback. We are particularly grateful that you approached the discussion from our perspective as authors and provided suggestions that are both technically insightful and realistically achievable within the tight rebuttal window.
> > >
> > >
> > > **Remainder term: making the bound explicit and checkable.**
> > >
> > > We totally agree that making the remainder term explicit and checkable greatly strengthens the result. We have revised the relevant parts of the theory section and the corresponding proof to reflect this decomposition, and we now clearly articulate how the TV condition and coverage assumption enter. In addition, we have added empirical evaluations (including the TV term and $\log \epsilon$), please see Sec. Q for details. A small synthetic illustration will also be included in the final version. Thanks for your understanding!
> > >
> > > **$AW \approx I$ and “identity on the relevant subspace.**
> > >
> > > Thank you for this very precise observation. We have updated results accordingly, and now additionally report the Frobenius norm, as shown in the revised figures below. This greatly improves clarity, and we appreciate the insightful suggestion.
> > >
> > > **Concept set is narrow; need semantic concept families.**
> > >
> > > Thank you for highlighting this limitation.  We have constructed *8 new counterfactual pairs* covering, such as: (i) positive vs.\ negative sentiment (e.g., good $\Rightarrow$ bad), (ii) truth vs.\ falsity (e.g., authentic $\Rightarrow$ fake), and (iii) toxicity vs.\ neutrality (e.g., dangerous $\Rightarrow$ harmless). Constructing such pairs is nontrivial and inherently challenging, most importantly, additional variability and noise is easily introduced, meaning the resulting pairs may deviate from the idealized case where only a single latent factor changes while all others remain. However, this also provides an opportunity to evaluate the robustness of our result under more realistic, noisy conditions. The full results are presented in Sec.~R, and we find that our conclusions remain stable even when additional noise is introduced.
> > >
> > > ----
> > >
> > > Much appreciated! Your feedback has genuinely helped us strengthen the quality of this work.
> > >
> > > Best,
> > >
> > > Authors

---

> ### Author Response · Authors · 2025-11-22
>
> **Q6: The intuition behind choosing nuclear-norm regularization**
>
> **R6:** Indeed, in our initial design, we considered implementing several popular structured sparsity techniques, including graphical model priors. One of our main concerns, however, is that we train the SAEs with relatively large batch sizes on large-scale text data, and we prefer to use the convenience of standard SGD-based optimization. Exploring other forms of structured sparsity is interesting and could be pursued in future work. In this paper, our primary goal is not to compare the performance of different structured sparsity methods, but rather to demonstrate the feasibility and benefits of incorporating structured sparsity into LLM analysis. Therefore, we chose nuclear-norm regularization as it is simple to implement and works well with our training setup.

---

### Author Response · Authors · 2025-12-02
**Summary**

To the AC:

Following the new guidance regarding the review-process reversion, we are posting this summary to provide a clear and self-contained overview for your final evaluation. We sincerely appreciate your time and effort in handling our submission. Below is a summary of the individual reviewers’ mian concerns and our responses.

**Reviewer TJva**

In the first round, Reviewer TJva provided a careful and technically detailed review. Their main concerns centered on two aspects:

1) Some assumptions in our initial draft appeared strong: Motivated by this feedback, we revisited the theory from a broader perspective and relaxed the assumptions, clarified the dimension/shape issue, and adjusted the proof to the remainder term $\mathbf{h}_y$.

2) Experimental rigor and completeness. The reviewer suggested additional controls, including: cross-validation, null baselines, results across multiple seeds. We carried out all requested improvements.

Following our initial response, the reviewer explicitly acknowledged that “the paper is clearly stronger: the dimension/shape issue is fixed; the remainder term is addressed; and the evaluation now includes CV, null baselines, and multiple seeds.”

The reviewer then suggested several additional, but feasible, refinements: making the bound explicit and checkable, improving the evaluation through post-processing, and adding experiments on a small set of semantic concepts. Importantly, they noted that if these were addressed, the paper would become more complete and they would be willing to raise their score. We have implemented all of these suggestions.

**Reviewer 2RWt**

In the first round, Reviewer 2RWt’s main concerns focused on the strength of several assumptions in our initial draft, points that were highly consistent with those raised by Reviewer TJva. We addressed these by relaxing the assumptions and revising the theoretical setup, as detailed in our response to Reviewer TJva.

After reviewing the revised version, Reviewer 2RWt raised a follow-up concern: under the revised setting, does our claim become unsuitable if “identifiability’’ is interpreted as **element-wise identifiability** (i.e., the ability to recover each latent posterior)? In our response, we clarified that throughout the paper, identifiability is defined in the context that latent posteriors is identified up to **a linear transformation**, as formally defined in Theorem 3.1, and **we do not claim element-wise identifiability**.

**Reviewer wuUB**

Reviewer wuUB’s concerns were primarily about scope and positioning. The questions focused on whether the theory might extend to continuous latents, how our SAE-based evaluation differs from probe-based approaches, and the intuition behind incorporating low-rank regularization. We addressed each of these points in detail in our response.

**Reviewer YUaf**

Reviewer YUaf’s concerns focused on:

1) the strength of assumptions (similar with those raised by Reviewers TJva and 2RWt). In the revised draft, we clarified and relaxed the assumptions.

2) the magnitude of the correction term $\mathbf{h}_y$. We designed a surrogate experiment following the reviewer’s suggestion.

3) the robustness of the SAE evaluation recipe. we conducted a broad sweep over probe regularization strengths, controlled class imbalance, and injected label noise.


-----

Best,

Authors

---

### Meta-Review · Area_Chair_AVYi · 2026-01-06

**Summary:**

This paper proves an identifiability result for models trained with the next-token prediction objective, showing that the learned output-layer representations can be modeled with posterior probabilities of latent data-generating variables up to an invertible linear transformation. The paper connects this result to the linear representation hypothesis of LLMs and practical algorithms such as sparse autoencoders.

Reviewers generally appreciate the significance of theoretical results, their implications for related research areas, and the empirical validation of these results. A common concern raised by the reviewers is the strength of assumptions used in the proof, to which the authors have relaxed some of them and made further clarifications in the rebuttal. Reviewers have also raised some concerns about the rigor of experiments, for which the authors have conducted additional experiments, adding more ablations and baselines. Reviewer wuUB raised concerns about the scope of the paper, especially mentioning that the theoretical result applies only to the final layers of LLMs rather than intermediate layers. The authors acknowledged this limitation and left further study as future work, which I think is acceptable. Overall, most of the concerns raised by reviewers are addressed during the rebuttal, and I agree with the reviewers that the paper provides valuable insights into representation learning of next-token prediction models such as LLMs. I thus recommend acceptance.

**Reviewer Concerns:**

A common concern raised by the reviewers is the strength of assumptions used in the proof, to which the authors have relaxed some of them and made further clarifications in the rebuttal. Reviewers have also raised some concerns about the rigor of experiments, for which the authors have conducted additional experiments, adding more ablations and baselines. Reviewer wuUB raised concerns about the scope of the paper, especially mentioning that the theoretical result applies only to the final layers of LLMs rather than intermediate layers. The authors acknowledged this limitation and left further study as future work, which I think is acceptable.

**Reviewer Scores:**

Reviewers TJva and 2RWt might increase their score from 4 to 6 (Reviewer TJva explicitly mentions this); other reviewers would have kept their original score of 6.

---

### Decision · Program_Chairs · 2026-01-26

Accept (Poster)